# Robust Ecosystem Demography (RED version 1.0): a parsimonious approach to modelling vegetation dynamics in Earth System Models

Arthur P. K. Argles[1], Jonathan R. Moore[1], Chris Huntingford[2], Andrew J. Wiltshire[3], Anna B. Harper[1], Chris D. Jones[3], and Peter M. Cox[1]

[1]College of Engineering, Mathematics, and Physical Sciences, University of Exeter, Exeter EX4 4QF, UK
[2]Centre for Ecology and Hydrology, Wallingford OX10 8BB, UK
[3]Met Office Hadley Centre, Fitzroy Road, Exeter EX1 3PB, UK

**Correspondence:** aa760@exeter.ac.uk, P.M.Cox@exeter.ac.uk, J.Moore3@exeter.ac.uk

**Abstract.** A significant proportion of the uncertainty in climate projections arises from uncertainty in the representation of land carbon uptake. Dynamic Global Vegetation Models (DGVMs) vary in their representations of regrowth and competition for resources, which results in differing responses to changes in atmospheric $CO_2$ and climate. More advanced cohort-based patch models are now becoming established in the latest DGVMs. These models typically attempt to simulate the size-distribution of trees as a function of both tree-size (mass or trunk diameter) and age (time since disturbance). This approach can capture the overall impact of stochastic disturbance events on the forest structure and biomass, but at the cost of increasing the number of parameters and ambiguity when updating the probability density function (pdf) in two-dimensions. Here we present the *Robust Ecosystem Demography (RED)*, in which the pdf is collapsed on to the single dimension of tree mass. RED is designed to retain the ability of more complex cohort DGVMs to represent forest demography, while also being parameter sparse and analytically solvable for the steady-state. The population of each Plant Functional Type (PFT) is partitioned into mass classes with a fixed baseline mortality along with an assumed power-law scaling of growth rate with mass. The analytical equilibrium solutions of RED allow the model to be calibrated against observed forest cover using a single parameter - the ratio of mortality to growth for a tree of a reference mass ($\mu_0$). We show that RED can thus be calibrated to the ESA LC_CCI (European Space Agency Land Cover Climate Change Initiative) coverage dataset for nine PFTs. Using Net Primary Productivity and litter outputs from the UK Earth System Model (UKESM), we are able to diagnose the spatially varying disturbance rates consistent with this observed vegetation map. The analytical form for RED circumnavigates the need to spin-up the numerical model, making it attractive for application in Earth System Models (ESMs). This is especially so given that the model is also highly parameter-sparse.

## 1 Introduction

A key requirement of Earth System Science is to estimate how much carbon the land surface will take-up in the decades ahead (Ciais et al., 2014). This is an important component of the total carbon budget consistent with avoiding global warming thresholds, such as $2°C$ (Schleussner et al., 2016). Unfortunately, projections of future land carbon storage still span a wide-range (Brovkin et al., 2013; Friedlingstein et al., 2014; Arora et al., 2019). Beyond the $CO_2$ and nutrient fertilisation effects and

land-use change, significant uncertainty also arises from the representation of vegetation demographics such as recruitment, compeitition and mortality (Brovkin et al., 2013; Ahlström et al., 2015). The representation of plant communities within Earth System Models (ESMs) is achieved through the use of Dynamic Global Vegetation Models (DGVMs). DGVMs employ a variety of biophysical, biogeographical and biochemical processes to simulate growth, competition and recruitment of vegetation.

The variety in the number and resolution of the processes contributes to the differences found at the Earth System level.

Within the context of modelling vegetation at a global level, there is a trade-off between the complexity of ecological process representation and the necessity of parsimony at scale (Fisher et al., 2018). DGVMs range from the simplistic, older, top-down approaches to that of complex individual-based DGVMs. For example, in the first instance the TRIFFID model (Cox, 2001) simulates the fractional area of each Plant Functional Type (PFT) using phenomenological Lotka-Volterra equations. The ben-

10 efit of the TRIFFID approach is its simplicity and robustness. However, the model suffers from the lack of size representation and other processes which results in the over-estimation of regrowth time (Burton et al., 2019). In the second-instance, individual based models can explicitly represent a multitude of biological and ecosystem processes at an individual plant level (Smith, 2001; Sato et al., 2007). The benefit of this is that size-dependent physiology and spatial heterogeneity can be explicitly represented. However, multiple ensemble-members are often needed to construct meaningful forest statistics, which makes

such models computationally expensive to run at large scales. Compromises between the complexity of individual-based and top-down DGVMs exist as a class of tree *cohort* models. In the ED model (Moorcroft et al., 2001; Medvigy et al., 2009) the tree population is partitioned between patch disturbance and biomass classes allowing for the scaling of process to be represented in both age and size. ED2 can realistically model forests around the world (boreal, rainforest and temperate) (Medvigy et al., 2009; Fisher et al., 2018). However, parameterisation of competition within cohort DGVMs can result in a wide spread

of outcomes when simulating climate change (Fisher et al., 2010; Scheiter et al., 2013).

In a similar vein other models have limited the number of cohort dimensions. The POP model (Haverd et al., 2014), uses stand-age cohorts as the dimension for population dynamics, every time-step applying crowding and resource limited mortality rates. Another example is the ORCHIDEE-MICT (Yue et al., 2018), which disaggregates the populations of a PFT into patch cohort functional types, with transitions between cohorts diagnosed when the average basal diameter passes a threshold.

This paper presents a simplified cohort model (*Robust Ecosystem Demography (RED)*) which updates the number of trees in each mass class, but does not separately track tree-age or patch-age. RED assumes that the tree size-distribution of a forest is determined by how the rates of tree growth and mortality vary with tree size (Kohyama et al., 2003; Coomes et al., 2003; Muller-Landau et al., 2006; Lima et al., 2016). We follow many other studies in assuming that tree-growth rates vary with the three-quarter power of tree mass ($m^{3/4}$), as suggested by metabolic scaling theory (West et al., 1997). Where tree mortality

rate can also be assumed to be approximately independent of tree mass, the demographic equation yields equilibrium tree-size distributions which follow a Weibull distribution. This is sometimes termed *Demographic Equilibrium Theory (DET)* (see Appendix B). These simplifications significantly reduce the number of free parameters in RED, but still enable it to fit forest inventory data in North America (Moore et al., 2018) and South America (Moore et al., 2020).

## 2 Description of the Model

A full list of variables, parameters and units are given in Table 1.

**Table 1.** Model variables, parameters and units

| Symbol | Definitions | Units |
|---|---|---|
| | **Dimensions** | |
| $t$ | Time | year |
| $m$ | Carbon mass of an individual within a PFT | kgC |
| | **ESM Inputs** | |
| $P$ | Total assimilate of Net Primary Productivity minus local (leaves, wood and roots) litterfall | $\text{kgC m}^{-2}\,\text{yr}^{-1}$ |
| $\gamma_{\text{d}}$ | Disturbance mortality rate, the fraction of population dying over a year due to explicitly modelled reasons | $\text{yr}^{-1}$ |
| | **Individual** | |
| $m_0$ | Lowest/sapling mass boundary | kgC |
| $g$ | Structural growth of an individual at a given mass and time | $\text{kgC yr}^{-1}$ |
| $g_0$ | Structural growth of an individual at the lowest mass boundary at a specific time | $\text{kgC yr}^{-1}$ |
| $a$ | Crown area of an individual at a given mass | $\text{m}^2$ |
| $a_0$ | Crown area of an individual at the lowest mass boundary | $\text{m}^2$ |
| $\phi_g$ | Constant describing the power law scaling of structural growth across mass | $-$ |
| $\phi_a$ | Constant describing the power law scaling of crown area across mass | $-$ |
| $\alpha$ | The fraction of total growth going into seedling recruitment | $-$ |
| | **Cohort** | |
| $n$ | Number density across mass space, the derivative of $N$ with respect to mass | $(\text{kgC})^{-1}\text{m}^{-2}$ |
| $N$ | Number density | $\text{m}^{-2}$ |
| $G$ | Growth density | $\text{kgC m}^{-2}\,\text{yr}^{-1}$ |
| $\nu$ | The fractional coverage | $-$ |
| $\gamma$ | Mortality rate, the summation of the baseline and additional mortalities across mass | $\text{yr}^{-1}$ |
| $\gamma_{\text{b}}$ | Baseline mortality rate, the fraction of population dying over a year due to non-explicitly modelled reasons | $\text{yr}^{-1}$ |
| $s$ | The fraction of space available for seedlings | $-$ |
| $F$ | The flux of population density over time | $\text{m}^{-2}\text{yr}^{-1}$ |
| $\Lambda_{\text{d}}$ | Demographic litter, the loss of carbon due to competition and mortality | $\text{kgC m}^{-2}\,\text{yr}^{-1}$ |
| $M$ | Biomass density | $\text{kgC m}^{-2}$ |
| $c_{k,l}$ | Competition coefficient, the fraction a PFT, $k$, that is shaded by the canopy of PFT $l$ | $-$ |

| Symbol | Definitions | Units |
|--------|-------------|-------|
| **Equilibrium** | | |
| $\mu_0$ | The boundary turnover parameter - the ratio of mass lost to gained due to growth in the boundary mass class | – |
| $\lambda_i$ | The proportional population of the $i^{\text{th}}$ class to the $i^{\text{th}} - 1$ class at equilibrium | – |
| eq | Subscript denoting a variable in equilibrium | – |
| **Numerical** | | |
| $k, l$ | Indices representing the PFT number | – |
| $i, j$ | Indices representing mass class number | – |
| $I$ | The largest mass class | – |
| $(k)$ | The current time-step | – |
| $\xi$ | The size scaling coefficient, where mass classes are defined as $m_j = \xi \, m_{j-1}$, with $\xi > 1$ | – |

## 2.1 Theory

The underlying theoretical model for RED is a continuity equation, for each PFT and spatial location, which describes the time-evolution of the number density $n$ of plants per unit area per unit mass $m$:

$$\frac{\partial n}{\partial t} + \frac{\partial}{\partial m} n \, g = -\gamma n \tag{1}$$

Here $g$ is the growth rate and $\gamma$ is the mortality rate of a plant of mass $m$. In general, $g$ and $\gamma$ could be any reasonable function of tree size. For large-scale applications we make simplifying assumptions for these functions consistent with observed $n$ from forest inventory data (Moore et al., 2018, 2020). By default we assume that $\gamma$ is independent of plant mass, and that $g$ follows a power-law of plant mass:

$$g = g_0 \left( \frac{m}{m_0} \right)^{\phi_g} \tag{2}$$

Here $g_0$ is the growth rate of a plant with the reference mass, $m_0$. A value of $\phi_g = 0.75$ is assumed by default, consistent with the analysis of field-based measurements by Niklas and Spatz (2004). We also follow Niklas and Spatz (2004) in assuming the scaling of plant canopy area $a$ with plant mass:

$$a = a_0 \left( \frac{m}{m_0} \right)^{\phi_a} \tag{3}$$

where $\phi_a = 0.5$ by default. Solutions for $n$ can be integrated over mass to derive the total plant number, $N = \int_0^\infty n \, , dm$, the
total growth rate, $G = \int_0^\infty g \, n \, dm$, the total biomass, $M = \int_0^\infty m \, n \, dm$, and the fractional area covered $\nu = \int_0^\infty a \, n \, dm$.

## 2.2 Discrete Mass Classes

We wish to produce a model of vegetation demography that can be updated numerically and which explicitly conserves vegetation carbon, providing a constraint on the number of plants moving between mass classes in the discrete form. In order to do

this we integrate Eq. (1) over finite mass ranges:

$$\frac{\partial N_i}{\partial t} + F_i - F_{i-1} = -\gamma N_i \tag{4}$$

where $i$ denotes the $i^{th}$ mass class; $F_i$ is the flux of plants growing out of the $i^{th}$ mass class and into the $(i+1)^{th}$ mass class; $F_{i-1}$ is the flux of plants growing out of the $(i-1)^{th}$ mass class and into the $i^{th}$ mass class; and $N_i$ is the number of plants per unit area in the $i^{th}$ mass class. For clarity, Eq. (4) is deliberately presented as continuous in time at this stage, as the focus in this subsection is on discretization of the mass profile. The fully numerical version of RED, which includes discretization of time, is described in Section 2.4 and 2.5. In order to explicitly conserve carbon, the flux $F_i$ must take the form (see Appendix A) :

$$F_i = \frac{N_i\, g_i}{(m_{i+1} - m_i)} \tag{5}$$

where $m_i$ is the mean mass of a plant in the $i^{th}$ mass class, and $g_i$ is the growth rate per plant of the $i^{th}$ mass class [kgC yr$^{-1}$ plant$^{-1}$].

## 2.3   Seedling production and gap competition

To solve Eq. (4) we also require a lower boundary condition which represents the rate at which seedlings of mass $m_0$ are introduced into the cohort. Here we assume that a fixed fraction, $\alpha$, of the total assimilate available to a PFT ($P$), is devoted to producing new seedlings, with the remainder $G = (1-\alpha)P$ being allocated to the growth of existing plants. Spreading is homogeneous across the entirety of the grid-box, but only seedlings established within 'unoccupied' space will survive to join the plant cohort. The net incoming flux of seedlings of mass $m_0$ is therefore:

$$F_0 = \frac{\alpha P}{m_0} s = \frac{\alpha}{(1-\alpha)} \frac{G}{m_0} s \tag{6}$$

where $s$ is the fractional gap area available for seedlings. The definition of $s$ is assumed to differ by PFT to reflect an underlying tree-shrub-grass dominance hierarchy, as shown schematically in Figure 1. Therefore, the rate of recruitment $F_0$ is the ratio of a fraction of the carbon assimilate allocated to reproduction, $\alpha P$, and $m_0$, multiplied by the gap area $s$.

The space available to the seedlings of the $k^{th}$ PFT is calculated from the area fractions of the PFTs to which it is subdominant:

$$s_k = 1 - \sum_l c_{kl}\, \nu_l \tag{7}$$

where $\nu_l$ is the area fraction of the $l^{th}$ PFT, and $c_{kl}$ is the competition coefficient for the impact of PFT $l$ on PFT $k$. If PFT $l$ is within the same plant functional group (trees, shrubs or grasses) as PFT $k$, or dominant over it, $c_{kl} = 1$. If PFT $k$ is dominant over PFT $l$, $c_{kl} = 0$ (Figure 1). This 'gap' boundary condition results in there being no equilibrium solution where the amount of coverage exceeds 1. Doing so would halt the recruitment flux such that mortality processes would bring the fractional coverage back below unity. This is a similar competition regime to the Lotka-inspired TRIFFID model (Cox, 2001), and allows

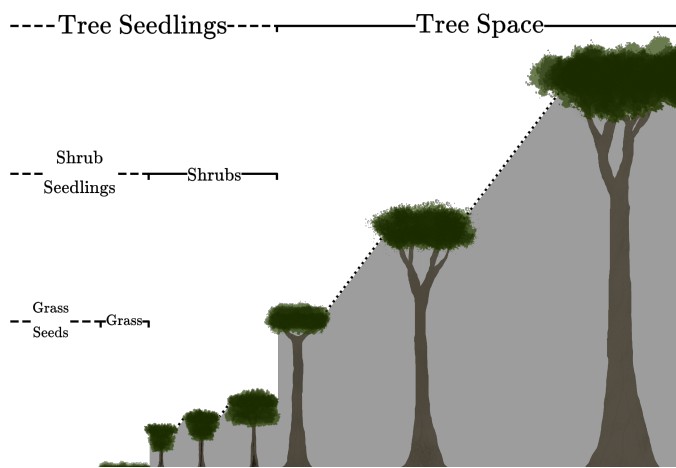

**Figure 1.** Schematic depicting the hierarchical PFT functional group regime within RED. Trees shade trees, shrubs and grasses. Shrubs shade shrubs and grasses, while grasses only shade grasses.

for the co-existence between inter-functional groups (trees, shrubs and grasses) of PFTs. For instance, a PFT such as Broadleaf Deciduous Tree can co-exist with a Deciduous Shrub and C3 Grass. The hierarchy also enables the simulation of succession during regrowth. Faster growing species of grasses will not be able to expand into space occupied by trees and shrubs, unless there is space created by disturbance. A summary of the competition coefficients is given in table 2.

**Table 2.** Competition coefficients assumed for different plant functional groups. A more detailed example of this is given for specific PFTs in table 3.

| | $c_{kl}$ | Trees | $l$ Shrubs | Grasses |
|---|---|---|---|---|
| | Trees | 1 | 0 | 0 |
| $k$ | Shrubs | 1 | 1 | 0 |
| | Grasses | 1 | 1 | 1 |

## 2.4 Coupling to Earth System Models

RED updates plant size distributions, biomass, and fractional areal coverage for an arbitrary number of PFTs at each spatial location, and can be driven by variables provided by a land carbon cycle model, an Earth System Model, or observations (see Figure 2). For each PFT, the minimum required input is a time-series of net carbon assimilate ($P$), defined as the difference

between Net Primary Productivity ($\Pi_{\mathrm{N}}$), and local litter production due to turnover of leaves, stems and roots ($\Lambda_{\mathrm{l}}$):

$$P = \Pi_{\mathrm{N}} - \Lambda_{\mathrm{l}} \tag{8}$$

We apply the $m^{3/4}$ scaling to $P$. We therefore implicitly assume the same scaling for both GPP and plant respiration. This is consistent with observations suggesting that plant production also scales approximately as $m^{3/4}$ (Enquist et al., 1998; Niklas and Enquist, 2001). Where available, additional mortality due to disturbance events such as droughts, fires and anthropogenic deforestation ($\gamma_{\mathrm{d}}$) can be added to the baseline mortality rates ($\gamma_{\mathrm{b}}$), for each PFT:

$$\gamma = \gamma_{\mathrm{b}} + \gamma_{\mathrm{d}} \tag{9}$$

Disturbance rates $\gamma_{\mathrm{d}}$ can in principle be both PFT-dependent and mass-dependent (e.g. to capture forestry practices).

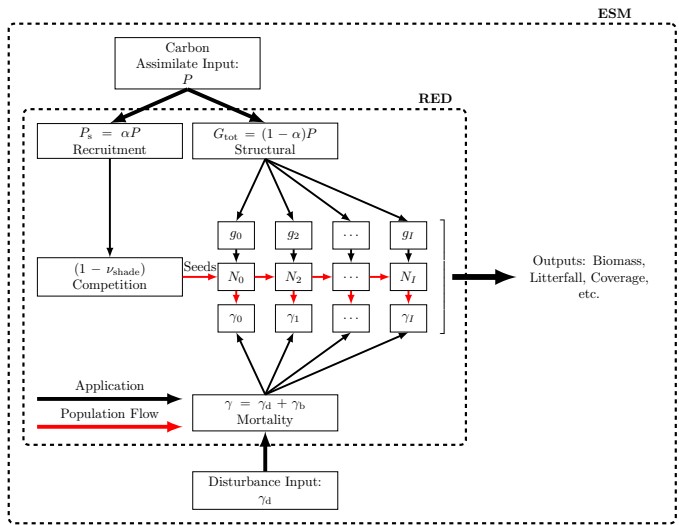

**Figure 2.** Schematic of RED coupled to an ESM or land carbon cycle model. RED is driven by a time-series of net carbon assimilate, $P$, which is then split between seedling production, $\alpha P$, and the growth of existing plants, $G = (1 - \alpha) P$. The seedling flux is limited by the available free space, $s$. Additional mortality rates diagnosed from disturbance models, $\gamma_{\mathrm{d}}$, can be added on to an assumed baseline mortality, $\gamma_{\mathrm{b}}$, as a function of both PFT and mass class.

The input values of net assimilate for each PFT ($P$), define the total structural growth rate, $G = (1 - \alpha) P$, and the seedling flux $F_0$ (via Eq. (6)), using PFT-specific values of the parameter $\alpha$ (see table 3). The definition of the total structural growth rate at a given time-step is:

$$G = \sum_{i} N_i g_i \tag{10}$$

can be combined with the growth-scaling given by Eq. (2), to derive the reference growth rate, $g_0$, from the net assimilate, $P$, which is a driving input:

$$g_0 = \frac{(1-\alpha)\,P}{\sum_i N_i \left(\frac{m_i}{m_0}\right)^{\phi_g}} \tag{11}$$

This in turn enables the growth rate of each mass class to be calculated using Eq. (2). For each PFT, the number of plants in mass class ($N_i$) is updated using a discretised form of Eq. (4):

$$N_i^{(j+1)} = N_i^{(j)} + \Delta t \left( F_{i-1}^{(j)} - F_i^{(j)} - \gamma^{(j)} N_i^{(j)} \right) \tag{12}$$

where $\Delta t$ is the RED time-step (typically 1 month), and the superscript $^{(j)}$ denotes the $j^{th}$ time-step. Our results are robust to changes in model timestep so long as the timestep remains small compared to the characteristic timescales associated with regrowth ($m_0/g_0 \sim 4$ years) and plant mortality ($1/\gamma \sim 20$ years). The lower boundary seedling flux is calculated from Eq. (6) using Eq. (7). We impose a zero-flux condition out of the upper mass class, under the assumption that there will be enough mass classes to ensure that this flux is negligible. However, to ensure carbon conservation on the land we add any plants that grow out of the upper mass class into a demographic litterfall term for each PFT, which is a RED output. This demographic litterfall term, $\Lambda_d$, keeps track of the carbon lost from the vegetation due to competition, mortality and the carbon in any such plants that grow out of the largest resolved mass class (class $I$):

$$\Lambda_d = \alpha\,P\,(1-s) + \sum_i \gamma_i M_i + g_I N_I \tag{13}$$

The first term on the righthand-side of this equation represents carbon loss due to the shading of seedlings; the second term represents mortality of the resolved mass classes (which may include disturbance events); and the third term, which is normally very small, is the loss of vegetation carbon due to plants growing beyond the modelled mass classes. In order to initiate regrowth from bare soil, RED also assumes a minimum effective fractional area of each PFT. Where the net assimilate would be sufficiently negative to take the vegetation fraction below this minimum, the minimum value is maintained by subtraction from the demographic litter. The demographic litterfall term therefore represents the net addition litter production consistent with the prescribed net assimilate flux, the disturbance rate, and the change in vegetation carbon modelled by RED. When coupling to an ESM or land carbon model, the demographic litterfall term ($\Lambda_d$) should be added to the input local litterfall ($\Lambda_l$) (as used in Eq. (8)), to calculate the total litterfall flux into the soil/litter system.

## 2.5 Steady-State

The steady-state of the continuum model defined by Eq. (1) and Eq. (2) can be solved analytical for each PFT (Moore et al., 2018, 2020). The continuum analytical solutions for the equilibrium mass distribution ($n_{eq}(m)$), the total plant number ($N_{eq}$), biomass ($M_{eq}$), growth rate ($G_{eq}$) and fractional area ($\nu_{eq}$) are summarised in Appendix B. The shape of the mass distribution and each of these parameters depend on the ratio of plant mortality to growth, which we choose to define for the reference mass class $m_0$:

$$\mu_0 = \frac{\gamma\,m_0}{g_0} \tag{14}$$

In order to initialise the numerical RED model in a drift-free initial state, we also derive the steady-state of the discrete model (of equation (12)), which will differ slightly from the continuum model for a finite number of mass classes. The equilibrium solution of Eq. (12) is derived in Appendix B2, based on the balance between seedling recruitment and total cohort mortality that defines the equilibrium state. The discretised version of RED thus yields formulae for the coverage (equation (B.28)) and biomass densities (equation (B.30)) which depend on the lowest mass class through the value of $\mu_0$. Similarly, analytical expressions can be derived for total plant number and total growth rate of each PFT at equilibrium:

1. $N_{\mathrm{eq}}$, the total equilibrium stand density:

$$N_{\mathrm{eq}} = N_0 X_N \tag{15}$$

2. The total equilibrium structural growth, $G_{\mathrm{eq}}$:

$$G_{\mathrm{eq}} = \sum_{i=0}^{I} N_i g_i = N_0 g_0 X_G \tag{16}$$

3. The total equilibrium coverage, $\nu_{\mathrm{eq}}$:

$$\nu_{\mathrm{eq}} = \sum_{i=0}^{I} N_i a_i = N_0 a_0 X_\nu \tag{17}$$

4. The total equilibrium carbon mass:

$$M_{\mathrm{eq}} = \sum_{i=0}^{I} N_i m_i = N_0 m_0 X_M \tag{18}$$

Here $X_N$, $X_G$, $X_\nu$ and $X_M$, are functions of $\mu_0$ (see Appendix B2). This equilibrium state is derived by setting $N_i^{(j+1)} = N_i^{(j)}$ in equation (B.17), such that the flux entering into a mass class is equal to the flux leaving that class, due to growth out of the class, and the loss of plants due to mortality.

The equations above therefore define the equilibrium state of the discrete system for given values of $N_0$ and $\mu_0$. The value of $\mu_0$ can be estimated from forest demographic data where this is available (Moore et al., 2018, 2020). However, for global applications we rarely have more observations than the fractional coverage of each PFT. Starting from the derived forms for $N_{\mathrm{eq}}$ (equation (15)) and $G_{\mathrm{eq}}$ (equation (16)), and requiring that the recruitment flux ($\alpha/(1-\alpha)G_{\mathrm{eq}}s$) is equal to that of the total population dying ($\gamma N_{\mathrm{eq}}$), we can derive an equation for the total equilibrium coverage (full details in Appendix B2):

$$\nu_{\mathrm{eq},k} = 1 - \left(\frac{1-\alpha}{\alpha}\right)\mu_0 \frac{X_N}{X_G} - \sum_{l \neq k} c_{kl}\nu_l \tag{19}$$

As the lefthand-side of this equation depends only on prescribed constants and $\mu_0$, Eq. (19) can be inverted (by numerical iteration) to estimate $\mu_0$ for observed values of the PFT fractions ($\nu_k$, $\nu_l$) and an assumed value of $\alpha$ (see Table 3). Once the value of $\mu_0$ has been derived in this manner, it can be used to calculate $X_\nu$, and therefore $N_0$ by inversion of Eq. (B.28):

$$N_0 = \frac{\nu_{\mathrm{eq}}}{a_0 X_\nu} \tag{20}$$

Equations (19) and (20) therefore allow us to define an initial equilibrium state ($N_i$) which is consistent with observed area fractions of each PFT. Furthermore, when paired with an estimate of the net carbon assimilate (from a model or observations), the $\mu_0$ estimate can be converted into a map of the implied mortality ($\gamma$) by PFT. We demonstrate this capability globally in the next section.

## 3 Modelling Results

For these runs, the numerical RED model is set up to use the 9 PFTs which are currently used in JULES (Harper et al., 2018). This enables us to directly use driving data - time series of the rate of net assimilation ($P$) - from a previous UKESM model simulation that includes JULES (Sellar et al., 2019). RED is integrated forward using a one month time-step and successive mass classes that differ by a multiplicative constant $\xi$, so that $m_i = \xi m_{i-1}$. The value of $\xi$ was chosen to optimally fit the analytical equilibrium solutions assuming 10 mass classes for trees, 8 mass classes for shrubs and 1 mass class for grasses, assuming $\mu_0 = 0.25$ (see Appendix B3). Other PFT-specific parameters are assumed as summarised in Table 3.

**Table 3.** List of PFT names and assumed allometric scaling parameters ($m_0, a_0, h_0$), seedling fraction ($\alpha$) and competition coefficient ($c_{\mathrm{pft,j}}$). The growth allometry of trees and shrubs across size is assumed to follow Niklas and Spatz (2004) ($\phi_g = 0.75$, $\phi_a = 0.5$, $\phi_h = 0.25$). The competition coefficients given describe which PFT functional group shades the current PFT, if $c_{\mathrm{pft,j}} = 1$, the PFT is shaded, otherwise it is not (Table 2).

| Long name | Abbrev | Classes | Scaling ($\xi$) | $\alpha$ | $m_0(\mathrm{kgC})$ | $a_0(\mathrm{m}^2)$ | Tree | Shrub | Grass |
|---|---|---|---|---|---|---|---|---|---|
| Broadleaf Evergreen Tree Tropical | BET-Tr | 10 | 2.32 | 0.10 | 1.00 | 0.50 | 1 | 0 | 0 |
| Broadleaf Evergreen Tree Temperate | BET-Te | 10 | 2.32 | 0.10 | 1.00 | 0.50 | 1 | 0 | 0 |
| Broadleaf Deciduous Tree | BDT | 10 | 2.35 | 0.10 | 1.00 | 0.50 | 1 | 0 | 0 |
| Needleleaf Evergreen Tree | NET | 10 | 2.35 | 0.10 | 1.00 | 0.50 | 1 | 0 | 0 |
| Needleleaf Deciduous Tree | NDT | 10 | 2.32 | 0.10 | 1.00 | 0.50 | 1 | 0 | 0 |
| Cool Season Grasses | C3 | 1 | 1.50 | 0.60 | 0.10 | 0.25 | 1 | 1 | 1 |
| Tropical Grasses | C4 | 1 | 1.50 | 0.60 | 0.15 | 0.25 | 1 | 1 | 1 |
| Evergreen Shrub | ESh | 8 | 2.80 | 0.35 | 0.15 | 0.25 | 1 | 1 | 0 |
| Deciduous Shrub | DSh | 8 | 2.80 | 0.35 | 0.50 | 0.25 | 1 | 1 | 0 |

The $c_{\mathrm{pft,j}}$ header spans Tree, Shrub, Grass columns.

### 3.1 Global: Diagnosed Plant Mortality Rates

Here we use the analytical forms for the equilibrium state (Section 2.5) and observations of global vegetation cover, to diagnose the corresponding map of PFT-specific mortality rates. These mortality rates are therefore consistent with the current observed

vegetation state, and rates of net assimilation ($P$) provided from UKESM (Sellar et al., 2019). The UKESM simulation provides NPP and local litterfall per unit area of each PFT. We multiply by PFT fraction to get the grid-box mean values required to drive RED (using ESA landcover data, as explained below). The observed maps of PFTs are provided by the ESA LC_CCI dataset for 2008-2012 (Poulter et al., 2015), projected onto the 9 JULES PFTs (Figure 3). Maps of the prescribed annual mean values of the rate of net assimilation ($P$) are shown in Figure 4.

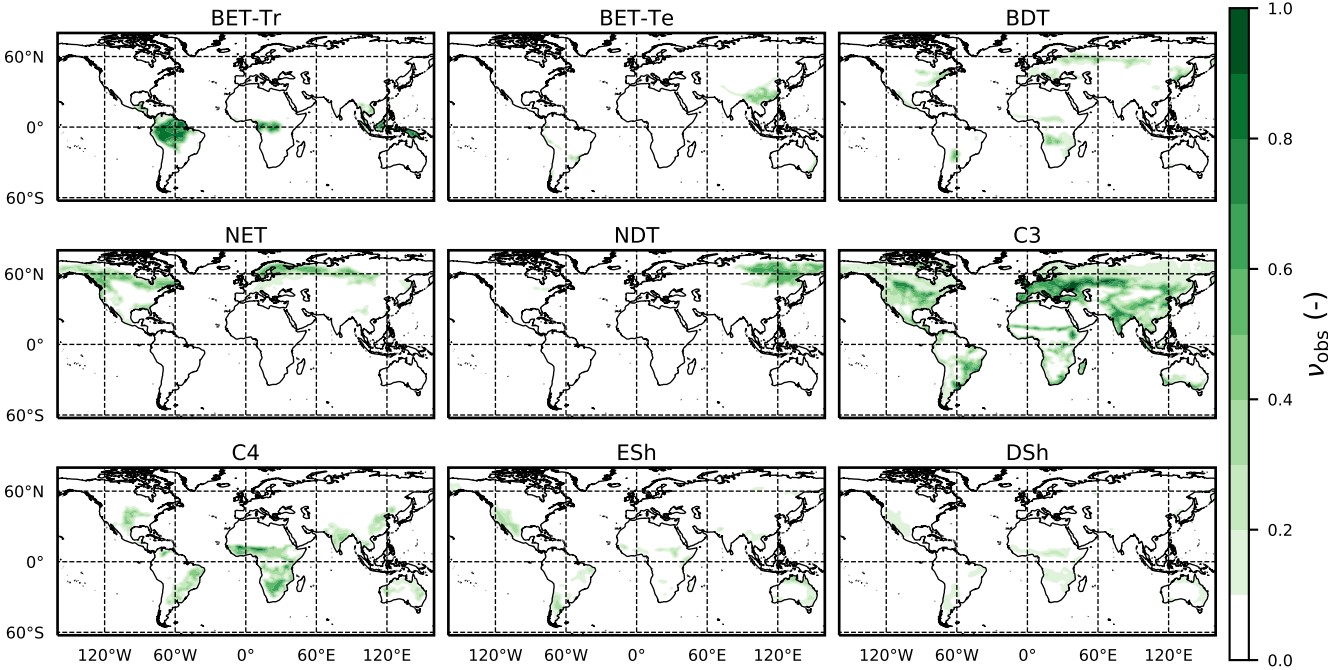

**Figure 3.** Observation-based dataset of the PFT area fractions for the nine JULES PFTs (Harper et al., 2016) as listed in Table 3.

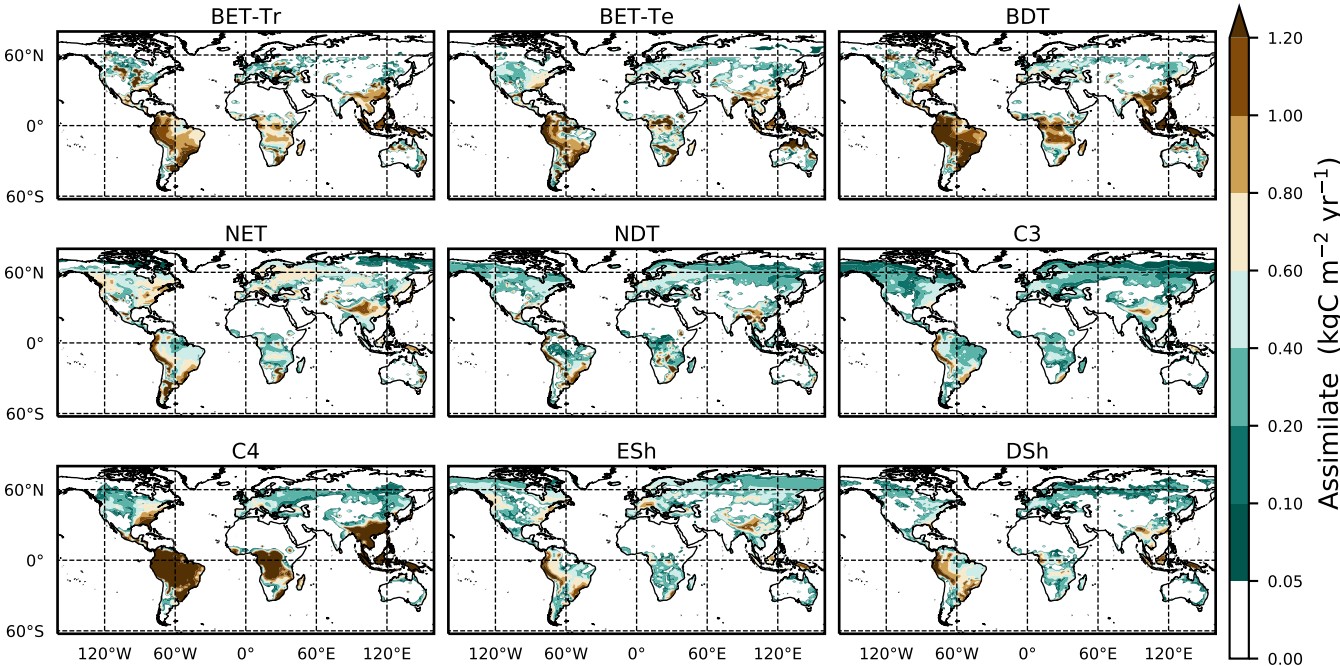

**Figure 4.** Mean net assimilate $P$ assimilate (equation (8)) from UKESM between 2000-2010. The mean is constructed by setting any negative growth rates to zero.

We use the procedure outlined in Section 2.5 to estimate spatially-varying values of $\mu_0$ for each PFT, using Eq. (B.32), and then Eq. (B.34) to estimate $N_0$. This method successfully reproduces the ESA map of dominant PFT to good accuracy, as shown in Figure 5 and Table 4.

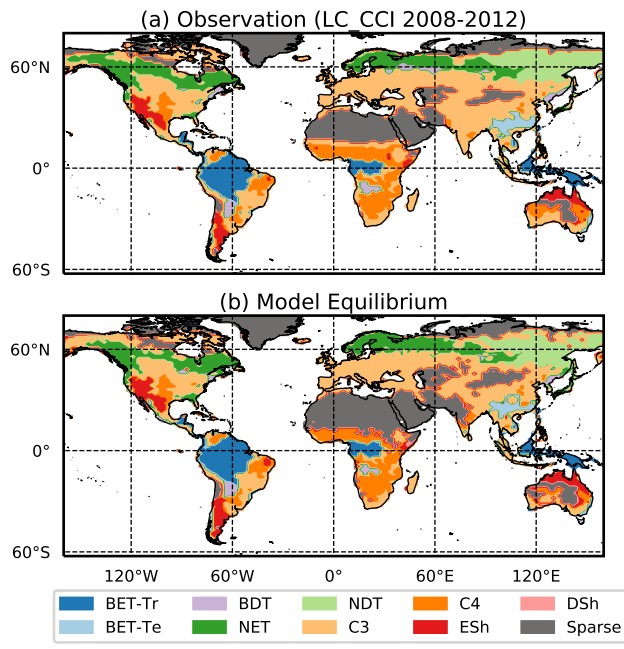

**Figure 5.** Maps of dominant PFT for (a) ESA LC_CCI dataset and (b) RED model equilibrium fractions. Sparse area is defined as where the total vegetation coverage is less than 10%.

**Table 4.** Goodness of fits for the RED equilibrium coverages to the coverages from ESA LC_CCI dataset across PFTs. $r$ represents the Pearson Correlation Coefficient, after weighting by the grid-box area to account for latitudinal variation of grid-box areas.

| PFT | $r$ | RMSE |
|---|---|---|
| BET-Tr | 0.990 | 0.030 |
| BET-Te | 0.935 | 0.030 |
| BDT | 0.783 | 0.053 |
| NET | 0.905 | 0.051 |
| NDT | 0.928 | 0.033 |
| C3 | 0.895 | 0.129 |
| C4 | 0.818 | 0.088 |
| ESh | 0.854 | 0.051 |
| DSh | 0.525 | 0.049 |

The fit of the RED equilibrium vegetation coverage to the ESA observations is generally very good (Table 4). However, it is imperfect in some areas (e.g. Central Asia, Sahel) where the driving net assimilate from UKESM is zero or negative. Also,

areas where the observational dataset indicates co-existing PFTs within the same vegetation class (e.g. broadleaf trees and needleleaf trees) are not well simulated by this first version of RED, which leads to competitive exclusion in the equilibrium state (see Discussion). Since we now have diagnosed values of $\mu_0$ and $N_0$, along with prescribed values of $P$, we can also diagnose the mean plant mortality rate $\gamma$, for each location and for each PFT, from Eq. (14) :

$$\gamma = \frac{\mu_0 \, g_0}{m_0} \tag{21}$$

where $g_0$ is given by Eq. (11) combined with Eq. (B.18) and Eq. (B.20). Maps of $\gamma$ values, derived in this way, are shown in Figure 6.

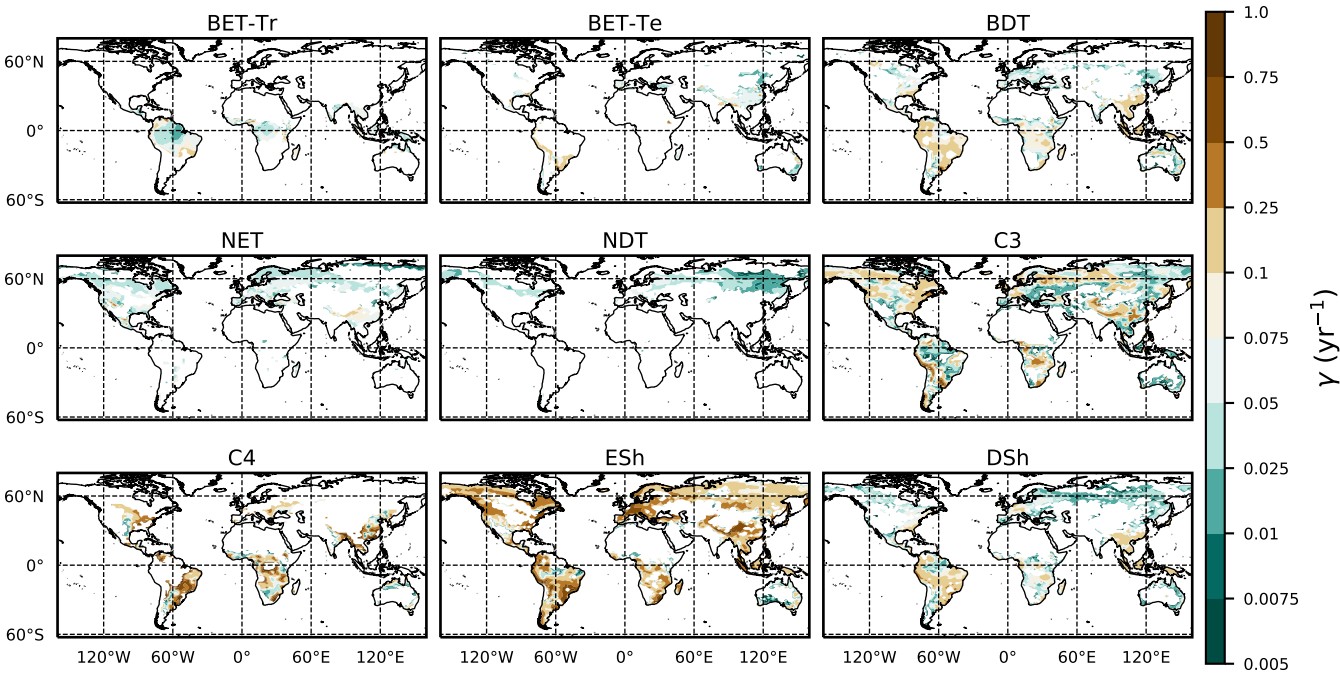

**Figure 6.** Diagnosed maps of mortality rates $\gamma$ for each PFT, as required for consistency with the ESA observations and the UKESM growth rates. White areas correspond with zero coverage and/or zero growth.

The mortality rate derived is dependent on the assumed areal coverage and the total assimilate. A high coverage with a low growth rate will result in a compensating low diagnosed mortality rate (and vice-versa). Furthermore, the choice of $\alpha$ (equation (11)) and $m_0$ also influence the diagnosed value of $\gamma$. An analysis of the sensitivity of the inferred value of $\gamma$ to these factors is presented in Appendix C. Assuming $\pm 20\%$ uncertainty on assimilate, $\alpha$, $m_0$ and $\pm 5\%$ on the coverage gives an uncertainty bound of $\pm 35\%$ on $\gamma$. Under the assumption that high coverages are indicative of the baseline mortality for a given PFT, we take a sub-sample of the grid-boxes that are within the top quartile of non-zero coverages ($\nu_{\mathrm{eq}} > 0.01$) (Table 5). The median $\mu_0$ value diagnosed from the top quartile of BET-Tr of $0.232^{+0.008}_{-0.007}$ (Table 5), is very close to the value calculated in our previous paper (Moore et al., 2020) of approximately $0.235$ for all of South America using the RAINFOR sites.

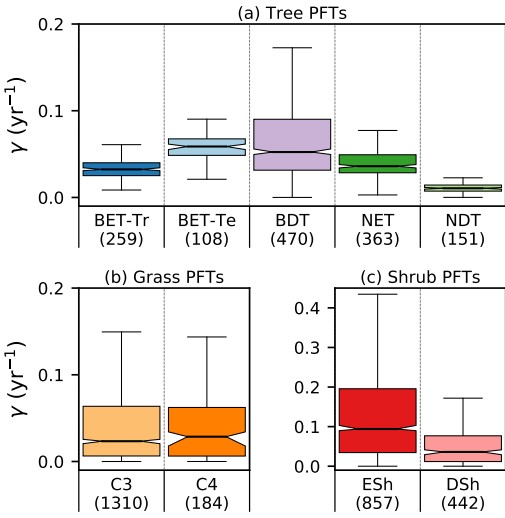

**Figure 7.** Diagnosed mortality rates for (a) trees, (b) grasses and (c) shrubs in the top quartile of coverage. Notches within the box represent the confidence bounds of the median. The confidence bounds are estimated using a bootstrap method. Bracketed numbers represent the number of grid-points.

**Table 5.** The area-weighted median values of observed coverage and driving net assimilate against $\mu_0$ and $\gamma$ for the upper quartile of grid-boxes for each PFT.

| PFT | Area weighted median | | | |
|---|---|---|---|---|
| | $\nu_{\mathrm{obs}}$ | $P$ (kgC m$^{-2}$ yr$^{-1}$) | $\mu_0$ | $\gamma$ (yr$^{-1}$) |
| BET-Tr | $0.793^{+0.019}_{-0.023}$ | $0.731^{+0.054}_{-0.041}$ | $0.232^{+0.008}_{-0.007}$ | $0.032^{+0.002}_{-0.001}$ |
| BET-Te | $0.402^{+0.020}_{-0.030}$ | $0.349^{+0.022}_{-0.028}$ | $0.340^{+0.006}_{-0.004}$ | $0.059^{+0.003}_{-0.003}$ |
| BDT | $0.238^{+0.011}_{-0.011}$ | $0.143^{+0.018}_{-0.014}$ | $0.377^{+0.013}_{-0.011}$ | $0.052^{+0.003}_{-0.003}$ |
| NET | $0.471^{+0.009}_{-0.011}$ | $0.281^{+0.005}_{-0.013}$ | $0.328^{+0.008}_{-0.009}$ | $0.036^{+0.002}_{-0.002}$ |
| NDT | $0.597^{+0.010}_{-0.015}$ | $0.112^{+0.009}_{-0.008}$ | $0.298^{+0.008}_{-0.007}$ | $0.011^{+0.001}_{-0.001}$ |
| C3 | $0.566^{+0.011}_{-0.007}$ | $0.124^{+0.008}_{-0.006}$ | $0.163^{+0.017}_{-0.013}$ | $0.023^{+0.002}_{-0.003}$ |
| C4 | $0.545^{+0.043}_{-0.053}$ | $0.123^{+0.084}_{-0.040}$ | $0.189^{+0.044}_{-0.027}$ | $0.029^{+0.006}_{-0.010}$ |
| ESh | $0.142^{+0.009}_{-0.007}$ | $0.028^{+0.002}_{-0.001}$ | $0.744^{+0.019}_{-0.021}$ | $0.094^{+0.010}_{-0.004}$ |
| DSh | $0.116^{+0.010}_{-0.015}$ | $0.024^{+0.006}_{-0.004}$ | $0.713^{+0.046}_{-0.027}$ | $0.036^{+0.005}_{-0.007}$ |

Site-level assessments of the rates of stand mortality within pan-tropical forests conclude a range of background rates (Lugo and Scatena, 1996; Phillips, 1996; Phillips et al., 2004). Phillips (1996) estimates mortality rates collected across 40 pan-tropical sites for tree sizes greater than $10 - 25$ cm dbh. Later work by Phillips et al. (2004) used the demographic data from

the RAINFOR dataset of trees $\geq 10$cm dbh. Using these site assessments, we can make a comparison to BET-Tr equilibrium mortality rates by looking at the values of $\gamma$ in areas where we would expect to see old growth forests. We use the top $25\%$ of coverages of the BET-Tr PFT to represent plausible areas of undisturbed forest. Figure 7 shows that the diagnosed baseline mortality rates are in reasonable agreement with these observational estimates for Amazonia.

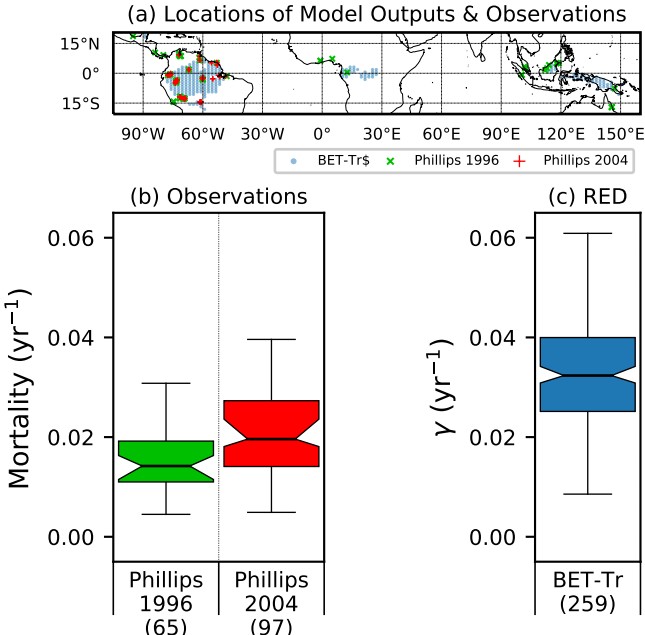

**Figure 8.** Comparison of observation-based estimates of tropical tree mortality (Phillips, 1996; Phillips et al., 2004) to $\gamma$ values diagnosed from RED for the BET-Tr PFT (for the top $25\%$ of fractions for this PFT). (a) location of observational sites (blue and green crosses) versus the chosen RED grid-points (red circles); (b) distribution of mortality across grid-boxes; (c) mortality distribution across the BET-Tr grid-points. Bracketed numbers in panel (b) represent the number of measurements, and in panel (c) the number of gridpoints.

There is a need to better understand the influence of mortality arising from disturbance events such as droughts and fire in order to constrain model projections (Pugh et al., 2020). Here we investigate if the equilibrium mortality rates implicitly capture areas of disturbances, by comparing the mean tree mortality rate to fire and land-use surveys (the mean mortality is defined here by weighting grid-box $\gamma$ values by grid-box fractional coverages). There are a number of surveys relating stand mortality in regions prone to wildfires (Swaine, 1992; Kinnaird and O'Brien, 1998; Peterson and Reich, 2001; Van Nieuwstadt and Sheil,
2005; Prior et al., 2009; Staver et al., 2009; Brando et al., 2014). In a broad sense, post-fire mortality rates can range from $0.06 \ \mathrm{yr}^{-1}$ to catastrophic rates around $0.8 \ \mathrm{yr}^{-1}$ and can vary quite considerably depending on tree species, fire frequency and drought severity. The drought-fire interaction is responsible for significantly increasing mortality post-fire and can be a driving cause of regional die-back (Allen et al., 2010; Brando et al., 2014). Using the ESA FIRE_CCI dataset (Chuvieco et al., 2019) we can estimate the burnt vegetation fraction per year. Taking the average burnt vegetation fraction for the months between
2000 and 2010, and converting into annual burn rate we gain an estimate of fire severity.

Another key issue is anthropogenic land-use and land-use change (Nepstad et al., 2008; Haddad et al., 2015). Fragmentation of natural forests is understood to raise the mortality of the remaining forest and to decrease the overall resilience of the ecosystem (Esseen, 1994; Laurance et al., 1998; Jönsson et al., 2007). In order to maintain a near-constant agricultural fraction, regular disruption such as grazing is needed to prevent re-colonisation and secondary succession (Dorrough and Moxham, 2005; Van Uytvanck et al., 2008; Chaturvedi et al., 2012). We carry out a comparison with land-use using the 2000 ESA LC_CCI inferred crop coverages (Li et al., 2019).

In Figure 9, we see the derived observations for burn area (a) and crop fraction (b), along with the derived mean $\gamma$ for the tree PFTs (c). From Figure 9 (d), we see that there are areas of large mortality ($\gamma > 0.075$ yr$^{-1}$) that do correspond to areas where we see large fire activity (burn rate $> 0.1$ yr$^{-1}$) and increased crop fraction ($> 0.25$). However, large burn rates are seen to overlap in parts of central Brazil around the Cernado region, Southern Africa and North Western Australia where fires are understood to play a significant part within the ecosystem (Coutinho, 1990; Medeiros and Miranda, 2008; Prior et al., 2009; Staver et al., 2009). There are also some areas of agriculture which correspond to deforestation, such as in the Atlantic forests of Brazil and in Indonesia (Higuchi et al., 2008; Curran et al., 2004). Areas of increased disturbances result in grasses and shrubs dominating (Figure 3).

Analysis of the RED equilibrium is an indirect approach to estimating tree mortality based on simple yet mechanistic principles of demography, and relying on few inputs (vegetation cover and assimilate). It is however conditional on the assumed estimates of vegetation coverage and net rates of assimilation.

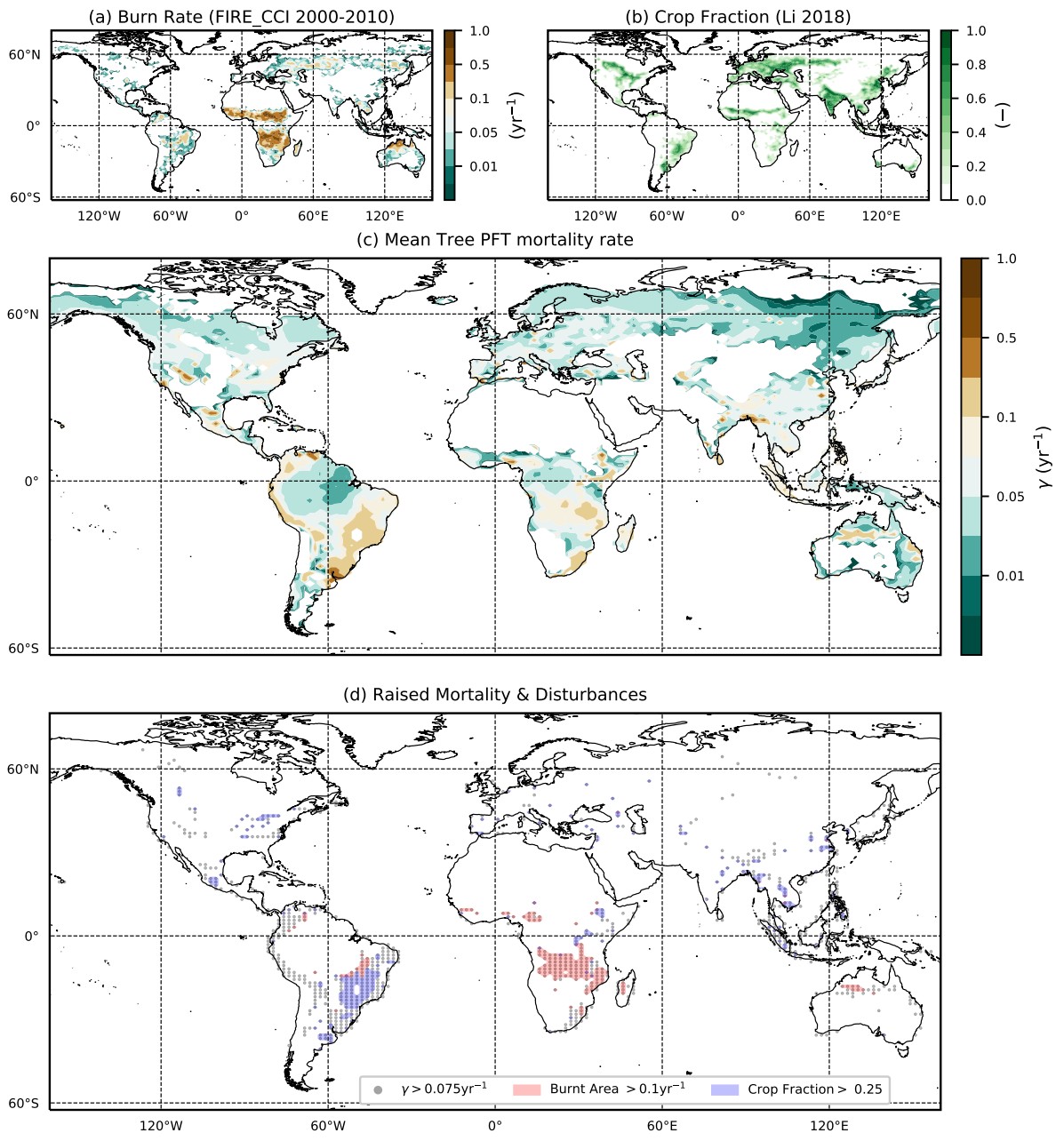

**Figure 9.** Comparison of diagnosed mortality rates, with observation-based maps of fire and land-use. (a) annual burnt area fraction from the ESA FIRE_CCI dataset; (b) crop fraction from the ESA LC_CCI 2000 dataset; (c) diagnosed mortality rate $\gamma$ for the tree PFTs (BET-Tr, BET-Te, BDT, NET, NDT); (d) overlap of areas of higher tree mortality rates ($\gamma > 0.075 \ \mathrm{yr}^{-1}$) with areas of fire (Burnt Area $> 0.1 \ \mathrm{yr}^{-1}$) and agriculture (Crop Fraction $\geq 25\%$).

## 3.2 Dynamical Simulations

### 3.2.1 Local: Simulating Succession

In this subsection we demonstrate the vegetation successional dynamics simulated by RED in an idealised spin-up from bare-soil, for a grid-box at the edge of the Amazonian rainforest (Figure 10). Under these circumstances, the diagnosed initial state is indeed the long-term equilibrium state, as evidenced by the horizontal dashed lines in panels a and b of Figure 10.

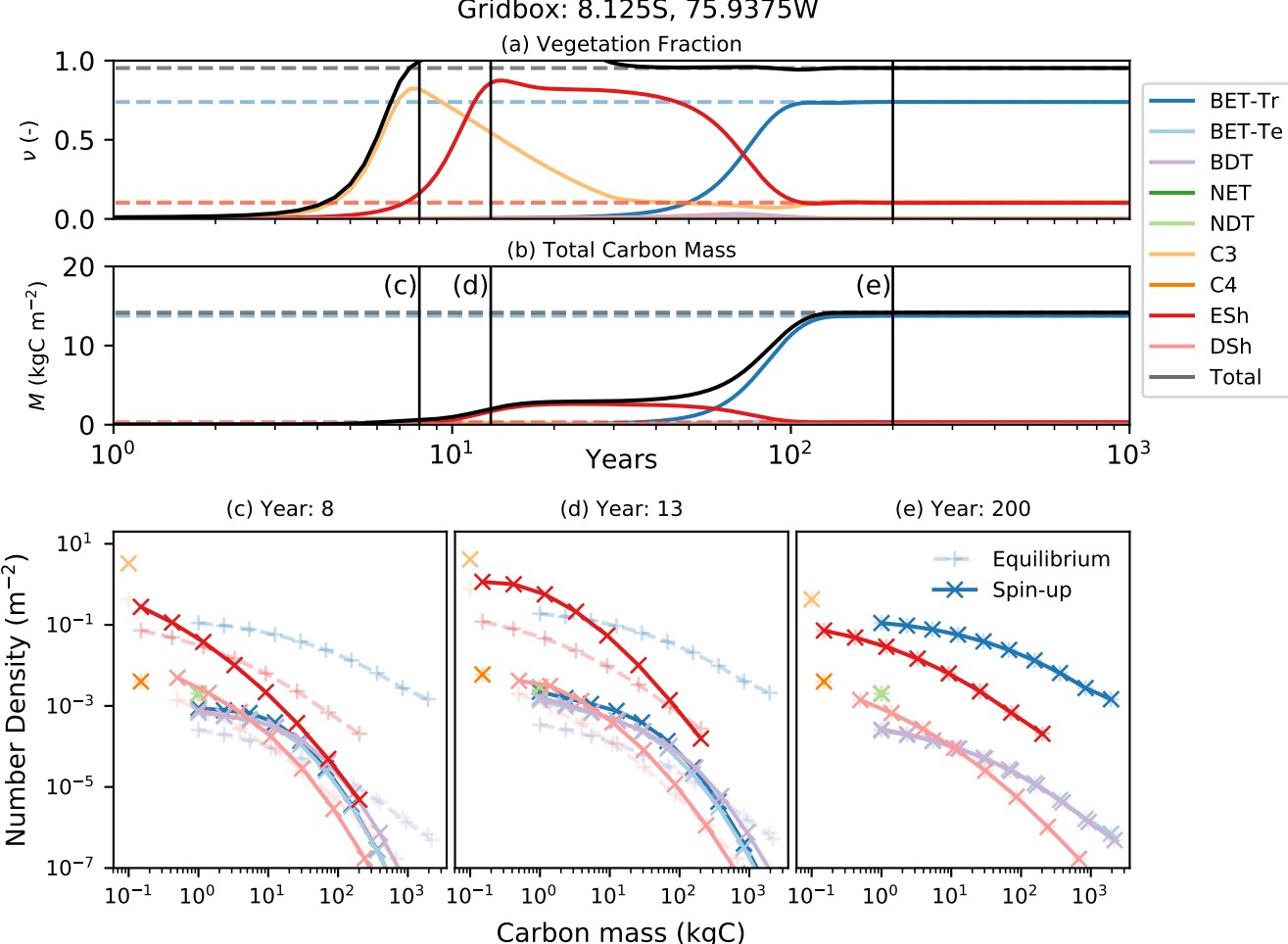

**Figure 10.** Dynamical runs of RED for a grid-box at the edge of the Amazonian rainforest, starting from bare soil (solid lines) and the diagnosed equilibrium state (dashed lines). (a) PFT fractions versus time; (b) biomass versus time; (c), (d) and (e) snapshots of the number density distribution of the PFTs across mass classes at different times. Lines marked as **+** are the equilibrium runs while **X** indicates the spin-up run. The ultimate steady-state is determined by the balance between recruitment and mortality (equation (6)). Intra- and inter-PFT occurs here through the shading of seedlings, which implies that just a fraction of the gridbox (s, 'space' or 'gap' fraction) is available to grow seedlings (equation (7)).

Faster growing grass PFTs dominate the grid-box within the first twelve years, before being replaced by evergreen shrubs which shade the grass seedlings. Eventually, Broad-leaf Evergreen Tropical Trees replace much of the shrub and grass, on a timescale determined in large part by the parameter $\alpha$ and the reference mass class $m_0$. With the parameters used here, the vegetation fraction reaches close to its equilibrium value after about 20 years (panel (a)), but full spin-up of the biomass takes around 150 years (panel (b)).

The modelled evolution of number density versus mass distribution for each PFT is shown in panel (c) (after 6 years), panel (d) (after 13 years) and panel (e) (after 100 years), with the eventual demographic equilibrium profiles shown by the dashed lines. It is clear that grass PFTs are close to their demographic equilibrium after only 6 years, but tree PFTs need more than 100 years to reach equilibrium.

The dashed lines in Figure 10 represent a dynamical RED simulation from the diagnosed demographic equilibrium state. This state is derived using the methodology described in Section 2.5, with one significant change. The competition rules given by Eq. (7) and Table 2 result ultimately in equilibria which have a single dominate PFT in each class of co-competing types (trees, shrubs, grasses). To avoid drifts associated with the competitive exclusion of the subdominant PFTs in each vegetation class, we choose to initialise the dominant PFT to have the total area fraction of all the PFTs in that vegetation class.

### 3.2.2   Global: Spin-up from Bare Soil

Transient simulations of global vegetation will be the subject of a future paper, but in the final subsection of this paper we wish to demonstrate the utility of the semi-analytical equilibrium for initialisation of global model runs. Figure 11 shows the time-evolution of global mean PFT fractions and biomass from a global run driven by net assimilation rates from the UKESM model. Once again, two RED simulations are shown, one started from bare soil (solid lines) and the other from the semi-analytical equilibrium state (dashed lines). Using a constant assimilate rate (Figure 4) and the mortality distribution (Figure 7), we see convergence of these two runs, but only after more than 1000 years of simulated time. The ability to diagnose the equilibrium state therefore has the potential to reduce model spin-up time hugely, especially for Earth System Models (ESMs) applications.

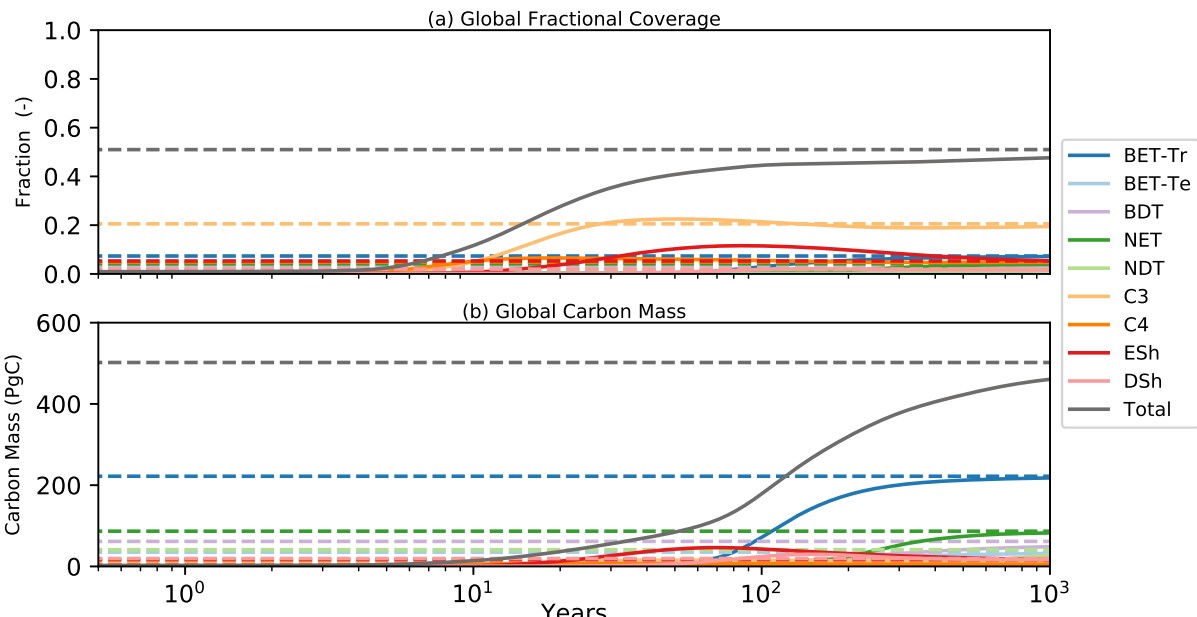

**Figure 11.** Global model spin-up from bare soil. As for figure 10, solid lines are spin-up from bare soil, dashed lines are the equilibrium instillation run. Panel (a) represents the fractional global coverage relative to the total land area; panel (b) represents the total biomass of the vegetation.

## 4 Discussion

The response of the land surface to climate change is a key uncertainty in climate projections. Ambitious climate targets also rely on land management practices such as reforestation and afforestation to increase the storage of carbon on land. First-generation Dynamic Global Vegetation Models (DGVMs) attempted to model the the land surface in terms of bulk properties
such as mean vegetation cover, vegetation carbon and leaf area index. These models lack information about the plant size-distribution, which compromised their ability to represent recovery from disturbance and the impact of land management. Providing useful guidance on these issues requires improved DGVMs which can represent changes in tree size distributions within forests (so called 'demography'). A number of much more sophisticated second-generation DGVMs are now under development. These models often explicitly simulate the number of plants within different size or mass classes, and on dif-
ferent patches of land, which are defined by the time since a disturbance event. Such second generation models are therefore in principle able to simulate variations in plant number density as both a function of patch age and plant size. However, this completeness is at the expense of much computational and parameter complexity.

Our previous work in evaluating demographic equilibrium theory for regional forest inventory datasets in North America (Moore et al., 2018) and using RAINFOR sites for South America (Moore et al., 2020), has provided the theoretical basis for
the development of RED. In those studies we found that the tree size-distributions observed at a large-scale in forests can be

satisfactorily understood in terms of demographic equilibrium in the size dimension alone. This is a reduction in complexity compared to other cohort models which are based on patch age, and yet an improvement in ecological fidelity compared to older phenomenological DGVMs such as TRIFFID (Cox, 2001). The modular design of RED allows for easy coupling to land-surface schemes, merely requiring the per unit grid-box total carbon assimilate rate and any additional mortality disturbance

rates as inputs for each grid-box (Figure 2). In principle, RED allows scope for more complex tree size-dependent processes, although in this first study we chose to assume size-independent (but spatially varying) mortality rates for each PFT. Our previous work suggests that this is a good first-order assumption (Moore et al., 2018, 2020).

Internally within the model we make a number of simplifications. Firstly, the number density for each PFT is treated as a function of plant mass alone. This immediately eliminates the need to explicitly represent patches, and therefore removes age

as an independent dimension. This is a distinct approach relative to cohort DGVMs which are based on patches defined by time since disturbance, such as the POP or ORCHIDEE-MICT models (Haverd et al., 2014; Yue et al., 2018). Secondly, we assume that plant growth rates vary as a power of plant mass. By default we assume a power of $\phi_g = 3/4$, which is consistent with Metabolic Scaling Theory (Enquist et al., 1998) and the empirically determined allometric relationships of Niklas and Spatz (2004).

Finally, we assume that competition is only significant for the lowest 'seedling' mass class. This enables us to represent gap dynamics among plants and resultant stages in succession. This represents a significant simplification compared to other approaches involving the Perfect Plasticity Assumption (PPA), as used within DGVMs such as LM3-PPA or CLM(ED) (Fisher et al., 2015; Weng et al., 2015), where canopies are assumed to perfectly fill gaps through photomorphism (Strigul et al., 2008). In LM3-PPA the radiative flux is limited by the available gap fraction in a given crown layer. PPA parallels our gap boundary

condition at the lowest mass class (equation (6)), but in RED the growth of a cohort is given by the disaggregation of total growth via metabolic scaling (equation (11)).

These simplifications allow RED to be solved analytically for the steady-state vegetation cover given information on the mortality and growth rates per unit area for each PFT. Such analytical steady-state solutions mean that RED can be easily initialised in drift-free pre-industrial states, which is vital to avoid spurious sources and sinks in climate-carbon cycle projections. The

analytical solutions also enable RED to be calibrated to the observed vegetation cover, via a single parameter ($\mu_0$) which represents the ratio of mortality to growth for a tree of an arbitrary reference mass. The existence of analytical steady-state solutions for RED also opens up other promising research avenues. For example, these solutions imply relationships between the fractional coverage of each PFT, total plant biomass, and the ratio of mortality-to-growth. This in turn allows RED to be calibrated using observations of any two of these quantities. The analytical solutions also allow optimality hypotheses to be

explored (e.g. the hypothesis that the fraction of net assimilate allocated to seed production maximises stand-density and/or biomass).

Aside from the existence of analytical steady-state solutions, RED is attractive for large-scale applications because it is both parameter sparse ('parsimonious') and requires very few driving variables. The main driving variable is the time-varying net plant growth rate for each PFT, which is defined as net primary production minus the local litterfall. These driving data can be

provided by a land-surface scheme, as we do in this study, or from observations. The only other driving variable for RED is

the mortality rate, which we treat in this study as a geographically-varying PFT-specific constant that is independent of mass. However, in principle RED could utilise mortality rates that depend on plant mass and time to represent individual disturbance events (e.g. forest fires, disease outbreaks). Despite its simplicity, the RED model is able to fit the global distribution of vegetation types (Figure 5), and simulate successional dynamics, including changes in forest demography (Figure 10).

There are inevitably weaknesses with any particular modelling approach. For RED, a current limitation is for competition to lead to a single PFT at each location within each co-competing vegetation class (i.e. tree, shrub, grass). The PFT with the highest equilibrium fraction will end up excluding sub-dominant PFTs within the same vegetation class. It was necessary for us to account for this eventual competitive exclusion to derive zero-drift steady-states for the global runs presented in Section 3.2.1. Such competitive exclusion is a common problem in DGVMs (Fisher et al., 2018). Currently, RED would therefore not

be the most appropriate DGVM to answer important questions regarding the role of biodiversity in ecosystem function (Pavlick et al., 2013; Levine et al., 2016). More sophisticated DGVMs are required to simulate plant diversity, such as individual-based models (Fischer et al., 2016), and DGVMs specifically-designed to capture sub-gridscale patch dynamics (Longo et al., 2019a, b). Adapting our 'gap' boundary condition (equation (7)) appears to be a promising way to allow greater PFT diversity in RED, without unduly increasing model complexity. We see this as a key priority for future research.

RED is currently being coupled to the JULES Land Surface Model, replacing TRIFFID as the default DGVM within that framework. In parallel, significant improvements are being made to the representation of physiological processes in JULES, most notably through the representation of non-structural carbohydrate ('SUGAR', Jones et al. (2019)), and through the inclusion of a coupled model of stomatal conductance and hydraulic failure under drought stress ('SOX', Eller et al. (2018, 2020)). Plans are also being made to derive the mortality rates for RED from the INFERNO forest-fire model (Burton et al., 2019).

These developments will allow us to simulate the effects of size-dependent tree mortality rates within the near future.

## 5  Conclusions

In this paper we have presented a new intermediate complexity second generation Dynamic Global Vegetation Model (DGVM), which captures important changes in forest demography. The *Robust Ecosystem Demography (RED)* model makes a number of important simplifications to achieve this. These simplifications are based on theoretical concepts (e.g. metabolic scaling theory

to estimate how plant growth rate varies with plant mass, and minimum crown overlap) and also comparison to observed forest demography (Moore et al., 2018, 2020). As a result, RED is parameter sparse, and can be driven with time-series of net plant growth rate (and optionally disturbance rates) for each Plant Functional Type (PFT). We have demonstrated that RED can be calibrated effectively to observed global vegetation maps, using a single fitting parameter (representing the ratio of mortality to growth for a plant of an arbitrary reference mass). The next stage will be to use RED in coupled climate-carbon

cycle projections so to assess how changes in vegetation demography impact future $CO_2$ and climate. We have made the prototype RED code publically available, and we hope that Earth System and land-surface modellers will make good use of this framework to further their own research.

*Code availability.* The RED model Python Code is archived at https://doi.org/10.5281/zenodo.3548678. Furthermore, RED is currently being coupled into JULES, where a basic integration currently exists as branch (vn5.4_veg3_ctrl) - this requires registration for the JULES repository (https://code.metoffice.gov.uk/trac).

## Appendix A: Functional Form of Flux $F_i$ in Discretised RED

5 For large-scale application in ESMs a primary concern is to ensure that the total vegetation carbon obeys carbon balance (i.e. only changes due to the net impact of total growth minus total mortality). Here we use that requirement to derive the functional form for $F_i$ as given in equation (5).

The total vegetation carbon in each mass class is $M_i = m_i N_i$. The update equation for $M_i$ is therefore Eq. (4) multiplied by $m_i$:

$$10 \quad \frac{\partial M_i}{\partial t} + m_i \left( F_i - F_{i-1} \right) = -\gamma M_i. \tag{A1}$$

The total carbon in the vegetation, $M$, is the sum of the carbon in each of the mass classes:

$$M = \sum_i M_i. \tag{A2}$$

Thus the update equation for the total carbon is:

$$\frac{\partial M}{\partial t} + \sum_i m_i \left( F_i - F_{i-1} \right) = -\gamma M, \tag{A3}$$

15 which can be rewritten as:

$$\frac{\partial M}{\partial t} + \sum_i F_i \left( m_i - m_{i+1} \right) = -\gamma M. \tag{A4}$$

Now substituting Eq. (5) into Eq. (A4) gives:

$$\frac{\partial M}{\partial t} = \sum_i N_i g_i - \gamma M. \tag{A5}$$

The first term on the righthand-side of this equation is the total carbon uptake due to growth, and the second term represents

20 the total carbon loss due to mortality, which is the required carbon conservation equation.

## Appendix B: Continuum Solutions and Demographic Equilibrium Theory

Equation (1), can be solved for the steady-state if we assume metabolic scaling of growth using Eq. (2) and a size-independent mortality (Moore et al., 2018, 2020):

$$n = n_0 \left( \frac{m}{m_0} \right)^{-\phi_g} \exp\left\{ \frac{\mu_0}{(1-\phi_g)} \left[ 1 - \left( \frac{m}{m_0} \right)^{1-\phi_g} \right] \right\}, \quad \mu_0 = \frac{\gamma m_0}{g_0}. \tag{B.1}$$

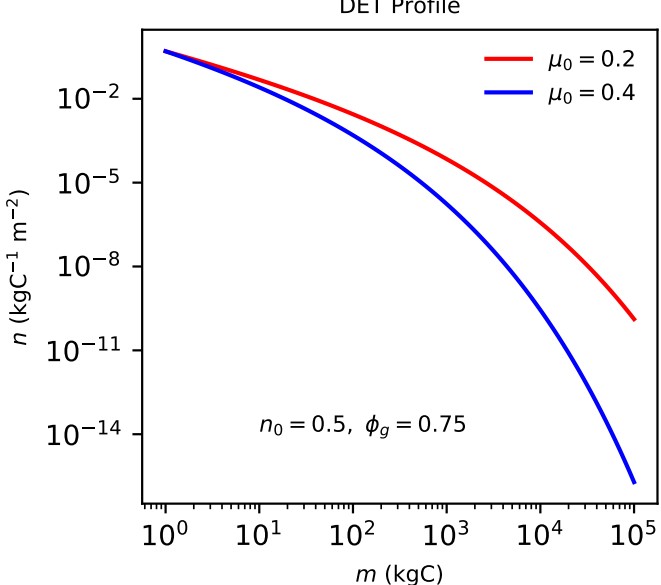

**Figure B1.** The quasi-Weibull number density solution to DET (equation (B.1)), assuming the same initial $n_0$ and growth scaling $\phi_g = 0.75$ but different $\mu_0$ values.

where $n_0$ is a boundary condition that describes the number density at the mass $m_0$. The parameter $\mu_0$ is the ratio of the rate biomass loss due to mortality to the rate of biomass gain due to growth, for the reference mass class $m_0$. Similar analytical solutions can be derived for other measures of tree-size, such as basal diameter or height (Moore et al., 2018, 2020).

Integrating Eq. (B.1) from $m_0$ to $\infty$ gives the total number density:

$$N_{\text{eq}} = \frac{n_0 g_0}{\gamma} = \frac{n_0 m_0}{\mu_0}. \tag{B.2}$$

Other cohort integrals can be derived by integrating over the number density distribution, such as total growth rate ($\int g\, n\, dm$):

$$G_{\text{eq}} = g_0 N_{\text{eq}} \left(\frac{\mu_0}{1 - \phi_g}\right)^{\frac{\phi_g}{\phi_g - 1}} \exp\left\{\frac{\mu_0}{1 - \phi_g}\right\} \Gamma\left(\frac{1}{1 - \phi_g}, \frac{\mu_0}{1 - \phi_g}\right) \tag{B.3}$$

total biomass ($\int m\, n\, dm$):

$$M_{\text{eq}} = m_0 N_{\text{eq}} \left(\frac{\mu_0}{1 - \phi_g}\right)^{\frac{1}{\phi_g - 1}} \exp\left\{\frac{\mu_0}{1 - \phi_g}\right\} \Gamma\left(\frac{1}{1 - \phi_g} + 1, \frac{\mu_0}{1 - \phi_g}\right) \tag{B.4}$$

and total vegetation cover ($\int a\, n\, dm$):

$$\nu_{\text{eq}} = a_0 N_{\text{eq}} \left(\frac{\mu_0}{1 - \phi_g}\right)^{\frac{\phi_a}{\phi_g - 1}} \exp\left\{\frac{\mu_0}{1 - \phi_g}\right\} \Gamma\left(\frac{\phi_a}{1 - \phi_g} + 1, \frac{\mu_0}{1 - \phi_g}\right) \tag{B.5}$$

where $\Gamma(a, b)$ is the incomplete upper gamma function.

As we assume the allometric exponents presented in Niklas and Spatz (2004) ($\phi_g = 3/4, \phi_a = 1/3$), these functional forms

simplify to:

$$G_{\text{eq}} = g_0 N_{\text{eq}} \left( 1 + \frac{3}{4\mu_0} + \frac{3}{8\mu_0^2} + \frac{3}{32\mu_0^3} \right) \tag{B.6}$$

$$M_{\text{eq}} = m_0 N_{\text{eq}} \left( 1 + \frac{1}{\mu_0} + \frac{3}{4\mu_0^2} + \frac{3}{8\mu_0^3} + \frac{3}{32\mu_0^4} \right) \tag{B.7}$$

$$\nu_{\text{eq}} = a_0 N_{\text{eq}} \left( 1 + \frac{1}{2\mu_0} + \frac{1}{8\mu_0^2} \right) \tag{B.8}$$

Finally, to convert a $\mu_0$ found using biomass ($\mu_{0,\text{tdm}}$) to one based on carbon mass, we use the formula:

$$\mu_0 = 2^{1-\phi_g} \mu_{0,\text{tdm}} \tag{B.9}$$

assuming that biomass is twice the carbon mass.

10 **B1  Closed Continuous Form**

The lowest population flux, $n_0 g_0$, is equal to the seedling boundary condition, $F_0$, in equation (6):

$$n_0 g_0 = \frac{\alpha}{1-\alpha} \frac{G}{m_0} s \tag{B.10}$$

Substituting the total number density, $N_{\text{eq}}$, equation (B.2), into the lefthand-side, and total growth, $G_{\text{eq}}$, Eq. (B.6), into the righthand-side, yields a solution for the equilibrium coverage, assuming $s = 1 - \nu_{\text{eq}}$:

$$15 \quad \gamma N_{\text{eq}} = \left( \frac{\alpha}{1-\alpha} \right) \frac{g_0}{m_0} N_{\text{eq}} (1 - \nu_{\text{eq}}) \left( 1 + \frac{3}{4\mu_0} + \frac{3}{8\mu_0^2} + \frac{3}{32\mu_0^3} \right) \tag{B.11}$$

which simplifies:

$$\nu_{\text{eq}} = 1 - \left( \frac{1-\alpha}{\alpha} \right) \frac{\mu_0}{1 + \frac{3}{4\mu_0} + \frac{3}{8\mu_0^2} + \frac{3}{32\mu_0^3}} \tag{B.12}$$

Using equation (B.8) we can write the total number density at equilibrium in terms of $\nu_{\text{eq}}$:

$$N_{\text{eq}} = \frac{\nu_{\text{eq}}}{a_0} \left( \frac{1}{1 + \frac{1}{2\mu_0} + \frac{1}{8\mu_0^2}} \right) \tag{B.13}$$

20  This enables equation (B.6) to be rewritten:

$$G_{\text{eq}} = \frac{\nu_{\text{eq}} g_0}{a_0} \left( \frac{1 + \frac{3}{4\mu_0} + \frac{3}{8\mu_0^2} + \frac{3}{32\mu_0^3}}{1 + \frac{1}{2\mu_0} + \frac{1}{8\mu_0^2}} \right) \tag{B.14}$$

This equation in turn defines the total assimilate:

$$P_{\text{eq}} = \left( \frac{1}{1-\alpha} \right) G_{\text{eq}} \tag{B.15}$$

Finally the total biomass can be written in closed form as:

$$25 \quad M_{\text{eq}} = \frac{\nu_{\text{eq}} m_0}{a_0} \left( \frac{1 + \frac{1}{\mu_0} + \frac{3}{4\mu_0^2} + \frac{3}{8\mu_0^3} + \frac{3}{32\mu_0^4}}{1 + \frac{1}{2\mu_0} + \frac{1}{8\mu_0^2}} \right) \tag{B.16}$$

## B2   Discrete Steady-State

To solve for the discrete model equilibrium, we start from the flow equation from Eq.(4) with the term $\partial N/\partial t \to 0$:

$$\gamma N_i + F_i = F_{i-1} \tag{B.17}$$

considering the population flux - equation (5), we find $N_i$ in relation to the lower mass class, $N_{i-1}$:

$$5 \quad N_i = N_{i-1}\left[\frac{g_{i-1}/(m_i - m_{i-1})}{g_i/(m_{i+1} - m_i) + \gamma}\right] = N_{i-1}\lambda_i \tag{B.18}$$

Assuming no population grows out of the top class, $\lambda_I$ is given as:

$$\lambda_I = \frac{g_{i-1}}{(m_i - m_{i-1})\gamma} \tag{B.19}$$

$\lambda_i$ can be simplified to depend only on $\mu_0$, by using $\mu_0 = (\gamma m_0/g_0)$ (equation (14)) and applying the mass scaling of growth rates $g_i = g_0(m_i/m_0)^{\phi}_g$. We can show that $\lambda_i$ and $\lambda_I$ are:

$$10 \quad \lambda_i = \frac{(m_{i-1}/m_0)^{\phi_g}\, m_0/(m_i - m_{i-1})}{(m_i/m_0)^{\phi_g}\, m_0/(m_{i+1} - m_i) + \mu_0}, \quad \lambda_I = \frac{(m_{i-1}/m_i)^{\phi_g}\, m_0}{(m_i - m_{i-1})\mu_0} \tag{B.20}$$

An expression for the total stand density at equilibrium, $N_{\text{eq}}$, can be derived. Using equation (B.18), we can represent any population of mass class $i$ in terms of the lowest mass class $N_0$:

$$N_i = N_0 \prod_{j=1}^{i} \lambda_j \tag{B.21}$$

15   Therefore, when finding the total number of stands relative to $N_0$ we get:

$$N_{\text{eq}} = N_0 \left[1 + \sum_{i=1}^{I}\prod_{j=1}^{i}\lambda_j\right] = N_0 X_N \tag{B.22}$$

where $X_N$ describes the sum of the all mass classes as a proportion of $N_0$. We can describe the total class growth rate in relation to $N_0$ as:

$$G_i = N_0 g_i \prod_{j=1}^{i} \lambda_i \tag{B.23}$$

20   By using the allometric relationship (equation (2)):

$$G_i = N_0 g_0 \left(\frac{m_i}{m_0}\right)^{\phi_g} \prod_{j=1}^{i} \lambda_j \tag{B.24}$$

we describe the total class growth rate in relation to the lowest class growth rate, $N_0 g_0$. Like $N_{\text{eq}}$, we can show the total growths across all classes is therefore:

$$G_{\text{eq}} = N_0 g_0 \left[1 + \sum_{i=1}^{I}\left(\frac{m_i}{m_0}\right)^{\phi_g}\prod_{j=1}^{i}\lambda_j\right] = N_0 g_0 X_G \tag{B.25}$$

We can repeat the same process for coverage:

$$\nu_i = N_0 a_i \prod_{j=1}^{i} \lambda_j \tag{B.26}$$

and using allometric relationship (equation (3)):

$$\nu_i = N_0 a_0 \left(\frac{m_i}{m_0}\right)^{\phi_a} \prod_{j=1}^{i} \lambda_j \tag{B.27}$$

This gives the total coverage, $\nu_{\mathrm{eq}}$ as:

$$\nu_{\mathrm{eq}} = N_0 a_0 \left[1 + \sum_{i=1}^{I} \left(\frac{m_i}{m_0}\right)^{\phi_a} \prod_{j=1}^{i} \lambda_j\right] = N_0 a_0 X_\nu \tag{B.28}$$

Finally, for the total carbon mass within the class:

$$M_i = N_0 m_i \prod_{j=1}^{i} \lambda_i \tag{B.29}$$

with the total carbon density equalling:

$$M_{\mathrm{eq}} = N_0 m_0 \left[1 + \sum_{i=1}^{I} \frac{m_i}{m_0} \prod_{j=1}^{i} \lambda_j\right] = N_0 m_0 X_M \tag{B.30}$$

In equilibrium, the rate of the recruitment of seedlings (equation (6)) must balance the rate of loss of plants due to total mortality ($\gamma N_{\mathrm{eq}}$):

$$\gamma N_{\mathrm{eq}} = \frac{\alpha}{(1-\alpha)} \frac{G_{\mathrm{eq}}}{m_0} s \tag{B.31}$$

Substituting in equation (B.22), Eq. (B.25) yields a balance equation for the $k^{th}$ PFT:

$$\left(\frac{\alpha}{1-\alpha}\right) \left(1 - \sum_l c_{kl} \nu_l\right) = \mu_0 \frac{X_N}{X_G} \tag{B.32}$$

We can get the equilibrium fraction of a PFT, $k$, by rearranging the above equation, assuming $c_{kk} = 1$:

$$\nu_{\mathrm{eq},k} = 1 - \left(\frac{1-\alpha}{\alpha}\right) \mu_0 \frac{X_N}{X_G} - \sum_{l \neq k} c_{kl} \nu_l \tag{B.33}$$

Once the value of $\mu_0$ has been derived in this manner, we can find $N_0$ by inversion of equation (B.28):

$$N_0 = \frac{\nu_{\mathrm{eq}}}{a_0 X_\nu} \tag{B.34}$$

Substituting equation (B.33) into Eq.(B.34) allows us to determine $N_0$ and hence most other total densities in terms of purely $\mu_0$ and prescribed constants.

## B3 Continuous-Discrete Convergence

Inevitably discretised models will not exactly reproduce exact continuum analytical solutions, as a result of numerical inaccuracies that arise from using a finite number of mass classes. However, where exact analytical solutions exist they can be used to benchmark numerical models and optimise discretisation schemes, which is what we set out to do in this appendix. We compare the continuum analytical solution for the equilibrium coverage (equation B.12) to results from RED with differing numbers of mass classes $m_i$ and a geometric mass class scaling, $m_{i+1} = \xi m_i$. Figure B2(a) shows how the relationship between $\nu_{eq}$ varies with $\mu_0$ for the exact continuum solution (black line) and variants of the numerical version of RED with different numbers of mass classes (coloured lines). As hoped, results from the discretised model converge on the exact solution as the number of mass classes increases.

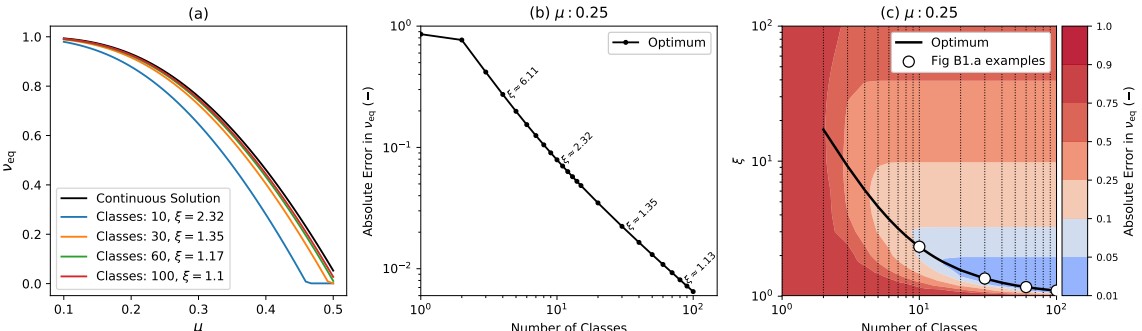

**Figure B2.** Comparison of the discretised model to the continuum analytical solution, showing convergence for higher numbers of mass classes. This example uses parameters for Broadleaf Evergreen Tropical trees (BET-Tr PFT) with $\alpha = 0.1$: (a) equilibrium coverage $\nu_{eq}$ versus $\mu_0$ for the exact continuum solution (black line) and discretisations of the mass dimension with varying numbers of mass classes and mass class width scaling ($\xi$); (b) absolute error in the modelled value of $\nu_{eq}$ against the number of mass classes using the optimum value of $\xi$ for each case; (c) optimum $\xi$ versus number of mass classes, with contours showing the absolute error in $\nu_{eq}$. Panels (b) and (c) assume $\mu_0 = 0.25$. The white dots in (c) have the same number of classes and scaling as the discrete lines in (a).

The numerical versions of RED shown in Figure B2(a) each use a value of $\xi$ that is near optimum for the number of mass classes, as shown in panels (b) and (c) of Figure B2. Optimum $\xi$ values reduce from about 2.3 for 10 mass classes to 1.1 for 100 mass classes. This variation results from a trade-off. For a given number of mass classes, small values of $\xi$ give greater numerical accuracy, but explicitly model less of the mass range, and the opposite is true of large $\xi$ values. As a result, optimum values of $\xi$ an be defined for each number of mass classes as outlined below.

For geometric scaling any mass can be expressed in terms of $m_0$, by writing $m_i = m_0(\xi)^i$. Therefore, by using $m_{i+1} - m_i = m_0(\xi)^i(\xi - 1)$, we find that our equilibrium form of $\lambda_i$ is reduced to:

$$\lambda_i = \frac{\xi^{(\phi_g - 1)(i-1)}}{\xi^{i(\phi_g - 1)} + \mu_0(\xi - 1)}, \quad \lambda_I = \frac{\xi^{(\phi_g - 1)(i-1)}}{\mu_0(\xi - 1)} \tag{B.35}$$

From figure B2 (c), we see that there is an optimum value for $\xi$, the geometric scaling for a given number of classes, which minimises the difference between the continuous and discrete forms. This can be found by taking the difference of the continuous and discrete coverages and differentiating with respect to $\xi$ to find the minima. It should be noted that as the continuous form is not dependent on $\xi$, we get:

$$\frac{\partial}{\partial \xi}\left[\nu_{\text{eq,continuous}} - \nu_{\text{eq}}\right] = -\frac{\partial}{\partial \xi}\left[\nu_{\text{eq}}\right] \tag{B.36}$$

where $\nu_{\text{eq}}$ corresponds with the discrete equilibrium (equation (B.32), with $\nu_{\text{eq}} = (1-s)$). Setting Eq. (B.36) equal to zero we reduce the relationship to only a dependence on $X_N$ and $X_G$:

$$0 = \frac{\partial}{\partial \xi}\left[\frac{X_N}{X_G}\right] = X_G X_N' - X_G' X_N \tag{B.37}$$

Finding the partial derivative of $X_N$, using the geometric form of equation (B.18), we get:

$$X_N' = \sum_{j=1}^{I}\left[\left(\prod_{i=1}^{j}\lambda_i\right)\left(\sum_{i=1}^{j}\frac{\lambda_i'}{\lambda_i}\right)\right] \tag{B.38}$$

and for $X_G$:

$$X_G' = \sum_{j=1}^{I}\left[\xi^{j\phi_g}\left(\prod_{i=1}^{j}\lambda_i\right)\left(j\phi_g\xi^{-1} + \sum_{i=1}^{j}\frac{\lambda_i'}{\lambda_i}\right)\right] \tag{B.39}$$

Finding $\lambda_i'$ we get:

$$\lambda_i' = \lambda_i\left[(1-i)(\phi_g-1)\xi^{-1} - \lambda_i\left(i(\phi_g-1)\xi^{\phi_g-2} + \mu_0\xi^{(i-1)(1-\phi_g)}\right)\right] \tag{B.40}$$

and for the top class, $\lambda_I'$:

$$\lambda_I' = \left(\frac{(1-\xi^{-1})(I-1)(\phi_g-1)-1}{\xi-1}\right)\lambda_I \tag{B.41}$$

To numerically solve for the minimum, we must differentiate Eq. (B.37), with respect to $\xi$. Through the product rule we get:

$$\frac{\partial^2}{\partial \xi^2}\left[\frac{X_N}{X_G}\right] = X_G X_N'' - X_G'' X_N \tag{B.42}$$

Differentiating equation (B.38) and simplifying gives:

$$X_N'' = \sum_{j=1}^{I}\left[\left(\prod_{i=1}^{j}\lambda_i\right)\left(\sum_{i=1}^{j}\frac{\lambda_i''}{\lambda_i}\right)\right] \tag{B.43}$$

and doing the same for Eq. (B.39) gives:

$$X_G'' = \sum_{j=1}^{I}\left[\xi^{j\phi_g}\left(\prod_{i=1}^{j}\lambda_i\right)\left(j\phi_g\xi^{-2}(j\phi_g-1) + \sum_{i=1}^{j}\frac{2j\phi_g\xi^{-1}\lambda_i' - \lambda_i''}{\lambda_i}\right)\right] \tag{B.44}$$

$\lambda_i''$ is given by:

$$\lambda_i'' = \lambda_i \left[ -\frac{\lambda_i'}{\lambda_i} \left( (i-1)(\phi_g - 1)\xi^{-1} \right) - (i-1)(\phi_g - 1)\xi^{-2} - \lambda_i(\phi_g - 1)\xi^{-1} \left( i(\phi_g - 1)\xi^{\phi_g - 2} - \mu_0(i-1)\xi^{(i-1)(1-\phi_g)} \right) \right]$$

$$(B.45)$$

For the double differential of $\lambda_i$ we get:

$$\lambda_i'' = \frac{\lambda_i''^2}{\lambda_i} + \frac{\lambda_i}{\xi - 1} \times \left( \frac{(I-1)(\phi - 1)}{\xi^2} - \frac{\lambda_i'}{\lambda_i} \right) \tag{B.46}$$

5   We now possess the identities needed to numerically find the optimum bin scaling for a given number of classes. In figure B2 (c), the optimum scaling, $\xi$, is shown as the solid black line.

## Appendix C: Sensitivity of Diagnosed Mortality Rates to Model Parameters

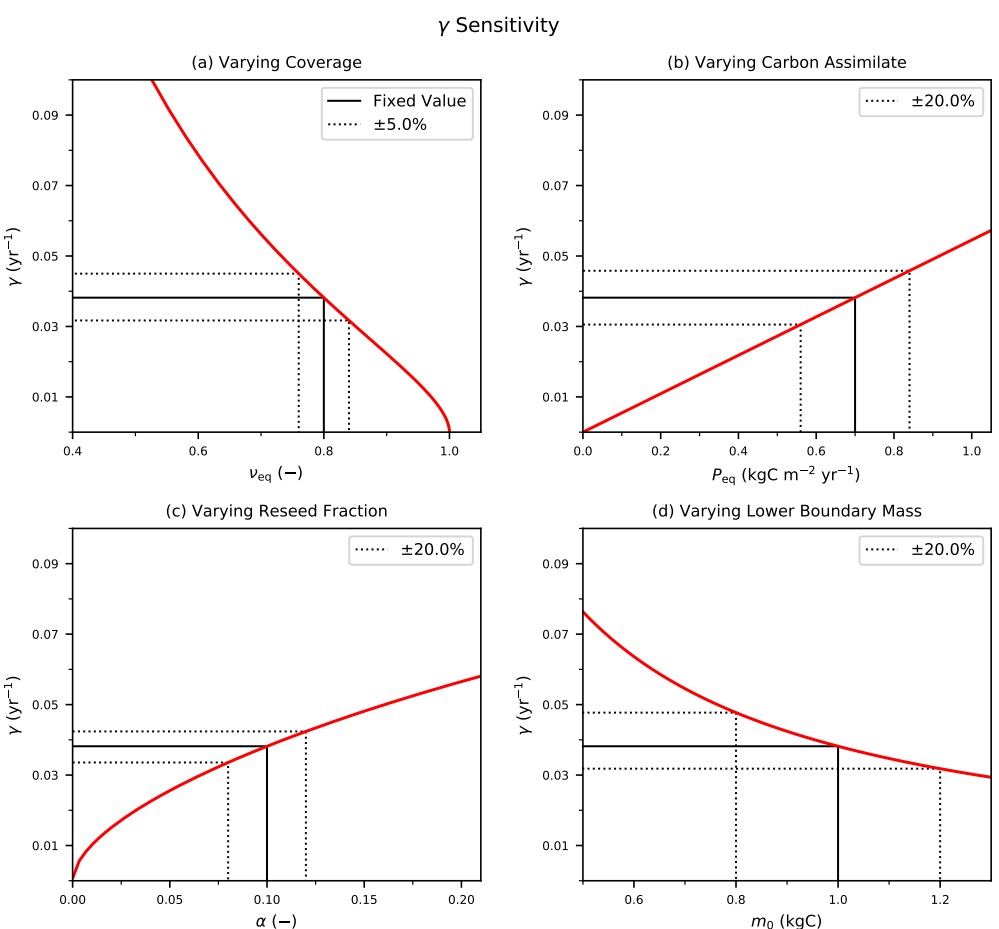

Figure C1. The sensitivity of the mortality rate to assumed input variables: coverage, $\nu_{eq}$ (a), and carbon assimilate rate, $P_{eq}$ (b), and model parameters: reseed fraction, $\alpha$ (c) and boundary mass, $m_0$ (d). The solid black line indicates the fixed values with corresponding $\pm20\%$ (b,c,d) or $\pm5\%$ (a) variation (dotted black lines).

The diagnosed mortality rates in figure 6 are sensitive to variation in model inputs and parameters. The mortality rate, $\gamma$, can be found for the continuous solutions by rearranging the boundary condition equation (6), and substituting in Eq.(B.2) and Eq.(B.13):

$$\gamma = \frac{\alpha P_{eq} a_0}{m_0}\left(\frac{1-\nu_{eq}}{\nu_{eq}}\right)\left[1 + \frac{1}{2\mu_0} + \frac{1}{8\mu_0^2}\right] \tag{C.1}$$

The key external inputs to this equation are the observed PFT fraction $\nu_{eq}$ and the net assimilate $P_{eq}$. In addition, our estimates of $\gamma$ are dependent on the internal model parameters, $\alpha$ and $m_0$.

The red lines in Figure C1 demonstrate how the estimate of $\gamma$ depends on these four inputs. The black dashed lines in Figure C1 indicate how uncertainties in each input relate to uncertainties in $\gamma$, for 'true' values typical of a tree PFT. We estimate uncertainties in the observed PFT fraction (e.g. from remote-sensing) to be $\pm 5\%$, and uncertainties in $P$ (e.g. from JULES) to be $\pm 20\%$, leading to errors of $\pm 17\%$ and $\pm 20\%$ respectively. Likewise, $\pm 20\%$ uncertainties in the internal parameters $\alpha$ and $m_0$ lead to $\pm 12\%$ and $\pm 20\%$ uncertainties in $\gamma$. Combining these sources of uncertainty leads to an overall uncertainty in our inferred estimate of $\gamma$ of about $\pm 35\%$.

*Author contributions.* Originally the model framework in JRM's thesis (Moore, 2016) under the supervision of PMC and CH. The description of PFT competition, the numerical model and the equilibrium solutions has been further developed by APKA, JRM, ABH, and PMC. Currently RED is being integrated into JULES with the supervision of AJW and CJ. AJW also provided and processed the UKESM growth rates needed to drive RED globally within this paper.

5    *Competing interests.*   The authors declare that they have no conflict of interest.

*Acknowledgements.*   This work has been funded as part of the Newton Fund with the Met Office Climate Science for Service Partnership Brazil (CSSP-Brazil) and by the European Research Council (ERC) ECCLES project. A.A was funded through the CASE studentship from the University of Exeter and the Met Office. We are grateful to the Met Office for considering implementation in JULES, via a ticket 902 within a branch of the code repository.

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
