# Peer review of "Robust Ecosystem Demography (RED version 1.0): a parsimonious approach to modelling vegetation dynamics in Earth System Models"

_Geoscientific Model Development, 2019_

## Short Comment (SC1) · 16 Dec 2019

Dear authors,

in my role as Executive editor of GMD, I would like to bring to your attention our Editorial version 1.2:

https://www.geosci-model-dev.net/12/2215/2019/

This highlights some requirements of papers published in GMD, which is also available on the GMD website in the 'Manuscript Types' section:

http://www.geoscientific-model-development.net/submission/manuscript_types.html

In particular, please note that for your paper, the following requirement has not been met in the Discussions paper:

- "The main paper must give the model name and version number (or other unique identifier) in the title."

Please add a version number for RED in the title upon your revised submission to GMD. Yours,

Astrid Kerkweg

—————————————————

**[GMDD](GMDD)**

---

## Referee Comment (RC1) · Anonymous Referee #1 · 14 Jan 2020

In this paper, Argles and co-authors introduce the 'Robust Ecosystem Demographic' (RED) model. RED is introduced as an alternative to cohort-based vegetation demographics models, and its justifications are largely presented as being in opposition to more complex approaches that discretize tree size and age since disturbance. Instead of discretizing age since disturbance and tracking individual cohorts performance, RED makes several simplifying assumptions including:

1. Productivity for each plant size class is not calculated as a function of it's resource availability within a PFT x age class matric (as for a typical ecosystem-demography based VDM) and instead is assumed to scale with plant size, as per the idealized

'Metabolic Scaling Theory'.

2. Thus, there is no possibility of relative plant size determining competition for light, and hence productivity and growth, and thus all plants of a given PFT are supposed to occupy the same area.

3. The horizontal area is divided into said PFT tiles, and another tile wherein disturbance and seed establishment occur.

On account of these various simplifications the model can be solved analytically for a given productivity and mortality rate and RED is generally proposed as an alternative method for the simulation of some aspects of vegetation demographics in Earth system models (ESMs). I appreciate the novelty of this approach, and think it is important that a diversity of avenues are taken towards improving the representation of the terrestrial carbon cycle within ESMs.

While this is an interesting set of concepts, and potentially an interesting 'middle ground' in complexity of representation of vegetation demographics, there are numerous issues with the presentation, description and validation of the model that I find problematic in this paper.

First, in the introduction, there is insufficient explanation of the existing diversity of approaches to the simplification of forest models. A class of models already exists which is much closer conceptually to RED, e.g. the POP (Harvard et al. 2014) and ORCHIDEE-MICT (Yue et al. 2018) models, that also track different size or age cohorts within a single tile devoted to each PFT. While RED additionally provides a DGVM capacity in the form of the competition for seed recruitment, it seems that this class of models certainly requires description at the very least.

Further, despite the numerous mentions of the PPA approach, the paper does not actually describe this alternative approach to defining 'tractable' solutions to demographic modeling. A comparison of the RED and PPA approaches would be interesting, in particular given the fact that the PPA requires slightly more parameters than RED. This is particularly relevant given that the PPA is also implemented in the GFDL ESM.

Instead of a description of the relevant literature, the current justification statements in the introduction focus on somewhat vague assertions that full ED-type size-and-age structured approximations are too cumbersome. A comparison with more similar models would be helpful, as would a more general depiction of the pro's and cons of the approach used here. The model clearly has some benefits in terms of simplicity and tractability, but also has some drawbacks in terms of reduced ecological fidelity compared to real ecosystems. Given this, it would be good if the paper at some point addresses the questions for which RED would and would not be appropriate.

Many demographic model development activities, for example, as specifically motivated by a desire to include greater diversity of functional types in ESMs, and to predict their distribution as a function of their plant traits, which in most models primarily impact upon growth. Removing the ability of the model to simulate growth-based competition for light, and indeed, to simulate a diversity of trees within the same class, means that this RED would not be suitable for that problem.

Further, the introduction suggests that part of the motivation for resolving tree size is to introduce size-dependant physiological processes, but by introducing the metabolic scaling of productivity from an arbitrary reference size to all of the other classes, RED is also unable to simulate how tree size actually affects physiology - e.g. plant hydraulics, light availability, fire damage, allometry (and thus allocation and demand for nutrients in pools of different stoichiometry), size dependant rooting depth (and thus uptake of water and nutrients) , burial by snow, etc. Many developments of demographics models are specifically motivated by the representation of size, so again,RED could not be used for those types of question.

As a parallel, RED also does not provide discretization of the time-since-disturbance continuum, and instead really divides the grid cell into various PFT tiles, with resolved

height, and one 'gap' tile, where new seedlings compete for space. Many demographic model developments are motivated by the ability to represent how the development along the successional trajectory impacts physiological boundary conditions. Examples of this include simulating the dominance of N fixers in early succession, of the matrix of post-fire disturbance conditions (including the vertical co-existance of grass and trees), representation of variation in light conditions to capture successional composition shift and the horizontal variation of vegetation height in systems which are buried by snow.

I am also highly skeptical of the authors claim that mortality rates can be backed out from spatial coverage of a particular PFT. Given that only a single not very convincing validation is presented, I remain far from convinced that this is a reasonable model inversion method. While it might be mathematically plausible, given the myriad simplifying assumptions of the model, I'd like to see how robust the mortality estimates are to variations in the seed production rates and minimum size, as well as assumptions on the spatial arrangement of crowns, and indeed, the uncertainties in the estimates of PFT areal coverage.

Lastly, the paper has some issues with clarity that I have tried to cover in some detail in the following comments. Some of the excessive mathematical details could be moved to the appendix as they rather detract from the flow of the paper. A more informed and nuanced description of how the model fits into existing demographic model literature and of it's strengths and weaknesses would, I think, be more useful for the general readership.

Specific Comments.

P1, L21: The statements in these first three sentences all need references.

P2, L7: I'm not sure what to take from this assertion that uncertainty 'can be attributed' to CO2 responses and regrowth. It can also be attributed to a lot of other features of LSMs. Is it really necessary to state this so definitively?

[Figure]

P2, L9: You didn't really describe or define what a DGVM is yet.

P2, L15: Here it is indicated that 'processes that are dependant on size' is a core motivation for the implementation of this concept, but RED actually ignores that size of all except the reference tree, using an assumption to scale to the other size classes. There are lots of processes that do actually depend on size (hydraulics, allocation, fire mortality, competition for light, wind damage, snow burial, etc.) and so this is a genuine justification for using a size-structured model, but it does not apply to RED. Therefore, a different justification is required.

Further, in ED-type models, the faster regrowth after disturbance is typically predicted on the use of multiple tree types that exist in early, mid and late successional systems (as opposed to an average, slower growing tree).

P2, L21: This is true, but you are also going to get lots of different outcomes of climate change from alternative parameterizations of RED - parameters that are absent from the simpler model are really just assumed to be fixed in RED (e.g. the decay coefficient of productivity with size, seed production, competition parameters). Making the parameters either assumed constants or round numbers doesn't make their uncertainties go away. It would be more interesting to actually look into these uncertainties and illustrate a succession experiment under a range of model assumptions.

P2, L24: Cohort models are numerically unwieldy and no-doubt more expensive, but as you attest later in the paper, it is disingenuous to state that they make a new patch every timestep when in fact ED-derived models immediately fuse the newest patch to the next largest one.

P2, L26: Cohort models can either track tree age or tree size, so adding this here to distinguish RED from a cohort model doesn't really make sense.

P3, L5: The way in which this equation is presented seems overly contrived. Surely it can be presented such that the dn/dt is the sole term on the left hand side?

P3, L7: Neither g(m) nor lambda(m) appear in the actual equation, so this is again a little hard to get ones head around.

P3, L11: What did Niklas and Spatz find or do, briefly?

P3, L16: I do not understand how the last term translates into fractional area, when it looks like it should just return 'area'. Further, is there no constraint on the area the trees can occupy? That seems strange and needs further discussion.

P4, L1: I'm not sure why you need to state that the model conserves carbon three times. All vegetation models must conserve carbon. This isn't very surprising.

P4, L3-15: I'm not sure what purpose is served by this sequence of equations.

P4, L20: This equation would be easier to read if it were split into terms for seed recruitment and growth.

P4, L23: The PPA assumes minimum overlap of crowns within each layer of the canopy. It distinctly does not assume no overlap of PFTs. It assumes that canopies are arranged into layers and within each layer there is no overlap. Competiton for light occurs at the boundary of the layers, and is a strong control on ecosystem assembly. In fact, much of RED is highly contradictory to the PPA concept, given the MST rejects the need to different growth parameters as a function of light availability (as demonstrated convincingly for tropical forests by Farrior et al. 2016). I think it's thus a little disingenuous to cite the PPA here as a justification for this assumption.

P4, L25: "injected"? How do trees get injected?

Figure 1: I don't find figure 1 particularly informative. It would be better to have a depiction of the actual area available for seeds and to illustrate how the different PFTs might affect the allocation to each PFT. This figure just tells me that shrubs are smaller than trees.

P5, L4: The calculation of the area occupied by each PFT, as it is introduced here,

needs a lot more explanation. In the description on L16 of P3, it simply states that the area of all the mass classes is added together, such that there is no overlap between the canopies of the trees in each plant type. This implicitly assumes that all the trees are in the 'canopy' layer, (using PPA terminology) and thus by implication that they should all get the same amount of light. Of course, via use of equation 2, the actual light environment of the plants is divorced from assumptions about their spatial arrangement, but it seems like a strong assumption to me to include no possibility of additional canopy layers. What happens when the total amount of space occupied by the plants exceeds the ground area available?

P5, L5: It should be noted here that the Cox 2001 paper is at-least inspired by the Lotka-Volterra approach, to better allow connection of this concept to community ecology literature.

P5, L7: Later on you state that the coexistence between PFTs of the same type doesn't actually work, so this statement that Eqn 12 allows for coexistence is a little misleading.

P5, L8: This allows succession as you note, but only between the PFT of different classes, not within a given class, unless I'm mistaken? . Figure 3: I'm not really sure what this Figure is supposed to illustrate. What are the red dotted lines in the middle of the triangle? There are three heavy double headed black arrows and not one (as implied by the legend).

Eq 28 and 29: These equations need a bit more explanation and description. This section feels like you are making a concerted effort to lose readers. Is it really necessary that everyone understands how the equilibrium solution of the model is derived? Could this go in an appendix?

P10, L1: As I said above, I am highly skeptical that this is a robust way of estimating turnover, given the uncertainties to do with seed production and spatial extent.

P10, L8: So, productivity was derived from JULES using TRIFFID? Were the outputs

saved for each month? Is there interannual variability? This needs a bit more detail.

P11, L3: Is this really how succession works in Amazonian forests? I think it's really mostly trees that are present in the formation of small to medium sized gaps.

P11, L5: Can you illustrate the dependance on alpha and m0?

P12, L11: What are we to take from this illustration of 'succession' in the model? There isn't any comparison with data, nor an illustration that the model fixes the issue of slow recovery from disturbance that was raised in the introduction. What controls the area fractions of the smaller PFTS? Is there always some gap fraction dedicated to them? How is this equilibrium maintained?

Figure 6: The inputs of productivity taken from JULES do not, for example, allow BETs to grow outside of the tropics, and so many of the critical questions related to the prediction of biome boundaries that are asked of DGVMs cannot be addressed in this circular analysis.

P14, L1: It seems that reproducing the PFT map should be a trivial matter given the productivity inputs illustrated in Figure 6.

P16, L8: This is confusing because the reference to Figure 10 comes before it is described. The use of the mortality rates in these simulations is not described in this section until now.

P17, L1: To what does this 'diagnosed mortality rates' refer? Isn't this sentence about diagnosing mortality rates? This adds another layer of confusion onto my previous comment.

Figure 9: This color map does not allow one to distinguish between most of the lower turnover areas. You need some sort of logarithmic variation in color with mortality rate.

P17, L8: How influential is the minimum recruit size? This definitely needs to be illustrated.

P17, L10: The sentence that begins "Under the assumption" isn't a whole sentence. Moreover, what is the aim of defining a 'healthy' environment? You need to state what you are trying to achieve first. . .

P17, L12: This is a very quick and potentially confusing switch to discussing the growth-mortality ratio and not mortality (you should maybe also re-state what ðİIJĞ is as this is a non-standard quantity.

P17, L13: This number seems extraordinarily high for the stem turnover rate of tropical forests? Comparison with data is, of course, where this aggregation idea is problematic, as mortality rates have clearly been shown to vary with tree size (Lines et al. 2010, Johnson et al. 2018), and thus the range of tree size with which one can compare these rates is unclear, particularly the lower size boundary.

P17, L13: Table 3 contains goodness of fit metrics, and not estimates of mortality.

P17, L15: The 'value within the paper' doesn't state which paper, nor why it needs converting. Thus is very confusing.

P18, L1: This text on the differences between the Moore paper value and this value (which are indeed extraordinarily close and probably don't need excusing) would be better spent describing first how the Moore method differs from RED. This section assumes the reader is familiar with, for example, the non-discretized nature of the Moore method.

P18, L5: "Potentially providing a future constraint on ESM growth rates for PFTs." is not a whole sentence.

Figure 10: The mortality numbers in figure 10 for tropical forests seem too high. (0.07-0.08). Again, it's hard to know what mortality rates they can be compared to. In Table 4, the numbers are different from the figure, perhaps because they are area weighted, but this isn't really clear from the text.

P18, L11: I'm not sure what "within the top 25% of coverages" means, nor what this

is trying to achieve. Further, there is no data in figure 10, so I am not sure why one is supposed to conclude that the model captures the data well. Maybe you actually mean figure 11, which reduces the RED estimate, but only down to about double the observations. Given the a doubled mortality rate is approximately equal to a halved biomass, I'm not sure that this provides a very convincing validation. Further, many estimates of mortality are lower than this. Lewis et al. 2004 find mortality rates of tropical forest from 1.5-1.7%, for example.

P20, L4: I could not find a definition of DET prior to this usage here.

P22, L1-10: I'm not sure what to take from this section about fire. The last line seems to suggest that RED overestimates fire mortality, when figures 12 and 13 seem to show the opposite. The logic of this section needs tightening.

P23, L1-6: This, and the paragraph above, are in need of more references.

P23, L6: This statement about patch merging is incorrect in its assertion that patches can only be merged after a certain age in ED-type models. Further, it does not illustrate that this is actually problematic, and simply asserts as such. Fusion criteria are indeed to some extent arbitrary, but that this is a genuine problem has not actually been demonstrated.

P5, L7: Which important features is it designed to capture exactly? This hasn't really been stated.

P23, L16: Metabolic scaling theory has been widely debunked by numerous studies comparing its predictions with observations (Muller‐Landau et al.20016; Russo et al. 2007; Coomes et al. 2010, Ruger and Condit 2012) in particular where asymmetric competition for light (e.g. in forests) is important.

P23, L18: I am not sure how the seed model allows you to capture the effects of light competition. It allows you to represent the impacts of recruitment competition, but seems to me that it explicit does not include light competition.

P24, L1: It is stated here that equation 12 is a promising method to deal with the problems of coexistence in RED, but equation 12 is already part of RED, thus how can it be the solution? Further, I do not know what 'gap boundary conditions' refers to here.

P24, L4: I am skeptical, without further much more robust testing and illustration, that these relationships would be meaningful.

P24, L13: I do not think that this model is 'based on' the ideas of the PPA in any meaningful way. The idea of the PPA is primarily concerned with how trees fill space, which is specifically ignored by RED, and also on the division of the canopy into discrete layers, which is definitively at-odds with the metabolic scaling method of disaggregating production solely based on tree size. P24, L16: It apparently can be fitted, but I'd argue that there has been no validation presented to show that this is 'effective'.

References

Arora, V. K., Katavouta, A., Williams, R. G., Jones, C. D., Brovkin, V., Friedlingstein, P., Schwinger, J., Bopp, L., Boucher, O., Cadule, P., Chamberlain, M. A., Christian, J. R., Delire, C., Fisher, R. A., Hajima, T., Ilyina, T., Joetzjer, E., Kawamiya, M., Koven, C., Krasting, J., Law, R. M., Lawrence, D. M., Lenton, A., Lindsay, K., Pongratz, J., Raddatz, T., Séférian, R., Tachiiri, K., Tjiputra, J. F., Wiltshire, A., Wu, T., and Ziehn, T.: Carbon-concentration and carbon-climate feedbacks in CMIP6 models, and their comparison to CMIP5 models, Biogeosciences Discuss., https://doi.org/10.5194/bg-2019-473, in review, 2019.

Bohlman, S. Pacala, A forest structure model that determines crown layers and partitions growth and mortality rates for landscape-scale applications of tropical forests. J. Ecol. 100, 508–518 (2012).

Coomes, D.A., Lines, E.R. and Allen, R.B., 2011. Moving on from Metabolic Scaling Theory: hierarchical models of tree growth and asymmetric competition for light. Journal of Ecology, 99(3), pp.748-756.

[Figure]

Farrior, C.E., Bohlman, S.A., Hubbell, S. and Pacala, S.W., 2016. Dominance of the suppressed: Power-law size structure in tropical forests. Science, 351(6269), pp.155-157

Haverd, V., Smith, B., Nieradzik, L.P. and Briggs, P.R., 2014. A stand-alone tree demography and landscape structure module for Earth system models: integration with inventory data from temperate and boreal forests. Biogeosciences, 11(15), pp.4039-4055.

Lewis, S.L., Phillips, O.L., Baker, T.R., Lloyd, J., Malhi, Y., Almeida, S., Higuchi, N., Laurance, W.F., Neill, D.A., Silva, J.N.M. and Terborgh, J., 2004. Concerted changes in tropical forest structure and dynamics: evidence from 50 South American long-term plots. Philosophical Transactions of the Royal Society of London. Series B: Biological Sciences, 359(1443), pp.421-436.

Lines, E.R., Coomes, D.A. and Purves, D.W., 2010. Influences of forest structure, climate and species composition on tree mortality across the eastern US. PLoS One, 5(10), p.e13212

Johnson, D.J., Needham, J., Xu, C., Massoud, E.C., Davies, S.J., Anderson-Teixeira, K.J., Bunyavejchewin, S., Chambers, J.Q., Chang-Yang, C.H., Chiang, J.M. and Chuyong, G.B., 2018. Climate sensitive size-dependent survival in tropical trees. Nature ecology & evolution, 2(9), p.1436. Muller‐Landau, H.C., Condit, R.S., Chave, J., Thomas, S.C., Bohlman, S.A., Bunyavejchewin, S., Davies, S., Foster, R., Gunatilleke, S., Gunatilleke, N. and Harms, K.E., 2006. Testing metabolic ecology theory for allometric scaling of tree size, growth and mortality in tropical forests. Ecology letters, 9(5), pp.575-588.

Rüger, N. and Condit, R., 2012. Testing metabolic theory with models of tree growth that include light competition. Functional Ecology, 26(3), pp.759-765.

Russo, S.E., Wiser, S.K. and Coomes, D.A., 2007. Growth–size scaling relationships

of woody plant species differ from predictions of the Metabolic Ecology Model. Ecology Letters, 10(10), pp.889-901.

Yue, C., Ciais, P., Luyssaert, S., Li, W., McGrath, M. J., Chang, J., and Peng, S.: Representing anthropogenic gross land use change, wood harvest, and forest age dynamics in a global vegetation model ORCHIDEE-MICT v8.4.2, Geosci. Model Dev., 11, 409–428, https://doi.org/10.5194/gmd-11-409-2018, 2018.

---

## Referee Comment (RC2) · Anonymous Referee #2 · 23 Jan 2020

The authors present a model development work on vegetation demography, and seek to incorporate it into an earth system model. The framework provides a simplified solution to model the global vegetation distribution based on the "Metabolic Scaling theory". Both the topic and the model concept are very interesting. However, there are numerous errors and ambiguous expressions throughout the current manuscript. The model descriptions are not clear enough, especially for the equations and units. At some points, I have to stop to calculate the units of each term. I'm also not fully convinced by the model outputs and validations. Extra information are necessary to be provided for a proper judgement, e.g., how the NPP data was created, which climate forcing and vegetation map were used. I suggest an overall revision and reorgnization of the

manuscript. My major question about this approach is how it can be used in transit-time simulations, especially for the future projections. From a modelling aspect, the model simply ignored many factors that can be modfied due to climate change. Nevertheless, it would be very exciting if enough evidences support that some important emergent properties from land ecosystems would remain constant in a fast changing world.

Specific comments:
P1
Abstract L7:cohort-based models?
..L8:These models
..L14:I feel it should not be the major reason to argue that RED would be a great contribution. Only mentioning the computing cost is not convincible enough.
..L15:pdf?
..L19:solvable?
..L26:Why only compared to this dataset.
Introduction:
..L41:2K? not clear enough, references needed
..L47:keep update with the new results?
..L44-51: The logic here is unclear. I assume that the authors want to stress the large uncertainties in modeling land C budget. But the topic of the study is model development, rather than uncertainty analysis. So I suggest to use 1-2 sentences to describe the uncertainty topic, and go to the model development faster.
..L53: According to my knowledge, LUC prediction is from another sector, which is not from DGVM.
Provide the LUC examples here seems irrelevant to the modeling of this study. Also, why the authors only picked examples from RCP8.5.
P2
..Line 2: Rewrite the sentence and focus on the topic of this study. Generally, DGVM includes biochemical, biogeographical, biophysical processes and other factors influencing vegetation.
..Line 5: How to define complex. What about the other "complex" models.
..Line 10: Why non-individual based models cannot do that?
..Line 13: What is top-down models? Area based?
..Line 15: are significantly simpler and more computationally efficient(reference?).
..Line 17: over-estimated(reference?)
..Line 34: The previous paragragh only explain one benefit of RED: reduce computational cost. To me, it is at least not the major reason for the RED development. I feel it is necessary to mention the theoretical foundations for RED development, e.g., the scaling theory. Although this study is mainly about model development, the explanation of the underlying mechanisms is necessary to facilitate the understanding of the model concept.

Description of the model
Overall, the equations should be carefully checked and the units need to be added in an appropriate way.
..Line 47-49: Check the symbol consistency between equ.1 and the corresponding descriptions. I suppose the equation has been simplified – it is assumed that gamma is independent from mass level already.
..Line 50: Any form of what?
..Line 53: follows a power..
..Line 59: Correct the reference format
..Line 70: Is that a basic requirement to build a vegetation model?
..Line 86: keep unit unified throughout the MS. why using per plant per unit area previously but using explicit unit here?
..Line 88: why it is a concern? To keep mass and energy balance is basic to develop a model.
..Line 66 the area term "a" does not appear before.

P3:
..Line 8: P has been defined before. Again, units miss

[Figure]

..Line 17: This part is mainly derived from PPA and TRIFFID, or new for RED? If it is former, I suggest to provide main equations and introduce them briefly.

P4:

..Line 1: I'm concerned about the "coupling" here. Based on the description, I feel RED has not been actually coupled with the ESM. Using prescribed NPP means an implicit vegetation distribution in itself. From equ.16, higher NPP would mean higher baseline growth-rate.

..Line 53-54: What is the loss of vegetation C due to plants growing beyond the modelled mass classes.

P7:

For the first paragraph of "Modelling results", Should it be part of the method section?

..Line 1: What tests?

..Line 2: Again, I'm concerned about the use of prescribed NPP. How you get NPP? Using which climate forcing? What period of NPP you used. And most importantly, how the NPP data from JULES defines the vegetation distribution? A predefined data or from a model? All the info needs to be added for a proper judgement. If fed a similar pattern from the data:ESA LC CCI to RED, then it is not surprising that they would have the similar output as showed in Figure 7.

..Line 10: Why choose this grid-box.

P16:

..Line 1: Discussion. The comparisons between RED and the other similar models are needed. But before that, I think the method description needs to be greatly improved, and the corresponding results should be further clarified.

---

## Author Comment (AC1) · 24 Mar 2020

Dear Professor,

We now have have now included a version number in our title: "***Robust Ecosystem Demography (RED version 1.0): a parsimonious approach to modelling vegetation dynamics in Earth System Models***".

Yours sincerely,

Arthur P. K. Argles on behalf of co-authors.

---

## Author Comment (AC2) · 24 Mar 2020

**Response to Reviewer 2:**

**Robust Ecosystem Demography (RED): a parsimonious approach to modelling vegetation dynamics in Earth System Models**

Arthur P. K. Argles, Jonathan R. Moore, and Peter M. Cox on behalf of co-authors. (*on behalf of the co-authors*)
24th March 2020

The referee was mainly concerned with specific detail, they did raise other points but mainly found that the paper was a bit confusing. We address each of the queries raised below. The relevant reviewer comments are written in italics below followed by our responses in plain font, changes are detailed in blue font.

> **Reviewer**:
>
> *The authors present a model development work on vegetation demography, and seek to incorporate it into an earth system model. The framework provides a simplified solution to model the global vegetation distribution based on the "Metabolic Scaling theory". Both the topic and the model concept are very interesting. However, there are numerous errors and ambiguous expressions throughout the current manuscript. The model descriptions are not clear enough, especially for the equations and units. At some points, I have to stop to calculate the units of each term. I'm also not fully convinced by the model outputs and validations. Extra information are necessary to be provided for a proper judgement, e.g., how the NPP data was created, which climate forcing and vegetation map were used. I suggest an overall revision and reorgnization of the manuscript. My major question about this approach is how it can be used in transit-time simulations, especially for the future projections. From a modelling aspect, the model simply ignored many factors that can be modfied due to climate change. Nevertheless, it would be very exciting if enough evidences support that some important emergent properties from land ecosystems would remain constant in a fast changing world.*

**Response (1)**:
We thank the reviewer for their comments and have sought to make edits that make the model paper clearer and help clarify definitions. On the point of the NPP data, we ran RED offline using outputs from the UKESM climate model. UKESM calculates phenology and litter fluxes using climatic data per area of each PFT (rather than per gridbox). We have elaborate further within the discussion on how RED can be used in transient simulations of future climate simulations. RED was built to be parameter sparse to reduce uncertainty at the global level. As seen from the results within the paper it is possible to capture regional vegetation accurately even within such a parasimonious model.

**Edit**:

"RED is currently being integrated into the JULES Land Surface Model replacing TRIFFID as DGVM. Significant improvements in representation of biogeochemical cycle of droughts, simulating stomata conductance /xylem embolism (SOX) (Eller et al., 2018, 2020) along with the non-structural carbohydrate model (SUGAR) (Jones et al., 2019) and fire through the INFERNO

model (Burton et al., 2019) are being developed. In the future size, related mortality and growth rates can be taken as inputs from these independent models and the updated demographic state given back. We see this a promising avenue for research understanding the resilience of regional ecosystems under climate change."

**Specific Comments**

**Reviewer**:
*P1 Abstract*
*L7: cohort-based models?*

**Response (2)**:
We further elaborate on this term within the abstract.

**Edit**:

"More advanced cohort-based patch models are now becoming established in the latest DGVMs. These models typically attempt to simulate the size-distribution of trees as a function of both tree-size (mass or trunk diameter) and age (time since disturbance)."

**Reviewer**:
*..L8:These models*

**Response (3)**:
Corrected "These typically..." to "These models typically...".

**Reviewer**:
*..L14:I feel it should not be the major reason to argue that RED would be a great contribution. Only mentioning the computing cost is not convincible enough.*

**Response (4)**:
We agree that this is not the definitive reason for the development of RED. Indeed the development of RED is driven by the need to have a robust and parameter sparse model of forest demography for global applicationslaw of parsimony. We therefore state that the additional problem arising from the balance of representation of ecological processes versus the number of uncertain parameters.

**Edit**:

"This approach can capture the overall impact of stochastic disturbance events on the forest structure and biomass, but at the cost of increasing the number of parameters and some ambiguity when updating the probability density function (pdf) in two-dimensions."

> **Reviewer**:
> *..L15:pdf?*

**Response (5)**:
We have appended "(pdf)" to the initial mention of "probability density function..." in the sentence beforehand.

> **Reviewer**:
> *..L19:solvable?*

**Response (6)**:
Corrected typo.

> **Reviewer**:
> *..L26:Why only compared to this dataset*

**Response (7)**:
We compare to this dataset partly because this dataset is classified using the same PFTs used within the UKESM.

> **Reviewer**:
> *..L41:2K? not clear enough, references needed*

**Response (8)**:
We have added in a reference to the Paris Agreement and changed the units to degree centigrade.

**Edit**:

"This is an important component of the total carbon budget consistent with avoiding global warming thresholds, such as 2°C (Schleussner et al., 2016)."

> **Reviewer**:
> *..L47:keep update with the new results?*

**Response (9)**:
We have included the new data from GCB 2019 (Friedlingstein et al., 2019)

> **Reviewer**:
> *..L44-51: The logic here is unclear. I assume that the authors want to stress the large uncertainties in modeling land C budget. But the topic of the study is model development, rather than uncertainty analysis. So I suggest to use 1-2 sentences to describe the uncertainty topic and go to the model development faster.*

**Response (10)**:

We agree that the motivation for including land C budget could be more concise. Uncertainty arising from the representation and parameterisation of processes is part of the motivation for RED. We have also included more discussion of other published models.

> **Reviewer**:
> *..L53: According to my knowledge, LUC prediction is from another sector, which is not from DGVM. Provide the LUC examples here seems irrelevant to the modeling of this study. Also, why the authors only picked examples from RCP8.5.*

**Response (11)**:
Agreed. Therefore, we only mention it in passing;

**Edit**:

"Beyond the fertilisation effect and land-use change, significant uncertainty arises from the representation of vegetation demographics such as recruitment, compeitition and mortality (Brovkin et al., 2013; Ahlström et al., 2015)."

> **Reviewer**:
> *P2*
> *..Line 2: Rewrite the sentence and focus on the topic of this study. Generally, DGVM includes biochemical, biogeographical, biophysical processes and other factors influencing vegetation.*

**Response (12)**:
We have changed the sentence to be more encompassing of what a DGVM includes.

**Edit**:

"The transient representation of plant communities within Earth System Models (ESMs) is achieved through the use of Dynamic Global Vegetation Models (DGVMs). DGVMs employ a variety of biophysical, biogeographical and biochemical processes to simulate growth, competition and recruitment of vegetation. The variety in the number and resolution of the processes helps to contribute to the differences found at the Earth System level."

> **Reviewer**:
> *..Line 5: How to define complex. What about the other "complex" models.*

**Response (13)**:
We have clarified this as "individual based models".

> **Reviewer**:
> *..Line 10: Why non-individual based models cannot do that?*

**Response (14)**:
Valid point, we now have redefined this as:

**Edit**:

"In the second-instance, individual models can explicitly represent a multitude of biological and ecosystem processes at a individual plant level (Smith, 2001; Sato et al., 2007)."

> **Reviewer**:
> *..Line 13: What is top-down models? Area based?*

**Response (15)**:
Yes, we think of top-down models as phenomenological models such as Lotka-Volterra. We clarify this point in the introduction.

**Edit**:

"DGVMs often range from the simplistic, older, top-down approach to that of complex individual-based DGVMs. For example, in the first instance the TRIFFID model (Cox, 2001; **?**) simulates the fractional area of each Plant Functional Type (PFT) using phenomenological Lotka-Volterra equations."

> **Reviewer**:
> *..Line 15: are significantly simpler and more computationally efficient(reference?).*

**Response (16)**:
We have edited the paragraph in the model description removing this statement.

> **Reviewer**:
> *Line 17: over-estimated(reference?)*

**Response (17)**:
We have now provided a reference: (Burton et al., 2019).

> **Reviewer**:
> *..Line 34: The previous paragragh only explain one benefit of RED: reduce computational cost. To me, it is at least not the major reason for the RED development. I feel it is necessary to mention the theoretical foundations for RED development, e.g., the scaling theory. Although this study is mainly about model development, the explanation of the underlying mechanisms is necessary to facilitate the understanding of the model concept.*

**Response (18)**:
A valid point. We have now stated the theoretical foundations of metabolic scaling theory. Added onto the last description of the introduction:

**Edit**:

"This paper presents a simplified cohort model (*Robust Ecosystem Demography (RED)*) which updates the number of trees in each mass class, but does not separately track tree-age or patch-age.

RED assumes that the tree size-distribution of a forest is determined by how the rates of tree growth and mortality vary with tree size (Kohyama et al., 2003; Coomes et al., 2003; Muller-Landau et al., 2006; Lima et al., 2016). We follow many other studies in assuming that tree-growth rates vary with the three-quarter power of tree mass ($m^{3/4}$), as suggested by metabolic scaling theory (West et al., 1997). Where tree mortality rate can also be assumed to be approximately independent of tree mass, the demographic equation yields equilibrium tree-size distributions which follow a Wiebull distriubution – this is sometimes termed 'Demographic Equilibrium Theory (DET)' (see Appendix B). These simplifications significantly reduce the number of free parameters in RED, but still enable it to fit forest inventory data in North America (Moore et al., 2018) and South America (Moore et al., 2020)."

**Reviewer**:
*Description of the model: Overall, the equations should be carefully checked, and the units need to be added in an appropriate way.*

**Response (19)**:
The units and equations have been thoroughly checked for this study and other related papers (Moore et al., 2018, 2020). We now also explicitly point to the table of variables, definitions and units in the Appendix A.

**Edit**:

 "A full list of variables, definitions and units are given in appendix A."

**Reviewer**:
*..Line 47-49: Check the symbol consistency between equ.1 and the corresponding descriptions. I suppose the equation has been simplified – it is assumed that gamma is independent from mass level already.*

**Response (20)**:
Edited for consistency.

**Reviewer**:
*..Line 50: Any form of what?*

**Response (21)**:
Edited to say: "of relationship with size".

**Reviewer**:
*..Line 53: follows a power..*

**Response (22)**:
Corrected.

**Reviewer**:

*..Line 59: Correct the reference format*

**Response (23)**:
Corrected.

**Reviewer**:
*..Line 70: Is that a basic requirement to build a vegetation model?*

**Response (24)**:
Yes - in the context of the carbon cycle and Earth System Modelling.

**Reviewer**:
*..Line 86: keep unit unified throughout the MS. why using per plant per unit area previously but using explicit unit here?*

**Response (25)**:
We now declare units to keep consistency throughout the manuscript at first mention.

**Reviewer**:
*..Line 88: why it is a concern? To keep mass and energy balance is basic to develop a model.*

**Response (26)**:
We have re-phrased this statement.

**Reviewer**:
*..Line 66 the area term "a" does not appear before.*

**Response (27)**:
The mean crown area "$a$'" - is defined in the previous paragraph.

**Reviewer**:
*P3:*
*..Line 8: P has been defined before. Again, units miss*

**Response (28)**:
We have now added units.

**Reviewer**:
*..Line 17: This part is mainly derived from PPA and TRIFFID, or new for RED? If it is former, I suggest to provide main equations and introduce them briefly.*

**Response (29)**:
These equations are developed for use in RED. We have removed the reference to PPA in response to other reviewer comments.

*P4:..Line 1: I'm concerned about the "coupling" here. Based on the description, I feel RED has not been coupled with the ESM. Using prescribed NPP means an implicit vegetation distribution in itself. From equ.16, higher NPP would mean higher baseline growth-rate.*

**Response (30)**:
RED was run offline using NPP and litter outputs from a UKESM run, there is no coupling. The UKESM runs were in terms of PFT area instead of grid-box area, therefore multiplying by coverage circumvents this issue. We have clarified this point in our introduction.

**Reviewer**:
*..Line 53-54: What is the loss of vegetation C due to plants growing beyond the modelled mass classes*

**Response (31)**:
The truncated growth $g_I N_I$ as seen within the demographic litter equation. However, this term is negligibly small because we resolve a large mass class range that is very unlikley to be exceeded.

**Reviewer**:
*P7:*
*For the first paragraph of "Modelling results", Should it be part of the method section?*

**Response (32)**:
No we don't think so. This paragraph is part of the explicit set-up rather than then the method and helps the results section have improved 'flow'.

**Reviewer**:
*..Line 1: What tests?*

**Response (33)**:
Response: Changed "tests" to "run".

**Reviewer**:
*..Line 2: Again, I'm concerned about the use of prescribed NPP. How you get NPP? Using which climate forcing? What period of NPP you used. And most importantly, how the NPP data from JULES defines the vegetation distribution? A predefined data or from a model? All the info needs to be added for a proper judgement. If fed a similar pattern from the data: ESA LC CCI to RED, then it is not surprising that they would have the similar output as showed in Figure 7.*

**Response (34)**:
For the sake of clarity, we now state that the UKESM data is defined by unit of vegetation

area rather than grid-box and include about the timescale of the dataset. We already state that this is a model inversion and is therefore essentially tuning the mortality rate within RED to fit the data.

**Edit**:

 "The UKESM simulation ran on a yearly time-step, and provides NPP and local litterfall per unit PFT. We multiply by PFT fraction to get the grid-box mean values required to drive RED (using ESA landcover data, as explained below)."

> **Reviewer**:
> *..Line 10: Why choose this grid-box*

**Response (35)**:
We choose this grid-box because it demonstrates a successional tropical sequence with many PFTs from bare soil. We could have shown many others.

> **Reviewer**:
> *P16:*
> *..Line 1: Discussion. The comparisons between RED and the other similar models are needed. But before that, I think the method description needs to be greatly improved, and the corresponding results should be further clarified.*

**Response (36)**:
Agreed. We have now included a comparison to other DGVMs which include forest demography within the discuss. Further we have tried be more clearer within the model description by keeping consistency in the equations and by moving some of more mathematically excessive sections into Appendix B. In addition to the above edits on the results, we have also reorientated the sections within the results section to improve the papers flow.

**Edit**:

 "In a similar vein a few other models have limited the number of cohort dimensions, for example looking at using patch-age while using allometric relationships to capture size scale. Firstly the POP model (Haverd et al., 2014), uses stand-age cohorts as the dimension for population dynamics, every time-step applying crowding and resource limited mortality rates. Another ex- ample is the ORCHIDEE-MICT (Yue et al., 2018), which disaggrates the populations of a PFT into patch "Cohort" functional types, with transitions between cohorts diagnosed when the average basal diameter passes a threshold."

**Edit**:

 Finally, we assume that light-competition is only significant for the lowest 'seedling' mass class. This enables us to capture the impacts of light competition on seedling emergence through a simple 'gap' boundary condition. This represents a significant simplication compared to other approaches involving the Perfect Placisity Assumption (PPA), as used within other DGVMs such as LM3-PPA or CLM(ED) (Fisher et al., 2015; Weng et al., 2015), where canopies are assumed to perfectly fill

gaps through photomorphism (Strigul et al., 2008). In LM3-PPA the radiative flux is limited by the available gap fraction in a given crown layer. PPA parallels our gap boundary condition at the lowest mass class (Equation (11)), but in RED the growth of a cohort is purely dictated by the the disaggregation of total growth assimilate assuming metabolic scaling (Equation (16)).

**Bibliography**

Ahlström, A., Xia, J., Arneth, A., Luo, Y., and Smith, B. (2015). Importance of vegetation dynamics for future terrestrial carbon cycling. *Environmental Research Letters*, 10(5):054019.

Brovkin, V., Boysen, L., Arora, V. K., Boisier, J. P., Cadule, P., Chini, L., Claussen, M., Friedlingstein, P., Gayler, V., Van den hurk, B. J., Hurtt, G. C., Jones, C. D., Kato, E., De noblet ducoudre, N., Pacifico, F., Pongratz, J., and Weiss, M. (2013). Effect of anthropogenic land-use and land-cover changes on climate and land carbon storage in CMIP5 projections for the twenty-first century. *Journal of Climate*, 26(18):6859–6881.

Burton, C., Betts, R., Cardoso, M., Feldpausch, T. R., Harper, A., Jones, C. D., Kelley, D. I., Robertson, E., and Wiltshire, A. (2019). Representation of fire, land-use change and vegetation dynamics in the joint uk land environment simulator vn4. 9 (jules). *Geoscientific Model Development*, 12(1):179–193.

Coomes, D. A., Duncan, R. P., Allen, R. B., and Truscott, J. (2003). Disturbances prevent stem size-density distributions in natural forests from following scaling relationships. *Ecology Letters*, 6(11):980–989.

Cox, P. M. (2001). Description of the" triffid" dynamic global vegetation model.

Eller, C. B., Rowland, L., Mencuccini, M., Rosas, T., Williams, K., Harper, A., Medlyn, B. E., Wagner, Y., Klein, T., Teodoro, G. S., et al. (2020). Stomatal optimisation based on xylem hydraulics (sox) improves land surface model simulation of vegetation responses to climate. *New Phytologist*.

Eller, C. B., Rowland, L., Oliveira, R. S., Bittencourt, P. R., Barros, F. V., da Costa, A. C., Meir, P., Friend, A. D., Mencuccini, M., Sitch, S., et al. (2018). Modelling tropical forest responses to drought and el niño with a stomatal optimization model based on xylem hydraulics. *Philosophical Transactions of the Royal Society B: Biological Sciences*, 373(1760):20170315.

Fisher, R. A., Muszala, S., Verteinstein, M., Lawrence, P., Xu, C., McDowell, N. G., Knox, R. G., Koven, C., Holm, J., Rogers, B. M., Spessa, A., Lawrence, D., and Bonan, G. (2015). Taking off the training wheels: the properties of a dynamic vegetation model without climate envelopes, clm4.5(ed). *Geoscientific Model Development*, 8(11):3593–3619.

Friedlingstein, P., Jones, M., O'Sullivan, M., Andrew, R., Hauck, J., Peters, G., Peters, W., Pongratz, J., Sitch, S., Le Quéré, C., et al. (2019). Global carbon budget 2019. *Earth System Science Data*, 11(4):1783–1838.

Haverd, V., Smith, B., Nieradzik, L. P., and Briggs, P. R. (2014). A stand-alone tree demography and landscape structure module for earth system models: integration with inventory data from temperate and boreal forests. *Biogeosciences*, 11(15):4039–4055.

Jones, S., Rowland, L., Cox, P., Hemming, D., Wiltshire, A., Williams, K., Parazoo, N. C., Liu, J., da Costa, A. C. L., Meir, P., Mencuccini, M., and Harper, A. (2019). The impact of a simple representation of non-structural carbohydrates on the simulated response of tropical forests to drought. *Biogeosciences Discussions*, 2019:1–26.

Kohyama, T., Suzuki, E., Partomihardjo, T., Yamada, T., and Kubo, T. (2003). Tree species differentiation in growth, recruitment and allometry in relation to maximum height in a bornean mixed dipterocarp forest. *Journal of Ecology*, 91(5):797–806.

Lima, R. A., Muller-Landau, H. C., Prado, P. I., and Condit, R. (2016). How do size distributions relate to concurrently measured demographic rates? evidence from over 150 tree species in panama. *Journal of Tropical Ecology*, 32(3):179–192.

Moore, J. R., Argles, A. P. K., Zhu, K., Huntingford, C., and Cox, P. M. (2020). Validation of demographic equilibrium theory against tree-size distributions and biomass density in amazonia. *Biogeosciences*, 17(4):1013–1032.

Moore, J. R., Zhu, K., Huntingford, C., and Cox, P. M. (2018). Equilibrium forest demography explains the distribution of tree sizes across North America. *Environmental Research Letters*, 13(8).

Muller-Landau, H. C., Condit, R. S., Harms, K. E., Marks, C. O., Thomas, S. C., Bunyavejchewin, S., Chuyong, G., Co, L., Davies, S., Foster, R., Gunatilleke, S., Gunatilleke, N., Hart, T., Hubbell, S. P., Itoh, A., Kassim, A. R., Kenfack, D., LaFrankie, J. V., Lagunzad, D., Lee, H. S., Losos, E., Makana, J. R., Ohkubo, T., Samper, C., Sukumar, R., Sun, I. F., Nur Supardi, M. N., Tan, S., Thomas, D., Thompson, J., Valencia, R., Vallejo, M. I., Muñoz, G. V., Yamakura, T., Zimmerman, J. K., Dattaraja, H. S., Esufali, S., Hall, P., He, F., Hernandez, C., Kiratiprayoon, S., Suresh, H. S., Wills, C., and Ashton, P. (2006). Comparing tropical forest tree size distributions with the predictions of metabolic ecology and equilibrium models. *Ecology Letters*, 9(5):589–602.

Sato, H., Itoh, A., and Kohyama, T. (2007). SEIB-DGVM: A new Dynamic Global Vegetation Model using a spatially explicit individual-based approach. *Ecological Modelling*, 200(3-4):279–307.

Schleussner, C.-F., Rogelj, J., Schaeffer, M., Lissner, T., Licker, R., Fischer, E. M., Knutti, R., Levermann, A., Frieler, K., and Hare, W. (2016). Science and policy characteristics of the paris agreement temperature goal. *Nature Climate Change*, 6(9):827–835.

Smith, B. (2001). Lpj-guess-an ecosystem modelling framework. *Department of Physical Geography and Ecosystems Analysis. INES, Sölvegatan*, 12:22362.

Strigul, N., Pristinski, D., Purves, D., Dushoff, J., and Pacala, S. (2008). Scaling from trees to forests: tractable macroscopic equations for forest dynamics. *Ecological Monographs*, 78(4):523–545.

Weng, E. S., Malyshev, S., Lichstein, J. W., Farrior, C. E., Dybzinski, R., Zhang, T., Shevliakova, E., and Pacala, S. W. (2015). Scaling from individual trees to forests in an earth system modeling framework using a mathematically tractable model of height-structured competition. *Biogeosciences*, 12(9):2655–2694.

West, G. B., Brown, J. H., and Enquist, B. J. (1997). A general model for the origin of allometric scaling laws in biology. *Science*, 276(5309):122–126.

Yue, C., Ciais, P., Luyssaert, S., Li, W., McGrath, M. J., Chang, J., and Peng, S. (2018). Representing anthropogenic gross land use change, wood harvest, and forest age dynamics in a global vegetation model orchidee-mict v8.4.2. *Geoscientific Model Development*, 11(1):409–428.

---

## Author Response (AR1)

Dear Professor Sierra,

We have now completed the edits to our manuscript titled: *"Robust Ecosystem Demography (RED): a parsimonious approach to modelling vegetation dynamics in Earth System Models"*.

Attached is a file containing the point by point responses to each reviewer's comments, and a latexdiff file showing the extensive changes that we have made to our paper. We believe that these changes have made our study much more coherent and robust.

We have added Dr Anna Harper from the University of Exeter as a co-author, as she provided driving data for our study and also invaluable insights during the revision process.

Thank you in advance for taking the time to look over our revisions. We look forward to your decision.

Yours sincerely,

Arthur Argles and Peter Cox (on behalf of co-authors)

**Response to Reviewer 1:**

Robust Ecosystem Demography (RED): a parsimonious approach to modelling vegetation dynamics in Earth System Models

Arthur P. K. Argles, Jonathan R. Moore, and Peter M. Cox on behalf of co-authors. (*on behalf of the co-authors*)
22nd April 2020

We thank the reviewer for their extremely detailed review. Most of the suggestions were on technicalities within the literature and definitions used within the paper. We address each of the queries raised below. The relevant reviewer comments are written in italics below followed by our responses in plain font with the changes given in blue.

> **Reviewer**:
> *In this paper, Argles and co-authors introduce the 'Robust Ecosystem Demographic' (RED) model. RED is introduced as an alternative to cohort-based vegetation demographics models, and its justifications are largely presented as being in opposition to more complex approaches that discretize tree size and age since disturbance. Instead of discretizing age since disturbance and tracking individual cohorts performance, RED makes several simplifying assumptions including:*
>
> 1. *Productivity for each plant size class is not calculated as a function of it's resource availability within a PFT x age class matrix (as for a typical ecosystem-demography based VDM) and instead is assumed to scale with plant size, as per the idealized 'Metabolic Scaling Theory'.*
>
> 2. *Thus, there is no possibility of relative plant size determining competition for light, and hence productivity and growth, and thus all plants of a given PFT are supposed to occupy the same area.*

**Response (1)**:

While it is true that we do not explicitly model light competition (except with respect to the net growth-rate of the recruitment flux), it is not true that plant size has no impact on growth-rate or that all plants of a given PFT occupy the same area. In RED we assume allometric relationships relating tree mass to growth-rate (Equation 2) and crown area (Equation 3). We have clarified these assumptions.

> **Reviewer**:
>
> 3 *The horizontal area is divided into said PFT tiles, and another tile wherein disturbance and seed establishment occur.*

**Response (2)**:

This is an interesting way of viewing RED, but it's not quite how we see it. In fact, PFT-dependent disturbance occurs across the whole of the grid-box, as does seed establishment. The confusion may arise from the fact that the latter is affected by light-competition

and therefore depends on the unvegetated fraction of the grid-box. We have clarified these points in a revised model description.

**Edit**:

"Spreading is homogeneous across the entirety of the grid-box, but only seedlings established within 'unoccupied space' will survive to join the plant cohort.""

**Edit**:

"Disturbance mortality rates from $\gamma_d$ can in principle be both PFT-dependent and mass-dependent (e.g. to capture forestry practices)."

> **Reviewer**:
> *On account of these various simplifications the model can be solved analytically for a given productivity and mortality rate and RED is generally proposed as an alternative method for the simulation of some aspects of vegetation demographics in Earth system models (ESMs). I appreciate the novelty of this approach, and think it is important that a diversity of avenues are taken towards improving the representation of the terrestrial carbon cycle within ESMs.*
> *While this is an interesting set of concepts, and potentially an interesting 'middle ground' in complexity of representation of vegetation demographics, there are numerous issues with the presentation, description and validation of the model that I find problematic in this paper.*
> *First, in the introduction, there is insufficient explanation of the existing diversity of approaches to the simplification of forest models. A class of models already exists which is much closer conceptually to RED, e.g. the POP (Haverd et al., 2014) and ORCHIDEE-MICT (Yue et al., 2018) models, that also track different size or age cohorts within a single tile devoted to each PFT. While RED additionally provides a DGVM capacity in the form of the competition for seed recruitment, it seems that this class of models certainly requires description at the very least.*

**Response (3)**:
As suggested, we now compare and contrast RED with these other published models, within our revised discussion and introduction.

**Edit**:

"In a similar vein other models have limited the number of cohort dimensions. The POP model (Haverd et al., 2014), uses stand-age cohorts as the dimension for population dynamics, every time-step applying crowding and resource limited mortality rates. Another example is the ORCHIDEE-MICT (Yue et al., 2018), which disaggregates the populations of a PFT into patch cohort functional types, with transitions between cohorts diagnosed when the average basal diameter passes a threshold."

**Edit**:

"This is a distinct approach relative to other intermediate complexity DGVMs which are based on

patches defined by time since disturbance, such as the POP or ORCHIDEE-MICT models (Haverd et al., 2014; Yue et al., 2018)."

> **Reviewer**:
> *Further, despite the numerous mentions of the PPA approach, the paper does not actually describe this alternative approach to defining 'tractable' solutions to demographic modeling. A comparison of the RED and PPA approaches would be interesting, in particular given the fact that the PPA requires slightly more parameters than RED. This is particularly relevant given that the PPA is also implemented in the GFDL ESM. A comparison of the RED and PPA approaches would be interesting, in particular given the fact that the PPA requires slightly more parameters than RED. This is particularly relevant given that the PPA is also implemented in the GFDL ESM.*

**Response (4)**:
We have included a description on PPA in the discussion and how this relates to the minimum overlap assumption within RED.

**Edit**:

"Finally, we assume that competition is only significant for the lowest 'seedling' mass class. This enables us to represent gap dynamics among plants and resultant stages in succession. This represents a significant simplication compared to other approaches involving the Perfect Plasticity Assumption (PPA), as used within DGVMs such as LM3-PPA or CLM(ED) (Fisher et al., 2015; Weng et al., 2015), where canopies are assumed to perfectly fill gaps through photomorphism (Strigul et al., 2008). In LM3-PPA the radiative flux is limited by the available gap fraction in a given crown layer. PPA parallels our gap boundary condition at the lowest mass class (equation (6)), but in RED the growth of a cohort is purely dictated by the the disaggregation of total growth assimilate assuming metabolic scaling (equation (11))."

> **Reviewer**:
> *Instead of a description of the relevant literature, the current justification statements in the introduction focus on somewhat vague assertions that full ED-type size-and age structured approximations are too cumbersome. A comparison with more similar models would be helpful, as would a more general depiction of the pro's and cons of the approach used here. The model clearly has some benefits in terms of simplicity and tractability, but also has some drawbacks in terms of reduced ecological fidelity compared to real ecosystems. Given this, it would be good if the paper at some point addresses the questions for which RED would and would not be appropriate.*

**Response (5)**:
As suggested, we have now included a discussion of the pros and cons of RED, and the implications for its applications, within the revised Discussion section. We have also explained where more complex approaches (such as ED) are required.

**Edit**:

"Our previous work in evaluating demographic equilibrium theory for regional forest inventory datasets in North America (Moore et al., 2018) and using RAINFOR sites for South America (Moore et al., 2020), has provided the theoretical basis for the development of RED. In those studies we found that tree-size distributions within observed forests can be satisfactorily understood in terms of demographic equilibrium in the size dimension alone. This is a reduction in complexity compared to other cohort models which are based on patch age, and yet an improvement in ecological fidelity compared to older phenomenological DGVMs such as TRIFFID (Cox, 2001). The modular design of RED allows for easy coupling to land-surface schemes, merely requiring the per unit grid-box total carbon assimilate rate and any additional mortality disturbance rates as inputs for each grid-box (Figure 2). In principle, RED allows scope for more complex tree size-dependent processes, although in this first study we chose to assume size-independent (but spatially varying) mortality rates for each PFT. Our previous work suggests that this is a good first-order assumption (Moore et al., 2018, 2020)."

**Edit**:

"There are inevitably weaknesses with any particular modelling approach. For RED, a current limitation is for competition to lead to a single PFT at each location within each co-competing vegetation class (i.e. tree, shrub, grass). The PFT with the highest equilibrium fraction will end up excluding sub-dominant PFTs within the same vegetation class. It was necessary for us to account for this eventual competitive exclusion to derive zero-drift steady-states for the global runs presented in Section 3.2.1 . Such competitive exclusion is a common problem in DGVMs (Fisher et al., 2018). Currently, RED would not be the most appropriate DGVM to answer important questions regarding the role of biodiversity in ecosystem function (Pavlick et al., 2013; Levine et al., 2016). More sophisticated DGVMs are required to simulate plant diversity, such as individual-based models (Fischer et al., 2016), and DGVMs specifically-designed to capture sub-gridscale patch dynamics (Longo et al., 019a,b). Adapting our 'gap' boundary condition (equation 7) appears to be a promising way to allow greater PFT diversity in RED, without unduly increasing model complexity. We see this as a key priority for future research."

> **Reviewer**:
> *Many demographic model development activities, for example, as specifically motivated by a desire to include greater diversity of functional types in ESMs, and to predict their distribution as a function of their plant traits, which in most models primarily impact upon growth. Removing the ability of the model to simulate growth-based competition for light, and indeed, to simulate a diversity of trees within the same class, means that this RED would not be suitable for that problem.*

**Response (6)**:
Agreed. The introduction has been rewritten to explain how in RED we choose to trade model complexity for reduced parameter uncertainty. This trade-off seems to be appropriate for the purposes of modelling large-scale forest demography and carbon storage (Moore et al., 2018, 2020), but it is indeed less appropriate for applications related to forest ecology and diversity.

**Reviewer**:

*Further, the introduction suggests that part of the motivation for resolving tree size is to introduce size-dependant physiological processes, but by introducing the metabolic scaling of productivity from an arbitrary reference size to all of the other classes, RED is also unable to simulate how tree size actually affects physiology - e.g. plant hydraulics, light availability, fire damage, allometry (and thus allocation and demand for nutrients in pools of different stoichiometry), size dependant rooting depth (and thus uptake of water and nutrients), burial by snow, etc. Many developments of demographics models are specifically motivated by the representation of size, so again, RED could not be used for those types of question.*

**Response (7)**:

RED is a demography model which requires net PFT growth rates and disturbance rates as inputs. In the study presented in this paper, net PFT growth-rates were provided by the JULES land-surface scheme. In principle, details on the tree size distribution can be fed back into JULES (or any other land-surface scheme) to enable size-dependent processes to be included (for example to represent size-dependent drought mortality). We include these possible future developments in our new Discussion section.

**Edit**:

"RED is currently being coupled to the JULES Land Surface Model, replacing TRIFFID as the default DGVM within that framework. In parallel, significant improvements are being made to the representation of physiological processes in JULES, most notably through the representation of non-structural carbohydrate ('SUGAR', Jones et al. (2019)), and through the inclusion of a coupled model of stomatal conductance and hydraulic failure under drought stress ('SOX', Eller et al. (2018, 2020)). Plans are also being made to derive the mortality rates for RED from the INFERNO forest-fire model (Burton et al., 2019). These developments will allow us to simulate the effects of size-dependent tree mortality rates within the near future."

**Reviewer**:

*As a parallel, RED also does not provide discretization of the time-since-disturbance continuum, and instead really divides the grid cell into various PFT tiles, with resolved height, and one 'gap' tile, where new seedlings compete for space. Many demographic model developments are motivated by the ability to represent how the development along the successional trajectory impacts physiological boundary conditions. Examples of this include simulating the dominance of N fixers in early succession, of the matrix of post-fire disturbance conditions (including the vertical co-existance of grass and trees), representation of variation in light conditions to capture successional composition shift and the horizontal variation of vegetation height in systems which are buried by snow.*

**Response (8)**:

As pointed out above (response 2), this picture of RED is not correct. We have clarified this in the revised model description. The other points concerning the age-dependent (rather than size-dependent) processes are covered in our revised introduction and discussion (see our response 3).

**Reviewer**:
*I am also highly skeptical of the authors claim that mortality rates can be backed out from spatial coverage of a particular PFT. Given that only a single not very convincing validation is presented, I remain far from convinced that this is a reasonable model inversion method. While it might be mathematically plausible, given the myriad simplifying assumptions of the model, I'd like to see how robust the mortality estimates are to variations in the seed production rates and minimum size, as well as assumptions on the spatial arrangement of crowns, and indeed, the uncertainties in the estimates of PFT areal coverage.*

**Response (9)**:
As requested, we have carried-out a senstivity analysis to show how our estimates of mortality-rates depend on the model parameters ($\alpha$, $m_0$) along with the 'observed' PFT areal coverage and UKESM carbon assimilate input. This is included as a new Appendix C.

**Edit**:

The diagnosed mortality rates in figure 6 are sensitive to variation in model inputs and parameters. The mortality rate, $\gamma$, can be found for the continuous solutions by rearranging the boundary condition equation (6), substituting in Eq.(B2) and Eq.(B13):

$$\gamma = \frac{\alpha P_{\text{eq}} a_0}{m_0} \left( \frac{1 - \nu_{\text{eq}}}{\nu_{\text{eq}}} \right) \left[ 1 + \frac{1}{2\mu_0} + \frac{1}{8\mu_0^2} \right] \tag{1}$$

The key external inputs to this equation are the observed PFT fraction $\nu_{eq}$ and the net assimilate $P_{eq}$. In addition, our estimates of $\gamma$ are dependent on the internal model parameters, $\alpha$ and $m_0$. The red lines in Figure C1 demonstrate how the estimate of $\gamma$ depends on these four inputs. The black dashed lines in Figure C1 indicate how uncertainties in each input relate to uncertainties in $\gamma$, for 'true' values typical of a tree PFT. We estimate uncertainties in the observed PFT fraction (e.g. from remote-sensing) to be $\pm 5\%$, and uncertainties in $P$ (e.g. from JULES) to be $\pm 20\%$, leading to errors of $\pm 17\%$ and $\pm 20\%$ respectively. Likewise, $\pm 20\%$ uncertainties in the internal parameters $\alpha$ and $m_0$ lead to $\pm 12\%$ and $\pm 20\%$ uncertainties in $\gamma$. Combining these sources of uncertainty leads to an overall uncertainty in our inferred estimate of $\gamma$ of about $\pm 35\%$. "

**Reviewer**:
*Lastly, the paper has some issues with clarity that I have tried to cover in some detail in the following comments. Some of the excessive mathematical details could be moved to the appendix as they rather detract from the flow of the paper. A more informed and nuanced description of how the model fits into existing demographic model literature and of it's strengths and weaknesses would, I think, be more useful for the general readership.*

**Response (10)**:
As suggested, we have moved mathematical details concerning carbon conservation to appendix A and the equilibrium state to Appendix B. The pros and cons of RED are

[Figure]

Figure 1: The sensitivity of the mortality rate to assumed input variables: coverage, $\nu_{eq}$ (a), and carbon assimilate rate, $P_{eq}$ (b), and model parameters: reseed fraction, $\alpha$ (c) and boundary mass, $m_0$ (d). The solid black line indicates the fixed values with corresponding $\pm 20\%$ (b,c,d) or $\pm 5\%$ (a) variation (dotted black lines).

discussed in more detail in our revised Discussion section (as outlined abov in response 5).

**Specific Comments**

**Reviewer**:
*P1, L21: The statements in these first three sentences all need references.*

**Response (11)**:
Done.

**Edit**:

"A key requirement of Earth System Science is to estimate how much carbon the land surface will take-up in the decades ahead (Ciais et al., 2014). This is an important component of the total carbon budget consistent with avoiding global warming thresholds, such as 2 C (Schleussner et al., 2016). Unfortunately, projections of future land carbon storage still span a wide-range (Brovkin et al., 2013; Friedlingstein et al., 2014; Arora et al., 2019).

*P2, L7: I'm not sure what to take from this assertion that uncertainty 'can be attributed' to CO2 responses and regrowth. It can also be attributed to a lot of other features of LSMs. Is it really necessary to state this so definitively?*

**Response (12)**:

Agreed - we have now removed this phrase.

**Reviewer**:

*P2, L9: You didn't really describe or define what a DGVM is yet.*

**Response (13)**:

DGVM now defined at the point of use.

**Edit**:

"The representation of plant communities within Earth System Models (ESMs) is achieved through the use of Dynamic Global Vegetation Models (DGVMs). DGVMs employ a variety of biophysical, biogeographical and biochemical processes to simulate growth, competition and recruitment of vegetation. The variety in the number and resolution of the processes contributes to the differences found at the Earth System level."

**Reviewer**:

*P2, L15: Here it is indicated that 'processes that are dependant on size' is a core motivation for the implementation of this concept, but RED actually ignores that size of all except the reference tree, using an assumption to scale to the other size classes. There are lots of processes that do actually depend on size (hydraulics, allocation, fire mortality, competition for light, wind damage, snow burial, etc.) and so this is a genuine justification for using a size-structured model, but it does not apply to RED. Therefore, a different justification is required.*

*Further, in ED-type models, the faster regrowth after disturbance is typically predicted on the use of multiple tree types that exist in early, mid and late successional systems (as opposed to an average, slower growing tree).*

**Response (14)**:

Please see our responses to points in (1), (2) and (7) above.

**Reviewer**:

*P2, L21: This is true, but you are also going to get lots of different outcomes of climate change from alternative parameterizations of RED - parameters that are absent from the simpler model are really just assumed to be fixed in RED (e.g. the decay coefficient of productivity with size, seed production, competition parameters). Making the parameters either assumed constants or round numbers doesn't make their uncertainties go away. It would be more interesting to investigate these uncertainties and illustrate a succession experiment under a range of model assumptions.*

**Response (15):**
Previous studies testing RED equilibrum profiles against observed forest demography for north and south America (Moore et al., 2018, 2020), suggest that our simplifying assumptions are sufficient to capture tree size distributions in many locations. However, we agree that it is important to assess the sensitivity of our simulations to the assumed fixed parameters. This is why we have included a new senstivity analysis in Appendix C, as per (9) above.

> **Reviewer:**
> *P2, L24: Cohort models are numerically unwieldy and no-doubt more expensive, but as you attest later in the paper, it is disingenuous to state that they make a new patch every timestep when in fact ED-derived models immediately fuse the newest patch to the next largest one.*

**Response (16):**
The algorithm used to 'fuse' patches is arguably an arbitrary feature of such models. However, we have toned-down our implied criticism of these alternative approaches in the introduction.

> **Reviewer:**
> *P2, L26: Cohort models can either track tree age or tree size, so adding this here to distinguish RED from a cohort model doesn't really make sense.*

**Response (17):**
The norm for cohort model is to track both tree age and tree size. As stated within the sentence RED is a "simplified" cohort model, the simplification being not tracking tree size.

> **Reviewer:**
> *P3, L5: The way in which this equation is presented seems overly contrived. Surely it can be presented such that the dn/dt is the sole term on the left hand side?*

**Response (18):**
We choose to write the equation in this way, because the lhs is essentially the continuity equation for a conserved variable (in this case tree number), while the rhs contains the source and sink terms. This is a standard way to write Fokker-Planck type equations.

> **Reviewer:**
> *Neither g(m) nor lambda(m) appear in the actual equation, so this is again a little hard to get ones head around.*

**Response (19):**
The terms $g$ and $\gamma$ (Rather than lambda?) appear in equation 1. We choose not to write these explicitly as functions of mass (e.g. "$g(m)$") for clarity. Again, this is standard practice.

**Response (20)**:

We have edited this sentence to include "...consistent with the meta-analysis of field-based measurements by of Niklas and Spatz (2004).."

**Reviewer**:

*P3, L16: I do not understand how the last term translates into fractional area, when it looks like it should just return 'area'. Further, is there no constraint on the area the trees can occupy? That seems strange and needs further discussion.*

**Response (21)**:

This term returns fractional area because of the dimensions of "$a$" ($m^2$) and "$n$" ($number/kgC/m^2$). Integrating over mass-classes therefore yields a unit of ($m^2/m^2$).

**Reviewer**:

*P4, L1: I'm not sure why you need to state that the model conserves carbon three times. All vegetation models must conserve carbon. This isn't very surprising.*

**Response (22)**:

This subsection describes the discrete equations for RED. The discrete form is now derived in appendix A,using the conservation of mass as a constraint on the net fluxes of plants moving between the mass classes. We now make this clear at the beginning of the subsection.

**Edit**:

"We wish to produce a model of vegetation demography that can be updated numerically and which explicitly conserves vegetation carbon, providing a constraint on the number of plants moving between mass classes in the discrete form."

**Reviewer**:

*P4, L3-15: I'm not sure what purpose is served by this sequence of equations.*

**Response (23)**:

Please see response (22) above.

**Reviewer**:

*P4, L20: This equation would be easier to read if it were split into terms for seed recruitment and growth.*

**Response (24)**:

The equation as written shows that seedling recruitment depends on the fraction of net-assimilate which goes into seedling production ($\alpha$), the net-assimilate ($P$), and the fractional gap area (s). We have added a sentence below the equation to make this clearer.

**Edit**:

"Therefore, the rate of recruitment $F_0$ is the ratio of a fraction of the carbon assimilate allocated to reproduction, $\alpha P$, and $m_0$, multiplied by the gap area $s$."

> **Reviewer**:
> *P4, L23: The PPA assumes minimum overlap of crowns within each layer of the canopy. It distinctly does not assume no overlap of PFTs. It assumes that canopies are arranged into layers and within each layer there is no overlap. Competition for light occurs at the boundary of the layers and is a strong control on ecosystem assembly. In fact, much of RED is highly contradictory to the PPA concept, given the MST rejects the need to different growth parameters as a function of light availability (as demonstrated convincingly for tropical forests by Farrior et al. (2016)). I think it's thus a little disingenuous to cite the PPA here as a justification for this assumption.*

**Response (25)**:
Based-on the reviewers' own comments here, it sounds to us like the minimum overlap assumption in RED and PPA are indeed related. However, the potential relationship to PPA is just an aside , so we have removed the reference to PPA here to avoid further concern from the reviewer on this point.

> **Reviewer**:
> *P4, L25: "injected"? How do trees get injected?*

**Response (26)**:
Changed to "recruited".

> **Reviewer**:
> *Figure 1: I don't find figure 1 particularly informative. It would be better to have a depiction of the actual area available for seeds and to illustrate how the different PFTs might affect the allocation to each PFT. This figure just tells me that shrubs are smaller than trees.*

**Response (27)**:
Others who have seen this diagram have found it useful, so we have retained it despite the reviewer's opinion on this point.

> **Reviewer**:
> *P5, L4: The calculation of the area occupied by each PFT, as it is introduced here, needs a lot more explanation. In the description on L16 of P3, it simply states that the area of all the mass classes is added together, such that there is no overlap between the canopies of the trees in each plant type. This implicitly assumes that all the trees are in the 'canopy' layer, (using PPA terminology) and thus by implication that they should all get the same amount of light. Of course, via use of equation 2, the actual light environment of the plants*

*is divorced from assumptions about their spatial arrangement, but it seems like a strong assumption to me to include no possibility of additional canopy layers. What happens when the total amount of space occupied by the plants exceeds the ground area available?*

**Response (28)**:

Our gap boundary condition given by equation 12 ensures that there are no steady-state solutions where the total vegetated fraction exceeds one. We have added a sentence to clarify this point.

**Edit**:

"This 'gap' boundary condition results in there being no equilibrium solution where the amount of coverage exceeds 1. Doing so would halt the recruitment flux such that mortality processes would bring the fractional coverage back below unity."

**Reviewer**:
*P5, L5: It should be noted here that the Cox 2001 paper is at-least inspired by the Lotka-Volterra approach, to better allow connection of this concept to community ecology literature.*

**Response (29)**:

Agreed. Rewritten as "..this is a similar competition regime to the Lotka-inspired TRIF-FID model..."

**Reviewer**:
*P5, L7: Later on you state that the coexistence between PFTs of the same type doesn't actually work, so this statement that Eqn 12 allows for coexistence is a little misleading.*

**Response (30)**:

Here we mean by "inter-functional group" we mean tree-shrub-grasses. We make that clearer in a revised sentence.

**Edit**:

"...allows for the co-existence between inter-functional groups (trees, shrubs and grasses) of PFTs. For instance, a PFT such as Broadleaf Deciduous Tree can co-exist with a Deciduous Shrub and C3 Grass."

**Reviewer**:
*P5, L8: This allows succession as you note, but only between the PFT of different classes, not within a given class, unless I'm mistaken? . Figure 3: I'm not really sure what this Figure is supposed to illustrate. What are the red dotted lines in the middle of the triangle? There are three heavy double headed black arrows and not one (as implied by the legend).*

**Response (31):**
Figure 3 shows that the RED equilibrium state can be determined using observed areal cover plus either growth or mortality rate. We have removed this figure.

> **Reviewer:**
> *Eq 28 and 29: These equations need a bit more explanation and description. This section feels like you are making a concerted effort to lose readers. Is it really necessary that everyone understands how the equilibrium solution of the model is derived? Could this go in an appendix?*

**Response (32):**
Equation 28 is important in the derivation of the analytical equilibrium. We have now moved the derivation into the description (Appendix B2) and have included more explanation on the mathematical expressions:

**Edit:**

To solve for the discrete model equilibrium, we start from the flow equation from Eq.(4) with the term $\partial N/\partial t \to 0$:

$$\gamma N_i + F_i = F_{i-1} \tag{2}$$

considering the population flux - equation (5), we find $N_i$ in relation to the lower mass class, $N_{i-1}$:

$$N_i = N_{i-1} \left[ \frac{g_{i-1}/(m_i - m_{i-1})}{g_i/(m_{i+1} - m_i) + \gamma} \right] = N_{i-1}\lambda_i \tag{3}$$

(Further down in appendix B2)

An expression for the total stand density at equilibrium, $N_{\text{eq}}$, can be derived. Using equation (B.18), we can represent any population of mass class $i$ in terms of the lowest mass class $N_0$:

$$N_i = N_0 \prod_{j=1}^{i} \lambda_j \tag{4}$$

Therefore, when finding the total number of stands relative to $N_0$ we get:

$$N_{\text{eq}} = N_0 \left[ 1 + \sum_{i=1}^{I} \prod_{j=1}^{i} \lambda_j \right] = N_0 X_N \tag{5}$$

where $X_N$ describes the sum of the all mass classes as a proportion of $N_0$.

(Further down in appendix B2)

We can repeat the same process for coverage:

$$\nu_i = N_0 a_i \prod_{j=1}^{i} \lambda_j \tag{6}$$

and using allometric relationship (equation 3):

$$\nu_i = N_0 a_0 \left( \frac{m_i}{m_0} \right)^{\phi_a} \prod_{j=1}^{i} \lambda_j \tag{7}$$

This gives the total coverage, $\nu_{\text{eq}}$ as:

$$\nu_{\text{eq}} = N_0 a_0 \left[ 1 + \sum_{i=1}^{I} \left( \frac{m_i}{m_0} \right)^{\phi_a} \prod_{j=1}^{i} \lambda_j \right] = N_0 a_0 X_\nu \tag{8}$$

> **Reviewer**:
> *P10, L1: As I said above, I am highly skeptical that this is a robust way of estimating turnover, given the uncertainties to do with seed production and spatial extent.*

**Response (33)**:
As response (9) and (15) states, we have conducted a sensitivity analysis for the RED equilibrium mortality within the new appendix C.

> **Reviewer**:
> *P10, L8: So, productivity was derived from JULES using TRIFFID? Were the outputs saved for each month? Is there interannual variability? This needs a bit more detail.*

**Response (34)**:
We have appended more detail around the UKESM input:

**Edit**:

"The UKESM simulation provides NPP and local litterfall per unit area of each PFT. We multiply by PFT fraction to get the grid-box mean values required to drive RED (using ESA landcover data, as explained below)."

> **Reviewer**:
> *P11, L3: Is this really how succession works in Amazonian forests? I think it's really mostly trees that are present in the formation of small to medium sized gaps.*

**Response (35)**:
In typical succession you see the establishment of faster-growing PFTs (C3, Esh), but ultimately slower-growing trees often dominate. We see this sort of successional dynamics in RED transient simulations, as shown in Figure 4 (now Figure 10).

> **Reviewer**:
> *P11, L5: Can you illustrate the dependance on alpha and m0?*

**Response (36)**:
See responses (9), (15) and (33). This is now done in appendix C.

> **Reviewer**:
> *P12, L11: What are we to take from this illustration of 'succession' in the model? There isn't any comparison with data, nor an illustration that the model fixes the issue of slow recovery from disturbance that was raised in the introduction. What controls the area fractions of the smaller PFTS? Is there always some gap fraction dedicated to them? How is this equilibrium maintained?*

**Response (37):**

Data on forest regrowth is unfortunately difficult to find. However, this successional sequence is broadly consistent with ecological understanding and other DGVMs. We show it here to demonstrate the dynamical nature of the model. Sub-dominant PFTs occupy space left by dominant PFTs (as determined by our gap lower boundary condition - equation 12). For all PFTs the equilibrium is maintained as a balance between mortality and seedling recruitment (which is dependent on net growth-rate and competition through equation 11). We have added text below Figure 4 (now Figure 10) to clarify.

**Appended onto the bottom of the caption on Figure 10:**
"The ultimate steady-state is determined by the balance between recruitment and mortality (equation (6)). Intra- and inter-PFT occurs here through the shading of seedlings, which implies that just a fraction of the gridbox (s, 'space' or 'gap' fraction) is available to grow seedlings (equation (7))."

> **Reviewer:**
> *Figure 6: The inputs of productivity taken from JULES do not, for example, allow BETs to grow outside of the tropics, and so many of the critical questions related to the prediction of biome boundaries that are asked of DGVMs cannot be addressed in this circular analysis.*

**Response (38):**

In fact JULES does allow BET to grow outside of the tropics. We have revised the colour scale of Figure 6 (now 5) to make this clearer. However, we also sense that the reviewer is under a false impresssion about the nature of RED. RED is a model of forest demography that is driven by net growth-rates and mortality rates that can come from land-surface models or observations. In this study we have driven RED with fluxes from the JULES land-surface scheme, but the current paper is not about JULES or even JULES-RED. We clarify this point in the model introduction (as per response 34).

[Figure]

Figure 2: Mean net assimilate $P$ assimilate (equation (8)) from UKESM between 2000-2010. The mean is constructed by setting any negative growth rates to zero.

> **Reviewer:**
> *P14, L1: It seems that reproducing the PFT map should be a trivial matter*

*given the productivity inputs illustrated in Figure 6.*

**Response (39)**:

We are simply performing a model inversion to analytically solve for the RED dynamical steady-state. We have not seen this approach from another stand-alone DGVM.

> **Reviewer**:
> *P 16 L8 This is confusing because the reference to Figure 10 comes before it is described. The use of the mortality rates in these simulations is not described in this section until now.*

**Response (40)**:

We have moved the mortality section so that it is now before the global dynamical plot.

> **Reviewer**:
> *P17, L1: To what does this 'diagnosed mortality rates' refer? Isn't this sentence about diagnosing mortality rates? This adds another layer of confusion onto my previous comment.*

**Response (41)**:

We can see how this might be confusing. Therefore, we have rearranged the sections to be more clearer (Modelling setup → Equilibrium mortality rates → Local simulation → Global simulations).

> **Reviewer**:
> *Figure 9: This color map does not allow one to distinguish between most of the lower turnover areas. You need some sort of logarithmic variation in color with mortality rate.*

**Response (42)**:

As suggested, we have now used a logarithmic color map in Figure 9.

[Figure]

Figure 3: Diagnosed maps of mortality rates $\gamma$ for each PFT, as required for consistency with the ESA observations and the UKESM growth rates. White areas correspond with zero coverage and/or zero growth.

> **Reviewer**:
> *Referee: P17, L8: How influential is the minimum recruit size? This needs to be illustrated.*

**Response (43)**:
As stated in (9), (15), (33) and (36) we have carried out a sensitivity analysis in appendix C which includes the sensitivity to $m_0$.

> **Reviewer**:
> *P17, L10: The sentence that begins "Under the assumption" isn't a whole sentence. Moreover, what is the aim of defining a 'healthy' environment? You need to state what you are trying to achieve first. . .*

**Response (44)**:
We have rewritten this statement for clarity. The use of 'healthy' is indeed rather vague – so we have clarified this to 'dominant'.

> **Reviewer**:
> *P17, L12: This is a very quick and potentially confusing switch to discussing the growth is as this mortality ratio and not mortality (you should maybe also re-state what is ($\mu_0$??) a non-standard quantity.*

**Response (45)**:
We think reviewer means $\mu_0$ (there appears to be an error in the reviewer document)? We have defined $\mu_0$ before this sentence (within the steady state section). $\mu_0$ is the ratio of mortality to growth-rate ($m_0\gamma/g_0$).

> **Reviewer**:
> *P17, L13: This number seems extraordinarily high for the stem turnover rate of tropical forests? Comparison with data is, of course, where this aggrega-*

*tion idea is problematic, as mortality rates have clearly been shown to vary with tree size (Lines et al., 2010; Johnson et al., 2018), and thus the range of tree size with which one can compare these rates is unclear, particularly the lower size boundary.*

**Response (46):**
The reviewer appears to be confusing $\mu_0$ with $\gamma$, we therefore have clarified $\mu_0$ as per response(45). The sensitivity of the lowest boundary and the derived $\gamma$ see the sensitivity analysis with $m_0$, (as with previous responses (9), (15), (33) and (36)) see appendix C. Assumptions of size independence of mortality has provided credible fits of size structure of the entire US forest inventory database (Moore et al., 2018) and plots across the tropics (Muller-Landau et al., 2006b; Lima et al., 2016; Moore et al., 2020). Interestingly, within Johnson et al. (2018) (Supplementary Figure 10), the plots have a similar mortality distribution within our papers analysis .

> **Reviewer:**
> *P17, L13: Table 3 contains goodness of fit metrics, and not estimates of mortality.*

**Response (47):**
A simple typo – we should have referenced Table 4 here. Now corrected.

> **Reviewer:**
> *P17, L15: The 'value within the paper' doesn't state which paper, nor why it needs converting. Thus is very confusing.*

**Response (48):**
We have made this clearer.

> **Reviewer:**
> *Referee: P18, L1: This text on the differences between the Moore paper value and this value (which are indeed extraordinarily close and probably don't need excusing) would be better spent describing first how the Moore method differs from RED. This section assumes the reader is familiar with, for example, the non-discretized nature of the Moore method.*

**Response (49):**
We have outlined the non-discretized form within Appendix B, which we now refer to and have added text to explain the relationship to the Moore et al. (2020) paper.

> **Reviewer:**
> *P18, L5: "Potentially providing a future constraint on ESM growth rates for PFTs." is not a whole sentence.*

**Response (50):**
We have removed this sentence.

**Reviewer**:
*Figure 10: The mortality numbers in figure 10 for tropical forests seem too high. (0.07- 0.08). Again, it's hard to know what mortality rates they can be compared to. In Table 4, the numbers are different from the figure, perhaps because they are area weighted, but this isn't really clear from the text.*

**Response (51)**:
This is because of a difference of sampling - "non-zero" grid-box fractions (figure 10) versus top quartile grid-boxes . We now use the same subset for all grid-boxes to calculate the mortality rates in figure 10 (now figure 7) (the top quartile of non-zero grid-box fractions) and make it more obvious what we are doing.

[Figure]

Figure 4: Diagnosed mortality rates for (a) trees, (b) grasses and (c) shrubs in the top quartile of coverage. Notches within the box represent the confidence bounds of the median. The confidence bounds are estimated using a bootstrap method. Bracketed numbers represent the number of grid-points.

**Reviewer**:
*P18, L11: I'm not sure what "within the top 25% of coverages" means, nor what this is trying to achieve. Further, there is no data in figure 10, so I am not sure why one is supposed to conclude that the model captures the data well. Maybe you actually mean figure 11, which reduces the RED estimate, but only down to about double the observations. Given the a doubled mortality rate is approximately equal to a halved biomass, I'm not sure that this provides a very convincing validation. Further, many estimates of mortality are lower than this. Lewis et al. (2004) find mortality rates of tropical forest from 1.5-1.7%, for example.*

**Response (52)**:
There are a few things the reviewer raises here – the seemingly arbitrary "top 25%" of coverage and the fitted mortality rates being too high. Firstly, we picked this threshold to identify areas where PFTs have greater coverage – and therefore mortality rates hypothetically closer to an undisturbed baseline. We have included a sensitivity analysis in Appendix C of how the diagnosed mortality rates depend on other model parameters (as stated in our previous responses (9), (15), (32), (34) and (40).

**Reviewer**:
*P20, L4: I could not find a definition of DET prior to this usage here.*

**Response (53)**:
We now refer to appendix B and have stated the definition of DET in the introduction.

**Edit**:

" This paper presents a simplified cohort model (*Robust Ecosystem Demography (RED)*) which updates the number of trees in each mass class, but does not separately track tree-age or patch-age. RED assumes that the tree size-distribution of a forest is determined by how the rates of tree growth and mortality vary with tree size (Kohyama et al., 2003; Coomes et al., 2003; Muller-Landau et al., 2006b; Lima et al., 2016). We follow many other studies in assuming that tree-growth rates vary with the three-quarter power of tree mass ($m^{3/4}$), as suggested by metabolic scaling theory (West et al., 1997). Where tree mortality rate can also be assumed to be approximately independent of tree mass, the demographic equation yields equilibrium tree-size distributions which follow a Weibull distribution. This is sometimes termed *Demographic Equilibrium Theory (DET)* (see Appendix B). These simplifications significantly reduce the number of free parameters in RED, but still enable it to fit forest inventory data in North America (Moore et al., 2018) and South America (Moore et al., 2020). "

**Reviewer**:
*P22, L1-10: I'm not sure what to take from this section about fire. The last line seems to suggest that RED overestimates fire mortality, when figures 12 and 13 seem to show the opposite. The logic of this section needs tightening.*

**Response (54)**:
The purpose this section is to investigate if we see a raised mortality rate in regard to areas with fire disturbance and land-use. We have now changed figure 12 (now figure 9) and removed figure 13 to indicate this more clearly and rewritten the paragraph:

**Edit**:

There is a need to better understand the influence of mortality arising from disturbance events such as droughts and fire in order to constrain model projections (Pugh et al., 2020). Here we investigate if the equilibrium mortality rates implicitly capture areas of disturbances, by comparing the mean tree mortality rate to fire and land-use surveys (the mean mortality is defined here by weighting grid-box $\gamma$ values by grid-box fractional coverages). There are a number of surveys relating stand mortality in regions prone to wildfires (Swaine, 1992; Kinnaird and O'Brien, 1998; Peterson and Reich, 2001; Van Nieuwstadt and Sheil, 2005; Prior et al., 2009; Staver et al., 2009; Brando et al., 2014). In a broad sense, post-fire mortality rates can range from $0.06$ yr$^{-1}$ to catastrophic rates around $0.8$ yr$^{-1}$ and can vary quite considerably depending on tree species, fire frequency and drought severity. The drought-fire interaction is responsible for significantly increasing mortality post-fire and can be a driving cause of regional die-back (Allen et al., 2010; Brando et al., 2014). Using the ESA FIRE_CCI dataset (Chuvieco et al., 2019) we can estimate the burnt vegetation fraction per year. Taking the average burnt vegetation fraction for the months between 2000 and

2010, and converting into annual burn rate we gain an estimate of fire severity.

Another key issue is anthropogenic land-use and land-use change (Nepstad et al., 2008; Haddad et al., 2015). Fragmentation of natural forests is understood to raise the mortality of the remaining forest and to decrease the overall resilience of the ecosystem (Esseen, 1994; Laurance et al., 1998; Jönsson et al., 2007). In order to maintain a near-constant agricultural fraction, regular disruption such as grazing is needed to prevent re-colonisation and secondary succession (Dorrough and Moxham, 2005; Van Uytvanck et al., 2008; Chaturvedi et al., 2012). We carry out a comparison with land-use using the 2000 ESA LC_CCI inferred crop coverages (Li et al., 2019).

In Figure 9, we see the derived observations for burn area (a) and crop fraction (b), along with the derived mean $\gamma$ for the tree PFTs (c). From Figure 9 (d), we see that there are areas of large mortality ($\gamma > 0.075 \ \mathrm{yr}^{-1}$) that do correspond to areas where we see large fire activity (burn rate $> 0.1 \ \mathrm{yr}^{-1}$) and increased crop fraction ($> 0.25$). However, large burn rates are seen to overlap in parts of central Brazil around the Cernado region, Southern Africa and North Western Australia where fires are understood to play a significant part within the ecosystem (Coutinho, 1990; Medeiros and Miranda, 2008; Prior et al., 2009; Staver et al., 2009). There are also some areas of agriculture which correspond to deforestation, such as in the Atlantic forests of Brazil and in Indonesia (Higuchi et al., 2008; Curran et al., 2004). Areas of increased disturbances result in grasses and shrubs dominating (Figure 3).

Analysis of the RED equilibrium is an indirect approach to estimating tree mortality based on simple yet mechanistic principles of demography, and relying on few inputs (vegetation cover and assimilate). It is however conditional on the assumed estimates of vegetation coverage and net rates of assimilation.

[Figure]

Figure 5: Comparison of diagnosed mortality rates, with observation-based maps of fire and land-use. (a) annual burnt area fraction from the ESA FIRE_CCI dataset; (b) crop fraction from the ESA LC_CCI 2000 dataset; (c) diagnosed mortality rate $\gamma$ for the tree PFTs (BET-Tr, BET-Te, BDT, NET, NDT); (d) overlap of areas of higher tree mortality rates ($\gamma > 0.075$ yr$^{-1}$) with areas of fire (Burnt Area $> 0.1$ yr$^{-1}$) and agriculture (Crop Fraction $\geq 30\%$).

**Reviewer**:
*P23, L1-6: This, and the paragraph above, are in need of more references.*

**Response (55)**:
As per the response above (54), we have now included more references.

**Reviewer**:
*P23, L6: This statement about patch merging is incorrect in its assertion that patches can only be merged after a certain age in ED-type models. Further, it does not illustrate that this is actually problematic, and simply asserts as*

*such. Fusion criteria are indeed to some extent arbitrary, but that this is a genuine problem has not actually been demonstrated.*

**Response (56)**:
We have now removed the statement. As stated previously (response (5)), we have now sought to discuss both the pros and cons of RED relative to other DGVMs.

> **Reviewer**:
> *P23, L7: Which important features is it designed to capture exactly? This hasn't really been stated.*

**Response (57)**:
We have now more clearly stated the important features of second-generation DGVMs within our updated introduction and discussion sections (as stated in response (5)).

> **Reviewer**:
> *P23, L16: Metabolic scaling theory has been widely debunked by numerous studies comparing its predictions with observations (Muller-Landau et al., 2006a; Russo et al., 2007; Coomes et al., 2011; Rüger and Condit, 2012) in particular where asymmetric competition for light (e.g. in forests) is important.*

**Response (58)**:
The reviewer is perhaps confusing metabolic scaling theory for tree growth-rate as a function of tree mass ($g \propto m^{3/4}$), with an extension of metabolic scaling-theory to simulate forest demography (Brown et al., 2004). Observed tree-size distributions do not seem to be consistent with the latter, but do seem consistent with the former, as discussed in Moore et al. (2020). We have revised the introduction to clarify (response 53).

> **Reviewer**:
> *P23, L18: I am not sure how the seed model allows you to capture the effects of light competition. It allows you to represent the impacts of recruitment competition, but seems to me that it explicit does not include light competition.*

**Response (59)**:
Agreed that we do not explicitly represent light competition. We have now removed this statement for the sake of clarity.

> **Reviewer**:
> *P24, L1: It is stated here that equation 12 is a promising method to deal with the problems of coexistence in RED, but equation 12 is already part of RED, thus how can it be the solution? Further, I do not know what 'gap boundary conditions' refers to here.*

**Response (60)**:
Co-existence can be achieved by having competition coefficients less than 1. This model paper is RED version 1, there is always scope for future improvements to the model in

this topic.

> **Reviewer**:
> *P24, L4: I am skeptical, without further much more robust testing and illustration, that these relationships would be meaningful.*

**Response (61)**:
Noted. We hope that future work involving closed-form DET and RED will help provide some illustration. See our responses concerning the sensitivity of this method for determining $\gamma$.

> **Reviewer**:
> *P24, L13: I do not think that this model is 'based on' the ideas of the PPA in any meaningful way. The idea of the PPA is primarily concerned with how trees fill space, which is specifically ignored by RED, and also on the division of the canopy into discrete layers, which is definitively at-odds with the metabolic scaling method of disaggregating production solely based on tree size.*

**Response (62)**:
As per response (4) and (25) we have now edited and removed the specific mentions of PPA from the manuscript.

> **Reviewer**:
> *P24, L16: It apparently can be fitted, but I'd argue that there has been no validation presented to show that this is 'effective'.*

**Response (63)**:
We have also tested for robustness in our new appendix C. (see also response to points (9), (15), (33), (36), (43) and (52)).

**Response to Reviewer 2:**

Robust Ecosystem Demography (RED): a parsimonious approach to modelling vegetation dynamics in Earth System Models

Arthur P. K. Argles, Jonathan R. Moore, and Peter M. Cox on behalf of co-authors. (*on behalf of the co-authors*)
22nd April 2020

The referee was mainly concerned with specific detail, they did raise other points but mainly found that the paper was a bit confusing. We address each of the queries raised below. The relevant reviewer comments are written in italics below followed by our responses in plain font, changes are detailed in blue font.

> **Reviewer**:
> *The authors present a model development work on vegetation demography, and seek to incorporate it into an earth system model. The framework provides a simplified solution to model the global vegetation distribution based on the "Metabolic Scaling theory". Both the topic and the model concept are very interesting. However, there are numerous errors and ambiguous expressions throughout the current manuscript. The model descriptions are not clear enough, especially for the equations and units. At some points, I have to stop to calculate the units of each term. I'm also not fully convinced by the model outputs and validations. Extra information are necessary to be provided for a proper judgement, e.g., how the NPP data was created, which climate forcing and vegetation map were used. I suggest an overall revision and reorgnization of themanuscript. My major question about this approach is how it can be used in transit-time simulations, especially for the future projections. From a modelling aspect, the model simply ignored many factors that can be modfied due to climate change. Nevertheless, it would be very exciting if enough evidences support that some important emergent properties from land ecosystems would remain constant in a fast changing world.*

**Response (1)**:
We thank the reviewer for their comments and have sought to make edits that make the model paper clearer and help clarify definitions. On the point of the NPP data, we ran RED offline using outputs from the UKESM climate model. UKESM calculates phenology and litter fluxes using climatic data per area of each PFT (rather than per gridbox). We have elaborate further within the discussion on how RED can be used in transient simulations of future climate simulations. RED was built to be parameter sparse to reduce uncertainty at the global level. As seen from the results within the paper it is possible to capture regional vegetation accurately even within such a parasimonious model.

**Edit**:

"RED is currently being coupled to the JULES Land Surface Model, replacing TRIFFID as the default DGVM within that framework. In parallel, significant improvements are being made to the representation of physiological processes in JULES, most notably through the representation

of non-structural carbohydrate ('SUGAR', Jones et al. (2019)), and through the inclusion of a coupled model of stomatal conductance and hydraulic failure under drought stress ('SOX', Eller et al. (2018, 2020)). Plans are also being made to derive the mortality rates for RED from the INFERNO forest-fire model (Burton et al., 2019). These developments will allow us to simulate the effects of size-dependent tree mortality rates within the near future."

**Specific Comments**

**Reviewer**:
*P1 Abstract*
*L7: cohort-based models?*

**Response (2)**:
We further elaborate on this term within the abstract.

**Edit**:

"More advanced cohort-based patch models are now becoming established in the latest DGVMs. These models typically attempt to simulate the size-distribution of trees as a function of both tree-size (mass or trunk diameter) and age (time since disturbance)."

**Reviewer**:
*..L8:These models*

**Response (3)**:
Corrected "These typically..." to "These models typically...".

**Reviewer**:
*..L14:I feel it should not be the major reason to argue that RED would be a great contribution. Only mentioning the computing cost is not convincible enough.*

**Response (4)**:
We agree that this is not the definitive reason for the development of RED. Indeed the development of RED is driven by the need to have a robust and parameter sparse model of forest demography for global applicationslaw of parsimony. We therefore state that the additional problem arising from the balance of representation of ecological processes versus the number of uncertain parameters.

**Edit**:

"This approach can capture the overall impact of stochastic disturbance events on the forest structure and biomass, but at the cost of increasing the number of parameters and ambiguity when updating the probability density function (pdf) in two-dimensions."

> **Reviewer**:
> *..L15:pdf?*

**Response (5)**:
We have appended "(pdf)" to the initial mention of "probability density function..." in the sentence beforehand.

> **Reviewer**:
> *..L19:solvable?*

**Response (6)**:
Corrected typo.

> **Reviewer**:
> *..L26:Why only compared to this dataset*

**Response (7)**:
We compare to this dataset partly because this dataset is classified using the same PFTs used within the UKESM.

> **Reviewer**:
> *..L41:2K? not clear enough, references needed*

**Response (8)**:
We have added in a reference to the Paris Agreement and changed the units to degree centigrade.

**Edit**:

"This is an important component of the total carbon budget consistent with avoiding global warming thresholds, such as 2°C (Schleussner et al., 2016)."

> **Reviewer**:
> *..L47:keep update with the new results?*

**Response (9)**:
We have now removed the reference to the GCB (Global Carbon Budget) results to streamline the introduction.

> **Reviewer**:
> *..L44-51: The logic here is unclear. I assume that the authors want to stress the large uncertainties in modeling land C budget. But the topic of the study is model development, rather than uncertainty analysis. So I suggest to use 1-2 sentences to describe the uncertainty topic and go to the model development faster.*

**Response (10)**:
We agree that the motivation for including land C budget could be more concise. Uncertainty arising from the representation and parameterisation of processes is part of the motivation for RED. We have also included more discussion of other published models (see response 36).

> **Reviewer**:
> *..L53: According to my knowledge, LUC prediction is from another sector, which is not from DGVM. Provide the LUC examples here seems irrelevant to the modeling of this study. Also, why the authors only picked examples from RCP8.5.*

**Response (11)**:
Agreed. Therefore, we only mention it in passing;

**Edit**:

"Beyond the fertilisation effect and land-use change, significant uncertainty arises from the representation of vegetation demographics such as recruitment, compeitition and mortality (Brovkin et al., 2013; Ahlström et al., 2015)."

> **Reviewer**:
> *P2*
> *..Line 2: Rewrite the sentence and focus on the topic of this study. Generally, DGVM includes biochemical, biogeographical, biophysical processes and other factors influencing vegetation.*

**Response (12)**:
We have changed the sentence to be more encompassing of what a DGVM includes.

**Edit**:

"The representation of plant communities within Earth System Models (ESMs) is achieved through the use of Dynamic Global Vegetation Models (DGVMs). DGVMs employ a variety of biophysical, biogeographical and biochemical processes to simulate growth, competition and recruitment of vegetation. The variety in the number and resolution of the processes contributes to the differences found at the Earth System level."

> **Reviewer**:
> *..Line 5: How to define complex. What about the other "complex" models.*

**Response (13)**:
We have clarified this as "individual based models".

> **Reviewer**:
> *..Line 10: Why non-individual based models cannot do that?*

**Response (14)**:

Valid point, we now have redefined this as:

**Edit**:

"In the second-instance, individual based models can explicitly represent a multitude of biological and ecosystem processes at a individual plant level (Smith, 2001; Sato et al., 2007)."

> **Reviewer**:
> *..Line 13: What is top-down models? Area based?*

**Response (15)**:

Yes, we think of top-down models as phenomenological models such as Lotka-Volterra. We clarify this point in the introduction.

**Edit**:

"DGVMs range from the simplistic, older, top-down approaches to that of complex individual-based DGVMs. For example, in the first instance the TRIFFID model (Cox, 2001) simulates the fractional area of each Plant Functional Type (PFT) using phenomenological Lotka-Volterra equations."

> **Reviewer**:
> *..Line 15: are significantly simpler and more computationally efficient(reference?).*

**Response (16)**:

We have edited the paragraph in the model description removing this statement.

> **Reviewer**:
> *Line 17: over-estimated(reference?)*

**Response (17)**:

We have now provided a reference: (Burton et al., 2019).

> **Reviewer**:
> *..Line 34: The previous paragragh only explain one benefit of RED: reduce computational cost. To me, it is at least not the major reason for the RED development. I feel it is necessary to mention the theoretical foundations for RED development, e.g., the scaling theory. Although this study is mainly about model development, the explanation of the underlying mechanisms is necessary to facilitate the understanding of the model concept.*

**Response (18)**:

A valid point. We have now stated the theoretical foundations of metabolic scaling theory. Added onto the last description of the introduction:

**Edit**:

"This paper presents a simplified cohort model (*Robust Ecosystem Demography (RED)*) which updates the number of trees in each mass class, but does not separately track tree-age or patch-age. RED assumes that the tree size-distribution of a forest is determined by how the rates of tree growth and mortality vary with tree size (Kohyama et al., 2003; Coomes et al., 2003; Muller-Landau et al., 2006b; Lima et al., 2016). We follow many other studies in assuming that tree-growth rates vary with the three-quarter power of tree mass ($m^{3/4}$), as suggested by metabolic scaling theory (West et al., 1997). Where tree mortality rate can also be assumed to be approximately independent of tree mass, the demographic equation yields equilibrium tree-size distributions which follow a Weibull distribution. This is sometimes termed *Demographic Equilibrium Theory (DET)* (see Appendix B). These simplifications significantly reduce the number of free parameters in RED, but still enable it to fit forest inventory data in North America (Moore et al., 2018) and South America (Moore et al., 2020)."

> **Reviewer**:
> *Description of the model: Overall, the equations should be carefully checked, and the units need to be added in an appropriate way.*

**Response (19)**:
The units and equations have been thoroughly checked for this study and other related papers (Moore et al., 2018, 2020). We have now moved the table of variables, definitions and units from appendix A to sit directly under the model description section.

> **Reviewer**:
> *..Line 47-49: Check the symbol consistency between equ.1 and the corresponding descriptions. I suppose the equation has been simplified – it is assumed that gamma is independent from mass level already.*

**Response (20)**:
Edited for consistency.

> **Reviewer**:
> *..Line 50: Any form of what?*

**Response (21)**:
Edited to say: "of relationship with size".

> **Reviewer**:
> *..Line 53: follows a power..*

**Response (22)**:
Corrected.

> **Reviewer**:
> *..Line 59: Correct the reference format*

**Response (23)**:
Corrected.

> **Reviewer**:
> *..Line 70: Is that a basic requirement to build a vegetation model?*

**Response (24)**:
Yes - in the context of the carbon cycle and Earth System Modelling.

> **Reviewer**:
> *..Line 86: keep unit unified throughout the MS. why using per plant per unit area previously but using explicit unit here?*

**Response (25)**:
We now declare all model variables, descriptions and units in table 1.

> **Reviewer**:
> *..Line 88: why it is a concern? To keep mass and energy balance is basic to develop a model.*

**Response (26)**:
We re-phrased this statement and have moved the discrete derivation into the appendix.

> **Reviewer**:
> *..Line 66 the area term "a" does not appear before.*

**Response (27)**:
The mean crown area "$a$'" - is defined in the previous paragraph.

> **Reviewer**:
> *P3:*
> *..Line 8: P has been defined before. Again, units miss*

**Response (28)**:
We have now added units in table 1.

> **Reviewer**:
> *..Line 17: This part is mainly derived from PPA and TRIFFID, or new for RED? If it is former, I suggest to provide main equations and introduce them briefly.*

**Response (29)**:
These equations are developed for use in RED. We have removed the reference to PPA in response to other reviewer comments.

> **Reviewer**:

*P4:..Line 1: I'm concerned about the "coupling" here. Based on the description, I feel RED has not been coupled with the ESM. Using prescribed NPP means an implicit vegetation distribution in itself. From equ.16, higher NPP would mean higher baseline growth-rate.*

**Response (30)**:
RED was run offline using NPP and litter outputs from a UKESM run, there is no coupling. The UKESM runs were in terms of PFT area instead of grid-box area, therefore multiplying by coverage circumvents this issue. We have now clarified this point in the section 3.1:

**Edit**:

"The UKESM simulation provides NPP and local litterfall per unit area of each PFT. We multiply by PFT fraction to get the grid-box mean values required to drive RED (using ESA landcover data, as explained below)."

**Reviewer**:
*..Line 53-54: What is the loss of vegetation C due to plants growing beyond the modelled mass classes*

**Response (31)**:
The truncated growth $g_I N_I$ as seen within the demographic litter equation. However, this term is negligibly small because we resolve a large mass class range that is very unlikley to be exceeded.

**Reviewer**:
*P7:*
*For the first paragraph of "Modelling results", Should it be part of the method section?*

**Response (32)**:
No we don't think so. This paragraph is part of the explicit set-up rather than then the method and helps the results section have improved 'flow'.

**Reviewer**:
*..Line 1: What tests?*

**Response (33)**:
Response: Changed "tests" to "run".

**Reviewer**:
*..Line 2: Again, I'm concerned about the use of prescribed NPP. How you get NPP? Using which climate forcing? What period of NPP you used. And most importantly, how the NPP data from JULES defines the vegetation distribution? A predefined data or from a model? All the info needs to be added for a proper judgement. If fed a similar pattern from the data: ESA LC CCI*

> *to RED, then it is not surprising that they would have the similar output as showed in Figure 7.*

**Response (34)**:
For the sake of clarity, we now state that the UKESM data is defined by unit of vegetation area rather than grid-box and include about the timescale of the dataset. We already state that this is a model inversion and is therefore essentially tuning the mortality rate within RED to fit the data.

**Edit**:

"The UKESM simulation ran on a yearly time-step, and provides NPP and local litterfall per unit PFT. We multiply by PFT fraction to get the grid-box mean values required to drive RED (using ESA landcover data, as explained below)."

> **Reviewer**:
> *..Line 10: Why choose this grid-box*

**Response (35)**:
We choose this grid-box because it demonstrates a successional tropical sequence with many PFTs from bare soil. We could have shown many others.

> **Reviewer**:
> *P16:*
> *..Line 1: Discussion. The comparisons between RED and the other similar models are needed. But before that, I think the method description needs to be greatly improved, and the corresponding results should be further clarified.*

**Response (36)**:
Agreed. We have now included a comparison to other DGVMs which include forest demography within the discuss. Further we have tried be more clearer within the model description by keeping consistency in the equations and by moving some of more mathematically excessive sections into Appendix B. In addition to the above edits on the results, we have also reorientated the sections within the results section to improve the papers flow.

**Edit**:

"In a similar vein a few other models have limited the number of cohort dimensions, for example looking at using patch-age while using allometric relationships to capture size scale. Firstly the POP model (Haverd et al., 2014), uses stand-age cohorts as the dimension for population dynamics, every time-step applying crowding and resource limited mortality rates. Another ex- ample is the ORCHIDEE-MICT (Yue et al., 2018), which disaggrates the populations of a PFT into patch "Cohort" functional types, with transitions between cohorts diagnosed when the average basal diameter passes a threshold."

**Edit**:

[revised manuscript text omitted]

Johnson, D. J., Needham, J., Xu, C., Massoud, E. C., Davies, S. J., Anderson-Teixeira, K. J., Bunyavejchewin, S., Chambers, J. Q., Chang-Yang, C.-H., Chiang, J.-M., et al. (2018). Climate sensitive size-dependent survival in tropical trees. *Nature ecology & evolution*, 2(9):1436–1442.

Jones, S., Rowland, L., Cox, P., Hemming, D., Wiltshire, A., Williams, K., Parazoo, N. C., Liu, J., da Costa, A. C. L., Meir, P., Mencuccini, M., and Harper, A. (2019). The impact of a simple representation of non-structural carbohydrates on the simulated response of tropical forests to drought. *Biogeosciences Discussions*, 2019:1–26.

[revised manuscript text omitted]

---

## Author Response (AR2)

Dear Professor Sierra,

We have now completed the edits to our manuscript titled: *"Robust Ecosystem Demography (RED version 1.0): a parsimonious approach to modelling vegetation dynamics in Earth System Models"*. Appended is our responses to the minor revisions suggested, in addition to a latexdiff version detailing explicit changes.

We sincerely thank you for reviewing our paper and we hope that you find our changes satisfactory.

Kind regards,

Arthur Argles and Peter Cox (on behalf of co-authors)

**Response to Reviewer 3:**

Robust Ecosystem Demography (RED version 1.0): a parsimonious approach to modelling vegetation dynamics in Earth System Models

Arthur P. K. Argles and Peter M. Cox on behalf of co-authors. (*on behalf of the co-authors*)
7th July 2020

We thank the reviewer for their comments. The relevant reviewer comments are written in italics below followed by our responses in plain font, changes are detailed in blue font.

> **Reviewer**:
> *It is obvious that a major aim for the development of this model is to compare predictions with observations from forest inventories, e.g. RAINFOR data. However, a major application of such a model would be to incorporate it as part of a larger ESM and predict land-biosphere C exchange. From this point of view, it is not clear to me how autotrophic respiration is dealt with in this model. The main input is NPP, from which autotrophic respiration is already subtracted. However, metabolic scaling theory not only deals with size dependent growth, but also with size dependent metabolism, and therefore respiration. Why is size-dependent respiration not represented in this model? Why bother to introduce only size dependent growth while whole ecosystem metabolism is not represented? I'm not asking the authors to completely reformulate their model, but to clarify why respiratory processes are not considered relevant in this new model development effort.*

**Response (1)**:
We do not ignore respiration. This point is clarified by adding the following text:

**Edit**:
"We apply the $m^{3/4}$ scaling to $P$. We therefore implicitly assume the same scaling for both GPP and plant respiration. This is consistent with observations suggesting that plant production also scales approximately as $m^{3/4}$ (Enquist et al., 1998; Niklas and Enquist, 2001)."

**Minor Comments**

> **Reviewer**:
> *- Pg 4, ln 14-15. What are the limits of integration? From zero to infinity?*

**Response (2)**:
Yes. We have added zero to infinity limits to these integrals.

> **Reviewer**:
> *- Eq. 12. It seems to me that the discretization time-step can be a major source of uncertainty. Did you test the sensitivity with respect to the size of*

$\Delta t$?

**Response (3)**:
We have indeed tested the sensitivity of our simulations to the model timestep, which is weak around our default timestep of 1 month. Text has been added to clarify this point:

**Edit**:

[revised manuscript text omitted]